# Involuntary feedback responses reflect a representation of partner actions

Seth R Sullivan[1]*, John H Buggeln[1,2], Jan A Calalo[3], Truc T Ngo[1], Jennifer A Semrau[1,2,4,5], Michael J Carter[6†], Joshua GA Cashaback[1,2,3,4,5]*†

[1]Department of Biomedical Engineering, University of Delaware, Newark, United States; [2]Biomechanics and Movement Science Program, University of Delaware, Newark, United States; [3]Department of Mechanical Engineering, University of Delaware, Newark, United States; [4]Department of Kinesiology and Applied Physiology, University of Delaware, Newark, United States; [5]Interdisciplinary Neuroscience Graduate Program, University of Delaware, Newark, United States; [6]Department of Kinesiology, McMaster University, Hamilton, Canada

## eLife Assessment

This **important** study combines a two-person joint hand-reaching paradigm with game-theoretical modeling to examine whether, and how, reflexive visuomotor responses are modulated by a partner's control policy and cost structure. The study provides a **convincing** set of behavioral findings suggesting that involuntary visuomotor feedback is indeed modulated in the context of interpersonal coordination. The work will be of interest to cognitive scientists studying the motor and social aspects of action control.

*For correspondence:
sethsull@udel.edu (SRS);
joshcash@udel.edu (JGAC)

†co-senior authors

## Abstract

We have a remarkable ability to seamlessly and rapidly coordinate actions with others, from double Dutch to dancing. Humans use high-level partner representations to jointly control voluntary actions, while other work shows lower-level involuntary feedback responses to sudden visual perturbations. Yet, it is unknown if a high-level partner representation can be rapidly expressed through lower-level involuntary sensorimotor circuitry. Here, we test the idea that a partner representation influences involuntary visuomotor feedback responses during a cooperative sensorimotor task. Using two experiments and dynamic game theory predictions, we show that involuntary visuomotor feedback responses reflect a partner representation and consideration of a partner's movement cost (i.e., accuracy and energy). Collectively, our results suggest there is top-down modulation from high-level partner representations to lower-level sensorimotor circuits, enabling fast and flexible feedback responses during jointly coordinated actions.

## Introduction

To successfully coordinate voluntary actions, humans form a representation of others to consider their partner's goals (*Schmitz et al., 2017*; *Vesper et al., 2017*; *Kourtis et al., 2014*) and movement costs (i.e., accuracy and energy; *Vesper et al., 2011*; *Török et al., 2021*). Other work has shown that the sensorimotor system modulates involuntary feedback responses based on the structure of the individual's own goal (*Nashed et al., 2012*). Yet, it is unknown if the sensorimotor system uses a partner representation to tune these rapid and involuntary feedback responses. Investigating the influence of high-level partner representations on lower-level involuntary sensorimotor responses is crucial to understanding how humans achieve coordinated interactions during rapid movements.

A representation of a partner to consider both their goals and costs has been shown to influence voluntary movements. Behavioral work examining human-human sensorimotor interactions has suggested that a partner representation influences reaction time (*Sebanz et al., 2003*), action planning (*Vesper et al., 2017*; *Török et al., 2021*), and reaching movements (*Schmitz et al., 2017*; *Vesper et al., 2017*; *Vesper et al., 2011*; *Vesper et al., 2016*; *Leibfried et al., 2015*; *Kourtis et al., 2019*). While work with a single individual has shown that humans minimize a self movement cost (*Todorov and Jordan, 2002*; *Scott, 2004*; *Tanis et al., 2023*; *Knill et al., 2011*), work with multiple individuals suggests that humans will select voluntary actions that minimize a joint cost that considers both self and a partner (*Török et al., 2021*). While these past works have broadened our understanding of voluntary coordinated actions, it remains unknown if a high-level partner representation and consideration of the partner's cost can influence lower-level involuntary sensorimotor feedback responses.

Elegant work has shown that the sensorimotor system has a remarkable ability to generate rapid and involuntary feedback responses—prior to voluntary control—that are tuned by task dynamics and goals (*Calalo et al., 2023*; *Franklin et al., 2017*; *Franklin and Wolpert, 2008*; *Nashed et al., 2012*). *Nashed et al., 2012* had participants reach to either a narrow target (task-relevant) or wide target (task-irrelevant; *Nashed et al., 2012*). The narrow target was task-relevant since participants needed to correct for lateral deviations to successfully hit the target. The wide target was task-irrelevant since participants did not need to correct for lateral deviations to hit their target. As early as 70 ms following a mechanical perturbation, they found greater muscular feedback responses when reaching to a narrow task-relevant target compared to a wide task-irrelevant target. Likewise, pioneering work by *Franklin and Wolpert, 2008* demonstrated that sensorimotor circuits also generate involuntary feedback responses to visual perturbations between 180 ms and 230 ms (*Franklin and Wolpert, 2008*). To measure involuntary feedback responses, they laterally constrained a participant's hand within a rigid force channel and recorded the lateral hand force in response to a lateral cursor jump. They found that these rapid and involuntary visuomotor feedback responses are also tuned according to relevant and irrelevant task demands. While considerable work has examined visuomotor feedback responses of a human acting alone, it is unknown whether the sensorimotor system uses a partner representation to tune involuntary visuomotor feedback responses.

Across two experiments, we tested the overarching idea that a high-level partner representation influences lower-level involuntary sensorimotor circuits. Human pairs were required to move a jointly controlled cursor into their own target. We manipulated the width of both participant targets to be either task-relevant or task-irrelevant. We measured visuomotor feedback responses following either a cursor (Experiment 1) or target jump (Experiment 2). We made a priori predictions using four unique dynamic game theory models. Each of these models tested a specific hypothesis on whether visuomotor feedback responses reflect: (i) a partner representation and, if so, (ii) a weighting of the partner cost. Collectively, our empirical and computational work provides novel insights into how humans rapidly control their actions with others.

## Results

### Experimental design

In Experiment 1 (n=48) and Experiment 2 (n=48), participants completed a joint reaching task with a partner (*Figure 1A*). Participants had vision of their own cursor, a partner's cursor, and a center cursor. The center cursor was at the midpoint of their own cursor and their partner's cursor. They also viewed their own self target and their partner's target on their screen. Participants were instructed to move and stabilize the center cursor in the self target within a time constraint. Participants received the message 'Good', 'Too Slow', or 'Too Fast' if they stabilized within their self target between 1400 ms and 1600 ms, > 1600 ms, or < 1400 ms, respectively. They were explicitly informed that their success in the task was determined by moving and stabilizing the center cursor only within the self target. Therefore, the instructions and timing constraints did not enforce participants to work together.

We manipulated the width of both the self and partner target (*Figure 1B*) to be either narrow (task-relevant) or wide (task-irrelevant). The narrow target is task-relevant since participants would need to correct for lateral deviations to hit their target. The wide target is task-irrelevant since participants do not need to correct for lateral deviations to hit their target. Thus, we used a 2 (Partner Irrelevant or Partner Relevant) x 2 (Self Irrelevant or Self Relevant) repeated measures experimental design with

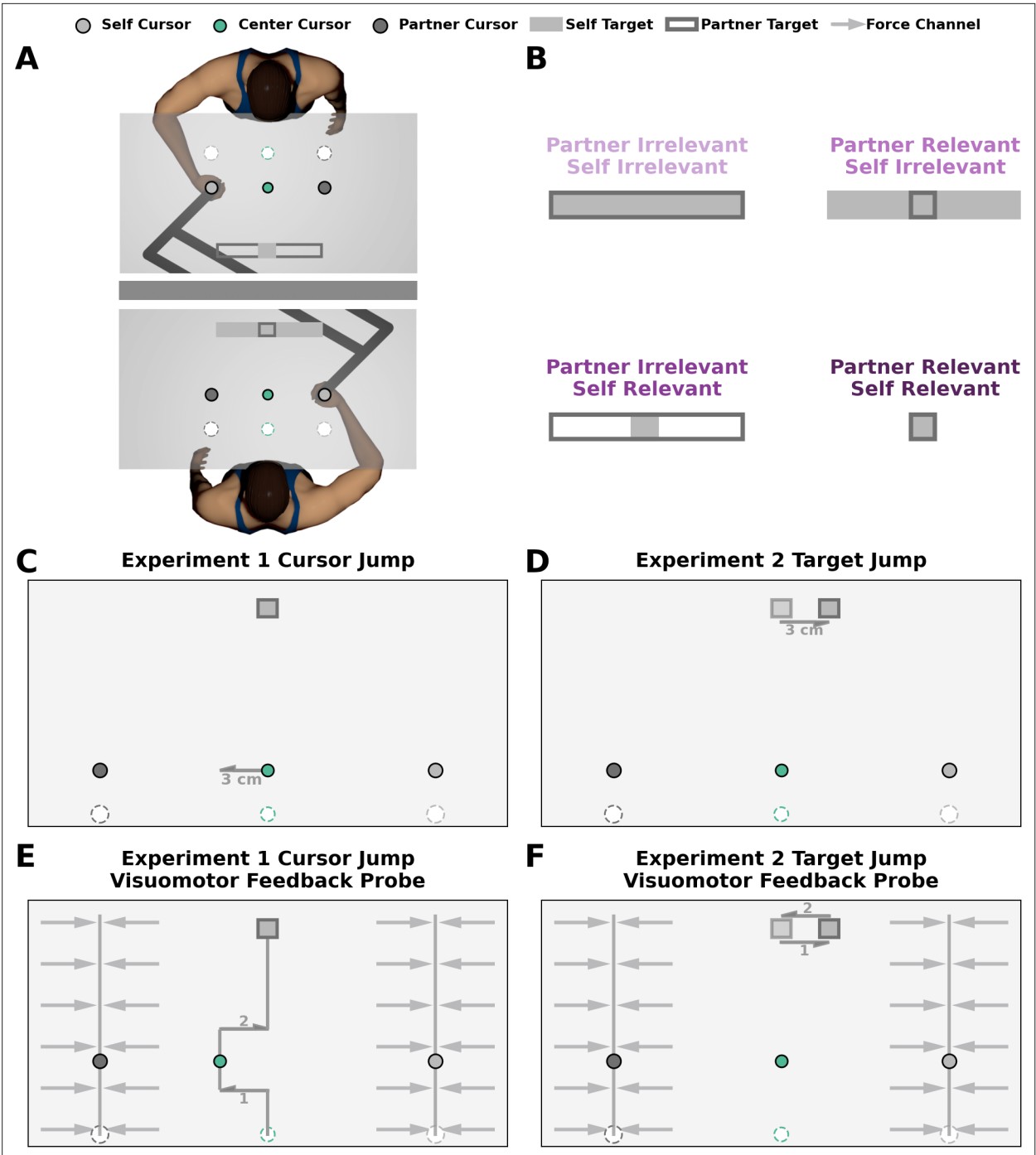

**Figure 1.** Experimental design. (**A**) In both experiments, each participant in the pair grasped the handle of a robotic manipulandum and made reaching movements in the horizontal plane. An LCD projected images (start position, targets, cursors) onto a semi-silvered mirror. Each trial began with each participant's hand (dark grey circle) within their respective start position (white circle). After a short and random time delay, the self target appeared as a filled dark grey rectangle and the partner target appeared as an unfilled light grey rectangle. Simultaneously, the center cursor (green circle) and partner cursor (light grey circle) also appeared on the screen. After a constant time delay of 500 ms, participants heard a tone that cued them to begin their reach. Participants were instructed to move the center cursor into their own target. Each participant received independent feedback once the center cursor was stabilized within their own target. (**B**) Experimental conditions. We manipulated the width of both the self and partner targets to be either narrow (task-relevant) or wide (task-irrelevant). The narrow target is task-relevant since participants would need to correct for lateral deviations to successfully complete their task. The wide target is task-irrelevant since participants do not need to correct for lateral deviations to successfully complete their task. Human pairs performed four blocked experimental conditions: (i) *partner-irrelevant/self-irrelevant,* (ii) *partner-relevant/self-irrelevant,* (iii) *partner-irrelevant/self-relevant,* (iv) *partner-relevant/self-relevant.* (**C–D**) Perturbation trials. On a subset of trials, the center cursor in

*Figure 1 continued on next page*

*Figure 1 continued*

(**C**) Experiment 1 or both targets in (**D**) Experiment 2 jumped 3 cm laterally to the left or right. (**E–F**) Visuomotor probe trials. On a subset of trials, the center cursor in Experiment 1 (**E**) or both targets in Experiment 2 (**F**) jumped 3 cm laterally for 225 ms, then jumped 3 cm back to the original lateral position. During these probe trials, the hand of both participants in the pair was constrained to a force channel. Here, we measured each participant's visuomotor feedback responses as the force (N) they applied to the wall of the stiff force channel.

four blocked experimental conditions: (i) *partner-irrelevant/self-irrelevant*, (ii) *partner-relevant/self-irrelevant*, (iii) *partner-irrelevant/self-relevant*, and (iv) *partner-relevant/self-relevant*.

The goal of Experiment 1 and Experiment 2 was to determine if a representation of a partner and consideration of their costs influences involuntary visuomotor feedback responses. To address this goal, we had participants perform non-perturbation trials, perturbation trials, and probe trials in each experimental condition. In both the non-perturbation trials and perturbation trials, participants reached freely in the lateral and forward dimensions. However, in perturbation trials (*Figure 1C–D*), either the center cursor (Experiment 1) or both targets (Experiment 2) jumped 3 cm to the right or left when the center cursor moved 25% of the forward distance to the targets. In probe trials (*Figure 1E–F*), both participants were constrained by a force channel and could only move along the forward dimension. Here, they experienced the cursor or target jump for 250ms before returning to the original lateral position. Critically, as a metric of visuomotor feedback responses, we measured the lateral force participants applied against the channel in response to center cursor or target jumps.

## Dynamic game theory model

We generated a priori predictions of hand trajectories and visuomotor feedback responses for each of the experimental conditions using a dynamic game theory model (*Figure 2A*). We modelled our task as a linear quadratic game of the form

$$x_{k+1} = Ax_k + B_1 u_{1,k} + B_2 u_{2,k}. \tag{1}$$

$x_k$ is the state (e.g., position) of the system at time step $k$, $A$ represents the task dynamics, $u_1$ and $u_2$ are the control signals, and $B_1$ and $B_2$ convert the control signals to a force that produces movement. Here, the subscripts 1 and 2 respectively refer to controller 1 and 2, representing a pair of participants in our task. Throughout, we describe the model with controller 1 as the self and controller 2 as the partner.

Controller 1 and 2 select their own control signal $u_1^*$ or $u_2^*$, which considers their respective costs. We can define individual cost functions $J_1$ and $J_2$ as:

$$J_1 = \frac{1}{2} \sum_{k=0}^{N-1} \left( x_k^T Q_1 x_k + u_{1,k}^T R_{11} u_{1,k} \right) + \frac{1}{2} x_N^T Q_{1,N} x_N \tag{2}$$

$$J_2 = \frac{1}{2} \sum_{k=0}^{N-1} \left( x_k^T Q_2 x_k + u_{2,k}^T R_{22} u_{2,k} \right) + \frac{1}{2} x_N^T Q_{2,N} x_N \tag{3}$$

Here, $J_1$ is the individual cost for controller 1 (e.g., self) and $J_2$ is the individual cost for controller 2 (e.g., partner). $N$ is the final step, which represents the end of a trial. The term $Q$ penalizes deviations of the center cursor relative to each target.

Depending on the experimental condition, we modeled (i) a task-relevant target using a higher value of $Q$ and (ii) a task-irrelevant target using a lower value of $Q$. The term $R$ penalizes the control signal ($u$), which would relate to an energetic cost. Further, we define a joint cost function as:

$$J_1^{\alpha_1} = J_1 + \alpha_1 J_2 \tag{4}$$

$$J_2^{\alpha_2} = J_2 + \alpha_2 J_1, \tag{5}$$

where $\alpha_i \in [0, 1]$ determines the degree to which controller $i$ considers their partner's cost function. The optimal control signal for controller 1 ($u_{1,k}^*$) and controller 2 ($u_{1,k}^*$) is determined by the time-varying feedback gains $F_1$ and $F_2$ that minimize the joint cost function $J_1^{\alpha_1}$ and $J_2^{\alpha_2}$, respectively:

$$u_{1,k}^* = -F_{1,k} \hat{x}_{1,k} \tag{6}$$

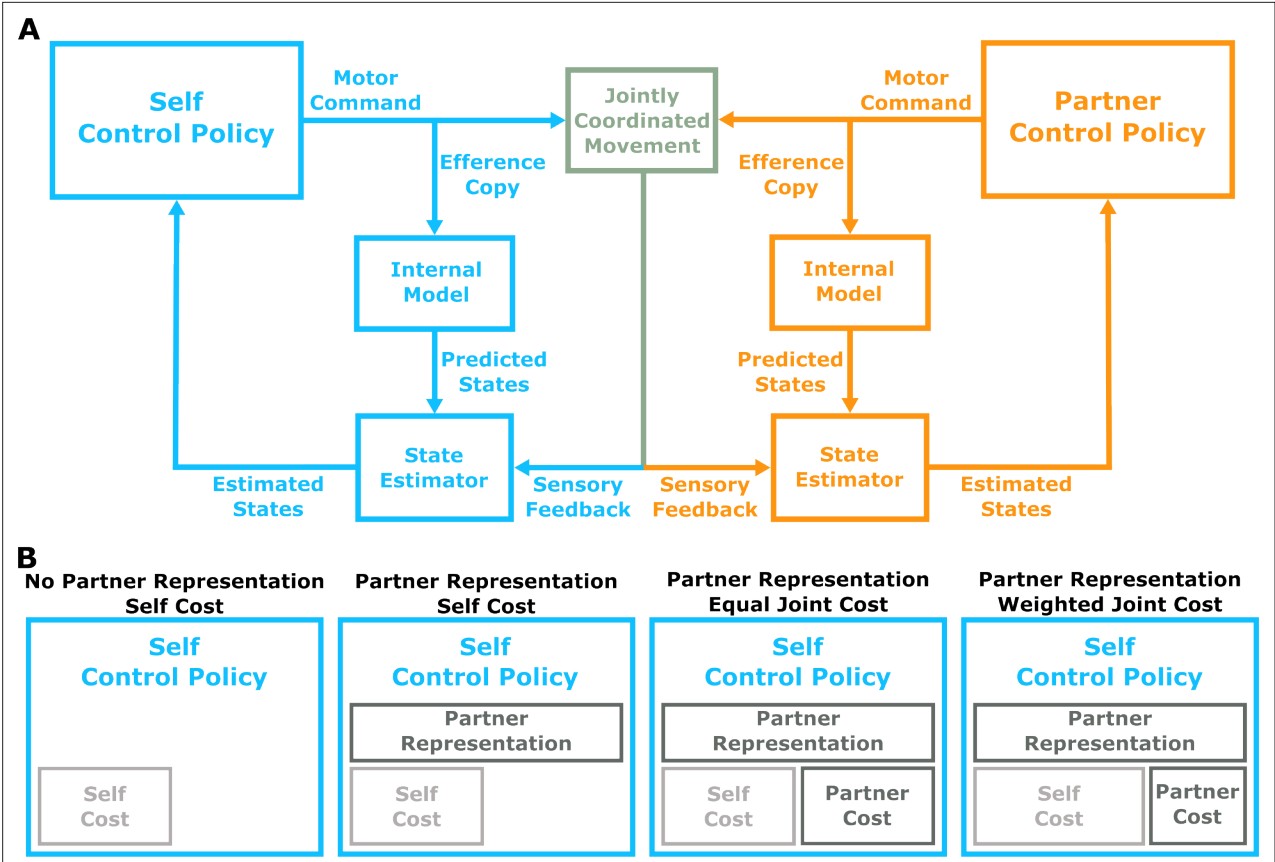

**Figure 2.** Control model framework and hypotheses. (**A**) Control model. Human pairs were modeled as controllers within a dynamic game theory framework. Here, we depict the feedback control loop from the perspective of one participant (i.e., the self). The self and partner control policy each generate a motor command to produce jointly controlled movement. An efference copy of the motor command passes through an internal model (representation of dynamics) to generate predicted states. Each controller also receives noisy and delayed sensory feedback on the states (e.g., position of the self and partner hand, center cursor, and self and partner targets). Both the self and partner controllers have a state estimator that combines the predicted state and sensory feedback in a statistically optimal manner to produce estimated states. The estimated states are used by the control policy to generate motor commands on each time step. (**B**) Hypotheses. The dynamic game theory framework allowed us to test four distinct hypotheses. The hypotheses test whether the control policy: (i) has a representation of a partner, and (ii) considers only a self cost or joint (self +partner) cost of accuracy and energy. No Partner Representation and Self Cost Hypothesis: The sensorimotor system has a control policy that does not use a representation of a partner, and only considers a self cost. Partner Representation and Self Cost Hypothesis: The sensorimotor system has a control policy that uses a representation of a partner, but only considers a self cost. Partner Representation and Equal Joint Cost Hypothesis: The sensorimotor system has a control policy that uses a representation of a partner, and equally considers both a self cost and partner cost (i.e., equal joint cost). Partner Representation & Weighted Joint Cost Hypothesis: The sensorimotor system has a control policy that uses a representation of a partner, and that weights the self cost greater than the partner cost (i.e., weighted joint cost). Each of the four hypotheses generate unique predictions of human hand movement (*Figure 3A-P*) and visuomotor feedback responses (*Figure 4*).

$$u^*_{2,k} = -F_{2,k}\hat{x}_{2,k} \tag{7}$$

Here, $\hat{x}_{i,k}$ for $i = \{1, 2\}$ is controller $i$'s posterior estimate of the state (see Methods). The feedback gains $F_1$ and $F_2$ constitute a Nash equilibrium solution to the linear quadratic game defined by *Equations 1–5*. Throughout, the feedback gains determine hand movement and visuomotor feedback responses. The Nash equilibrium solution $F_1$ that minimizes $J_1^{\alpha_1}$ can utilize knowledge of the partner's control policy $F_2$ through the coupled algebraic Riccati equations (see Appendix 3).

## Modelling partner representation

A partner representation is defined as knowledge of the partner's control policy $F_2$. That is, a person accounts for their partner's actions. No partner representation would reflect the case where $F_1$ is

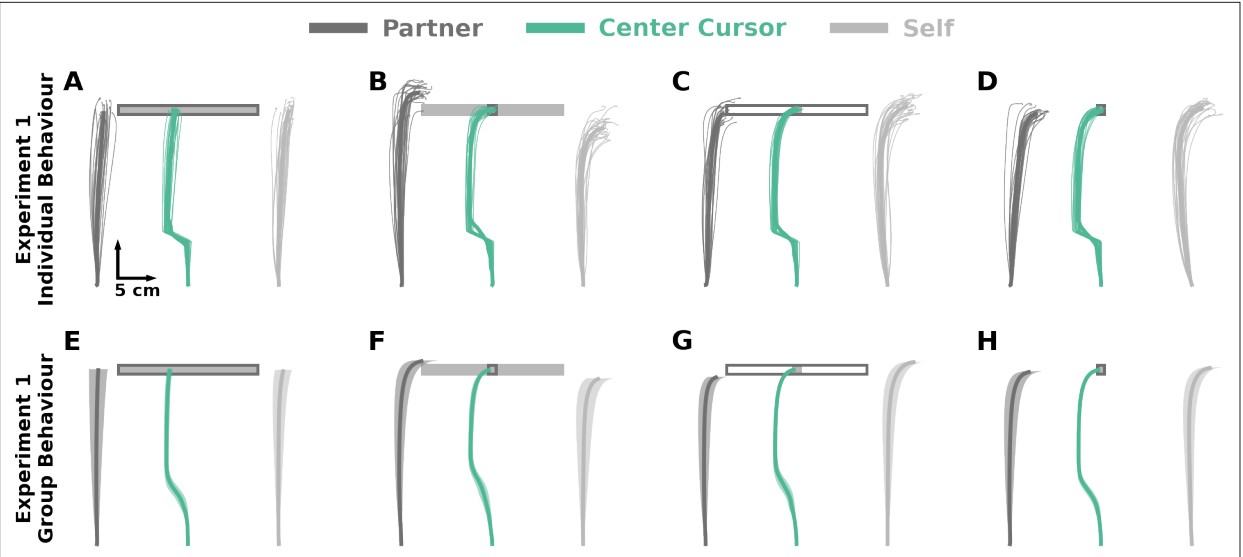

**Figure 3.** Experiment 1: Hand and center cursor trajectories. Collectively, the self cursor in models with only a self cost does not laterally deviate to correct for the cursor jump in the *partner-relevant/self-irrelevant* condition. In contrast, the self cursor in models that consider a self and partner cost laterally deviates to correct for the cursor jump in the *partner-relevant/self-irrelevant* condition. (**A–D**) Individual hand and center cursor positions of an exemplar pair for each condition in Experiment 1. Thin traces represent each trial. Thick traces represent the average across trials for the human pair. (**E–H**) Group average hand and center cursor positions in Experiment 1. Traces represent the mean and shaded regions reflect ±1 standard error of the mean.

selected under the assumption that $F_2 = 0$. More simply, a person does not account for their partner's actions.

## Modelling self and partner cost

We also modelled the degree to which a person considers their self cost, or some joint cost of both self and partner. In *Equation 19*, $\alpha_1$ determines the degree to which controller 1 considers its partner's cost. $\alpha_1 = 0$ reflects only a self cost, which would imply a person does not consider their partner's cost. Conversely, $\alpha_1 = 1$ reflects an equal joint cost that would imply a person considers their self cost and partner cost equally. Finally, $\alpha_1 = 0.5$ reflects a higher weighting on the self cost than the partner cost, implying that a person primarily considers their own cost and to a lesser extent their partner's cost.

Through our computational framework, we considered four alternative hypotheses, each testing how a partner representation and consideration of a partner's cost influences sensorimotor behavior (*Figure 2B–E*): (i) No Partner Representation & Self Cost, (ii) Partner Representation and Self Cost, (iii) Partner Representation and Equal Joint Cost, (iv) Partner Representation and Weighted Joint Cost. For each experimental condition, we used these four models to make a priori predictions of reaching trajectories (*Appendix 1—figure 1*) and visuomotor feedback responses (Figure 4).

## Hand and center cursor trajectories

An exemplar pair (*Figure 3A–D*) and group average (*Figure 3E–H*) hand and center cursor trajectories are shown for each experimental condition in Experiment 1. Note that while both participants in the pair began each trial to the right of the center cursor (see *Figure 1A*), we refer to one of the participants as the 'self' and the other participant as the 'partner' (see Methods for details). In the *partner-irrelevant/self-irrelevant* condition (*Figure 3A and E*), neither participant laterally deviated to correct for the cursor jump since both targets were irrelevant. In the *partner-relevant/self-irrelevant* condition (*Figure 3B and F*), the self cursor laterally deviated less than the partner cursor. In the *partner-irrelevant/self-relevant* condition (*Figure 3C and G*), the self cursor laterally deviated more than the partner cursor. Finally, in the *partner-relevant/self-relevant* condition (*Figure 3D and H*), both the self and partner cursor had a similar amount of lateral deviation.

The group average trajectories in both Experiment 1, Experiment 2 (see *Appendix 1—figure 2*), and final lateral hand deviation (see Appendix 2) aligned closest with the Partner Representation and

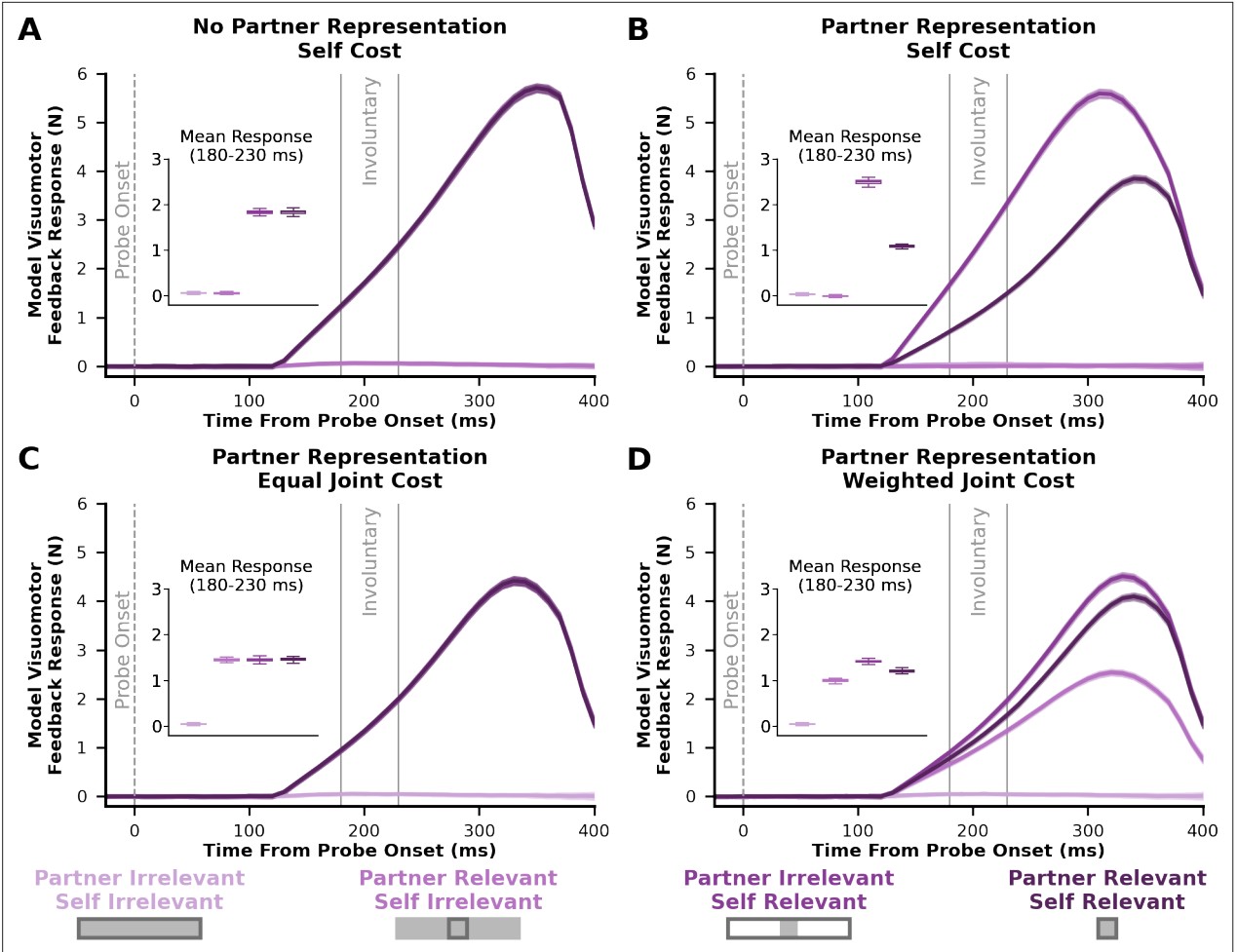

**Figure 4.** Model visuomotor feedback responses. Model predictions of visuomotor feedback responses (y-axis) over the time from probe onset (x-axis) for each condition considering the (**A**) No Partner Representation and Self Cost, (**B**) Partner Representation and Self Cost, (**C**) Partner Representation and Equal Joint Cost, and (**D**) Partner Representation and Weighted Joint Cost models. Solid lines reflect the average visuomotor feedback response to probe trials and shaded error bars reflect ±1 standard deviation of the mean. The inset axis shows the mean visuomotor feedback response between 180 ms and 230 ms, which aligns with the involuntary time epoch (*Franklin and Wolpert, 2008*). Across the different models, a greater visuomotor feedback response in the *partner-relevant/self-irrelevant* condition compared to the *partner-irrelevant/self-irrelevant* condition implies that there is a partner representation and a consideration of the partner's cost. Likewise, a lower feedback response in the *partner-relevant/self-relevant* condition relative to the *partner-irrelevant/self-relevant* condition would indicate a partner representation, as well as a higher weighting of the self cost compared to the partner cost.

Weighted Joint Cost model (see *Appendix 1—figure 1M–P*). Together, the model predictions and empirical hand trajectories support the notion that voluntary sensorimotor control reflects a partner representation and a consideration of the partner's cost.

## Visuomotor feedback responses

In these experiments, we were primarily interested in the involuntary feedback responses to visual probes. We modelled these visuomotor feedback responses computationally and measured them experimentally using cursor and target jumps.

## Model visuomotor feedback responses

We also simulated probe trials by constraining the models to a force channel and calculating the force the models produce in response to the cursor jump (see Methods: Dynamic Game Theory model). For each model and condition, *Figure 4* shows the visuomotor feedback responses over time in response to cursor jump probe trials. The inset within each of the subplots displays the average visuomotor

response between 180 ms and 230 ms, which aligns with the involuntary time epoch (*Franklin and Wolpert, 2008*).

Models that only consider the self cost predict no change in visuomotor feedback responses between the *partner-relevant/self-irrelevant* and *partner-irrelevant/self-irrelevant* condition (*Figure 4A and B*). Thus, models that only consider a self cost do not help their partner achieve their goal. Conversely, models that consider both a self and partner cost predict a greater visuomotor feedback response in the *partner-relevant/self-irrelevant* condition compared to the *partner-irrelevant/self-irrelevant* condition (*Figure 4C and D*). That is, models that consider a joint cost attempt to help a partner achieve their goal. If the involuntary sensorimotor circuits leverage a partner representation and consideration of partner costs, we would expect to see an increased visuomotor feedback response in the *partner-relevant/self-irrelevant* condition compared to the *partner-irrelevant/self-irrelevant* condition.

If there is a partner representation, there are different visuomotor feedback response predictions when the self controller has an equal joint cost versus a weighted joint cost. In the Partner Representation and Equal Joint Cost model, the self controller is willing to spend the same amount of energy to help their partner or itself achieve a goal. As a result, this model predicts no difference between the *partner-relevant/self-irrelevant*, *partner-irrelevant/self-relevant,* and *partner-relevant/self-relevant* conditions (*Figure 4C*).

On the contrary, the self controller in the Partner Representation & Weighted Joint Cost model primarily spends energy to achieve its own goal, while spending comparatively less energy to help a partner achieve their goal. During the *partner-irrelevant/self-relevant* condition, the self controller is only expecting a partial visuomotor feedback response from the partner since the partner has an irrelevant target. But in the *partner-relevant/self-relevant* condition, the self controller is expecting a comparatively greater visuomotor feedback response from the partner since the partner also has a relevant target. Therefore, the Partner Representation and Weighted Joint Cost model predicts a greater visuomotor feedback response in the *partner-irrelevant/self-relevant* condition compared to the *partner-relevant/self-relevant* condition (*Figure 4D*).

## Experiment 1: Visuomotor feedback responses

Here, we show group level visuomotor feedback responses over time (*Figure 5A*), and the average visuomotor feedback response during the involuntary (180–230 ms), semi-involuntary (230–300 ms), and voluntary (300–400 ms) epochs.

There was a significant interaction between self target and partner target (F[1,47]=61.61, p < 0.001) on involuntary visuomotor feedback responses in Experiment 1. Interestingly, we found a significantly greater involuntary visuomotor feedback responses in the *partner-relevant/self-irrelevant* condition compared to the *partner-irrelevant/self-irrelevant* condition (p < 0.001, $\hat{\theta}$ = 89.58). Crucially, these results support the idea that the involuntary sensorimotor circuits have a partner representation and a consideration of the partner's cost.

Further, there was a significantly different involuntary visuomotor feedback response between the *partner-relevant/self-relevant* and *partner-irrelevant/self-relevant* conditions (p=0.002, $\hat{\theta}$ = 62.50). A lower involuntary visuomotor feedback response in the *partner-relevant/self-relevant* condition compared to the *partner-irrelevant/self-relevant* condition further suggests a partner representation, as well as a greater weighting of the self cost compared to the partner cost.

*Figure 5C and D* show the semi-involuntary and voluntary visuomotor feedback responses. We also found a significant interaction between self target and partner target for semi-involuntary (F[1,47]=79.76, p < 0.001) and voluntary (F[1,47]=79.85, p < 0.001) visuomotor feedback responses. Follow-up mean comparisons showed the same significant differences in both the semi-involuntary and voluntary visuomotor feedback responses, as seen in the involuntary visuomotor feedback responses.

The involuntary, semi-involuntary, and voluntary visuomotor feedback responses in each condition closely match the predictions of the Partner Representation and Weighted Joint Cost model (compare *Figures 4D and 5*). Remarkably, the results in Experiment 1 suggest that a partner representation and consideration of a partner's cost not only influence voluntary behavior, but also involuntary sensorimotor circuits.

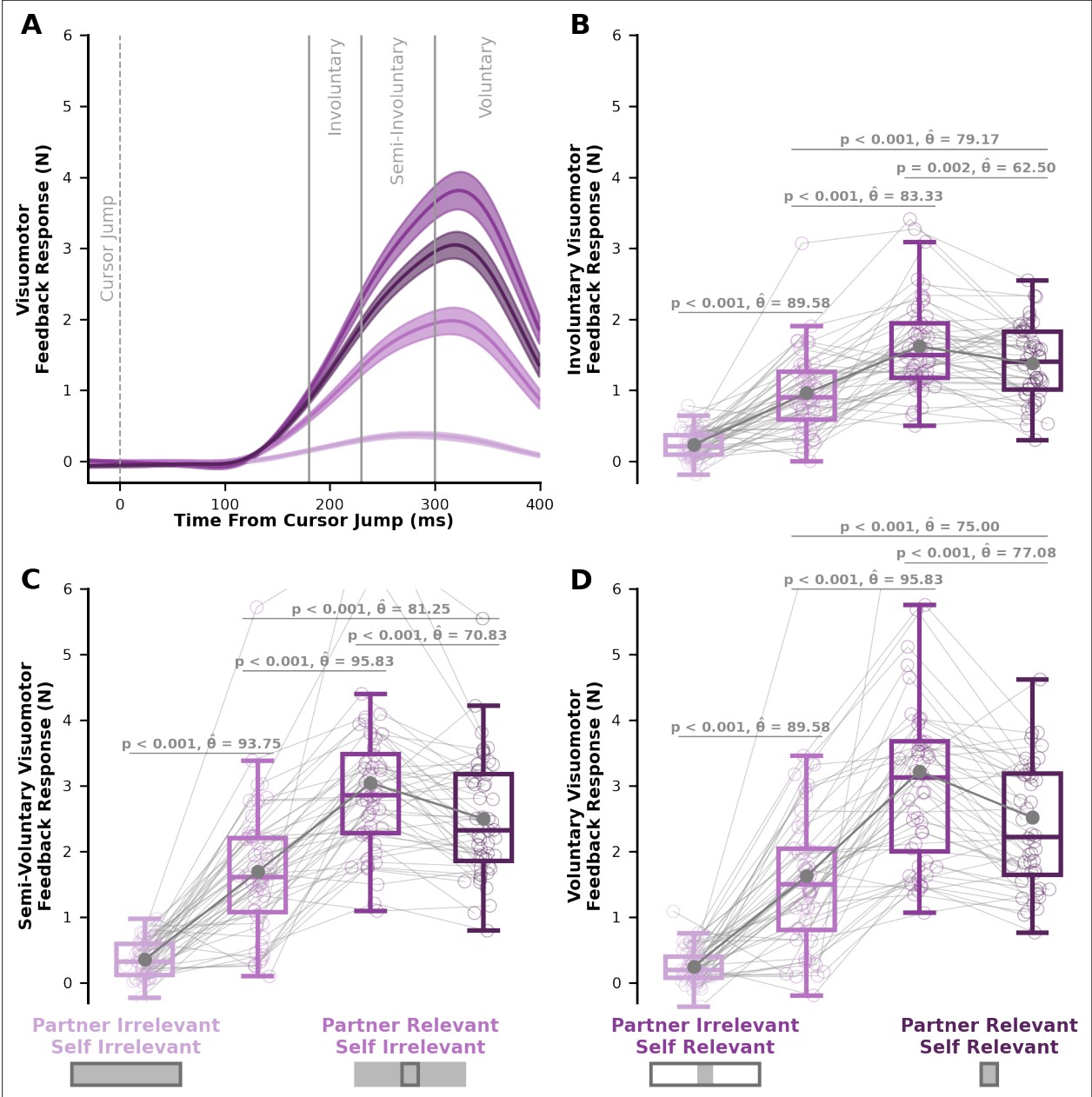

**Figure 5.** Visuomotor feedback responses in Experiment 1. (**A**) Visuomotor feedback response (y-axis) over time (x-axis), where 0ms corresponds to the initial cursor jump. Solid lines represent the group average visuomotor feedback response for each condition. Shaded regions represent ±1 standard error. Vertical grey lines separate involuntary (180–230ms), semi-involuntary (230–300ms), and voluntary (300–400 ms) visuomotor feedback responses. Average B involuntary, (**C**) semi-involuntary, and (**D**) voluntary visuomotor feedback response for each condition. Box and whisker plots show 25%, 50%, and 75% quartiles. (**B**) We see significant differences in involuntary visuomotor feedback responses between each condition, matching the predictions of the Partner Representation and Weighted Joint Cost model (see *Figure 4D*). Crucially, a greater involuntary visuomotor feedback response in the *partner-relevant/self-irrelevant* condition compared to the *partner-irrelevant/self-irrelevant* condition (p < 0.001) suggests a partner representation and some consideration of the partner's cost. Further, a smaller involuntary visuomotor feedback response in the *partner-relevant/self-relevant* condition compared to the *partner-irrelevant/self-relevant* condition (p=0.002) suggests a higher weighting of the self cost compared to the partner cost. Taken together, our results support the idea that involuntary visuomotor feedback responses express a representation of a partner, while using a joint cost that more heavily weights the self cost over the partner cost.

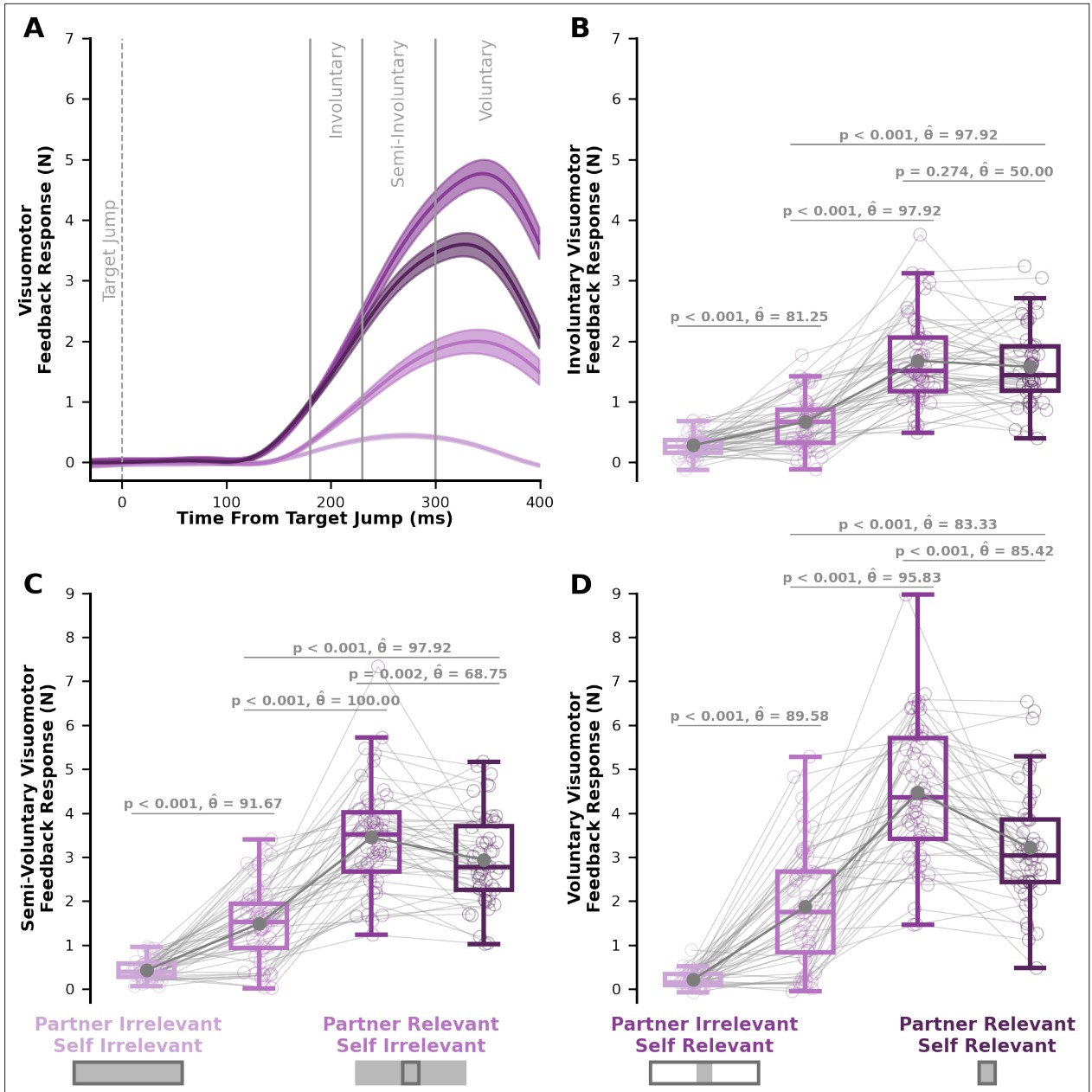

**Figure 6.** Visuomotor feedback responses in Experiment 2. (**A**) Visuomotor feedback response (y-axis) over time (x-axis), where 0 ms corresponds to the initial target jump. Solid lines represent the group average visuomotor feedback response for each condition. Shaded regions represent ±1 standard error. Vertical grey lines separate involuntary (180–230 ms), semi-involuntary (230–300 ms), and voluntary (300–400 ms) visuomotor feedback responses. Average (**B**) involuntary, (**C**) semi-involuntary, and (**D**) voluntary visuomotor feedback response for each condition. Box and whisker plots show 25%, 50%, and 75% quartiles. (**B**) Critically, a greater involuntary visuomotor feedback response in the *partner-relevant/self-irrelevant* condition compared to the *partner-irrelevant/self-irrelevant* condition (p < 0.001) suggests a partner representation and some consideration of the partner's cost.

## Experiment 2: Visuomotor feedback responses

Here, we show group-level visuomotor feedback responses over time (*Figure 6A*), and the average visuomotor feedback response during the involuntary (180–230 ms), semi-involuntary (230–300 ms), and voluntary (300–400 ms) epochs.

We found a significant interaction between self target and partner target (F[1,47]=20.54, p < 0.001) for involuntary visuomotor feedback responses in Experiment 2. Follow-up mean comparisons again showed a significant increase in the visuomotor feedback response in the *partner-relevant/self-irrelevant* condition compared to the *partner-irrelevant/self-irrelevant* condition (p < 0.001, $\hat{\theta} = 81.25$).

As shown in Experiment 1, these Experiment 2 results further support the idea that the involuntary sensorimotor circuits have a partner representation and a consideration of the partner's cost.

Between the *partner-irrelevant/self-relevant* condition and the *partner-relevant/self-relevant* conditions, we did not find a significant difference (p=0.274, $\hat{\theta}$ = 50.00). Nevertheless, the involuntary visuomotor feedback responses in each condition still most closely matched the predictions of the Partner Representation & Weighted Joint Cost model. Further, we also found a significant interaction between self target and partner target for the semi-involuntary (F[1,47]=68.82, p < 0.001) and voluntary (F[1,47]=133.04, p < 0.001) visuomotor feedback responses. Aligning with the results from Experiment 1, we found a significant difference between the *partner-irrelevant/self-relevant* condition and *partner-relevant/self-relevant* condition for the semi-involuntary (*Figure 6C*; p < 0.001, $\hat{\theta}$ = 68.75) and voluntary (*Figure 6D*; p < 0.001, $\hat{\theta}$ = 85.42) visuomotor feedback responses. Taken together, the visuomotor feedback responses in Experiment 1 and Experiment 2 closely match the Partner Representation and Weighted Joint Cost model predictions. Remarkably, the involuntary visuomotor feedback responses across two experiments support our hypothesis that a high-level partner representation and a consideration of a partner's cost influence low-level involuntary sensorimotor circuits.

## Discussion

Our primary finding across two experiments was that a partner representation and consideration of a partner's cost influences involuntary visuomotor feedback responses. Specifically, involuntary visuomotor feedback responses closely matched the hypothesis that the sensorimotor system uses a partner representation and weighted joint cost, where the self cost is prioritized more than the partner cost. Taken together, our empirical results and computational modeling support the idea that a high-level partner representation and a joint cost influence lower-level involuntary sensorimotor circuits.

In this paper, we demonstrated how a representation of a partner and consideration of their costs influences rapid and involuntary visuomotor feedback responses during a cooperative sensorimotor reaching task. In Experiments 1 and 2, we found that participants displayed increased involuntary visuomotor feedback responses when there was a relevant partner target and an irrelevant self target, compared to when both targets were irrelevant. Aligned with model predictions, these findings suggest that involuntary feedback responses reflect a partner representation and a joint cost. In Experiment 1, we found a significant decrease in involuntary visuomotor feedback responses to cursor jumps when both the self and partner target were relevant, compared to the condition with an irrelevant partner target and relevant self target. The different involuntary visuomotor feedback responses between these conditions suggest that the sensorimotor system uses a partner representation and weighted joint cost to modulate involuntary visuomotor feedback responses. Interestingly, this result suggests that the sensorimotor system modulates involuntary visuomotor feedback responses based on a prediction of a partner's control policy. Further, it highlights that high-level partner representations modulate lower-level sensorimotor circuits and are rapidly expressed via involuntary visuomotor feedback responses.

In Experiment 1 and Experiment 2, we found the same significant differences between conditions for the semi-involuntary and voluntary visuomotor feedback responses. However, in Experiment 2, we did not see a decrease in involuntary visuomotor feedback responses to target jumps when both partner and self targets were relevant, compared to an irrelevant partner target and relevant self target. One possibility for this finding is that there may be longer visuomotor feedback response latencies to target jumps compared to cursor jumps (*Scott, 2016*; *Brenner and Smeets, 2003*; *Dimitriou et al., 2013*). However, other work by *Franklin et al., 2016* found no difference in visuomotor feedback response latencies between cursor and target jumps (*Franklin et al., 2016*). Another possibility is that visuomotor feedback responses to a self target jump are expressed at a different latency than responses to a partner target jump.

Overall, we found greater involuntary visuomotor feedback responses for a relevant self target compared to an irrelevant self target. This finding aligns with single-person studies that examined how the relevancy of a mechanical or visual perturbation to the behavioral goal influences rapid feedback responses, prior to volitional control. *Nashed et al., 2012* showed larger long-latency muscular responses (50–100 ms) to mechanical perturbations when reaching to a narrow (circular) relevant target compared to a wide (rectangular) irrelevant target (*Nashed et al., 2012*). This modification of feedback responses based on a relevant/irrelevant task goal has also been shown in response to

visual perturbations (**Knill et al., 2011**; **Cross et al., 2019**). Further, Franklin and colleagues (2008) designed a visual perturbation to be relevant or irrelevant when reaching to the same target (**Franklin and Wolpert, 2008**). They showed greater involuntary visuomotor feedback responses to a relevant visual perturbation compared to an irrelevant visual perturbation. These prior studies suggest that the sensorimotor system can tune involuntary feedback responses based on higher-level task goals. Our novel experimental paradigm has extended these findings to understand how humans integrate their own goal with their partner's goal during jointly controlled actions. Importantly, we found that involuntary visuomotor processes can express not only an individual goal, but also an integrated representation of both the self and partner goals.

Our hypothesis that the sensorimotor system uses a representation of a partner and considers the partner's costs to modify involuntary visuomotor feedback responses can parsimoniously explain all of our experimental findings. There are a few alternative hypotheses that could explain a subset of results. One alternative hypothesis is that participants simply learned the hand-to-center cursor mapping in each experimental condition. That is, instead of using a model of their partner, participants simply adapted to the dynamics of the center cursor. However, this hypothesis would not predict an increased involuntary visuomotor feedback response in the *partner-relevant/self-irrelevant* condition compared to the *partner-irrelevant/self-irrelevant* condition. If participants did not form a model of their partner nor consider their partner's costs, then they would not display an increased feedback response when they had an irrelevant target and their partner's target was relevant. An increased feedback response to help a partner achieve their goal is captured by our hypothesis that the sensorimotor system uses a representation of a partner and considers the partner's costs to modify involuntary visuomotor feedback responses. Another alternative hypothesis would be that the sensorimotor system was responding only to the relevant target displayed on the screen. Again, this hypothesis would only explain a subset of our results. In particular, this relevant target hypothesis cannot explain the observed feedback response differences between the *partner-relevant/self-irrelevant* and *partner-irrelevant/self-relevant* conditions in both Experiments 1 and 2. Finally, we also considered whether time to target (**Česonis and Franklin, 2022**; **Česonis and Franklin, 2020**; Appendix 4), participant forward hand position (Appendix 5), or learning (**Franklin et al., 2012**; Appendix 7–8) influenced feedback responses, but found that none impacted the observed differences between experimental conditions nor changed our interpretation. Our hypothesis that the sensorimotor system uses a representation of a partner and considers the partner's costs to modify involuntary visuomotor feedback responses parsimoniously accounts for the differences observed between all conditions.

Optimal feedback control has been a powerful framework to understand how the nervous system selects movements (**Todorov and Jordan, 2002**; **Scott, 2004**; **Todorov, 2004**; **Calalo et al., 2025**). Past work, including our own (**Lokesh et al., 2023**), has extended optimal feedback control to human-human interaction by having two separate optimal feedback controllers interact (**Takagi et al., 2018**). In these works, the control policy for each of the controllers was selected in isolation. That is, the controllers do not select a control policy using knowledge of the partner's control policy (i.e., partner representation). The dynamic game theory framework further extends the separate feedback controller approach by allowing each controller to select a control policy using a partner representation. This dynamic game theory framework has successfully been used to model human-robot (**Li et al., 2019**) and human-human sensorimotor interactions (**De Vicariis et al., 2024a**; **Chackochan and Sanguineti, 2019**). The aforementioned studies have suggested people form a partner representation in their control policy to produce voluntary movements. Critically, we are the first to our knowledge to measure a proxy of the control policy, assessing how a partner representation influences rapid and involuntary visuomotor feedback responses. Our dynamic game theory model supports the hypothesis that involuntary visuomotor feedback responses reflect a partner representation and joint cost. It would also be interesting to investigate whether other rapid feedback responses, such as the long-latency stretch response, can also express a partner representation.

Both the optimal feedback control and dynamic game theory frameworks view human movement as a process of minimizing a cost function. This cost function is designed such that the controller (i.e., sensorimotor system) achieves some goal state, such as accurately hitting a target, while minimizing an energetic cost. Not correcting for deviations along an irrelevant dimension reduces energetic cost. In our paper, we extended this concept to understand not only how the sensorimotor system considers its own self cost, but also a joint cost that considers both the self cost and partner cost. In

both experiments, we found increased involuntary visuomotor feedback responses in the relevant partner target and irrelevant self target condition compared to both targets being irrelevant. That is, we found that participants' visuomotor feedback responses reflected a consideration of not only the relevancy of their own self-target (i.e., self-cost), but also that of their partner (i.e., partner cost). Furthermore, this result is predicted by our dynamic game theory models that include the partner's costs in the self-cost function. In other words, a dynamic game theory model that selects feedback gains to minimize both the self and partner cost reflects an altruistic control policy. Our experimental and computational results suggest that involuntary visuomotor feedback responses reflect the sensorimotor system's willingness to sacrifice energy to help a partner.

Classic and contemporary theories of action selection, such as Gibson's theory of affordances (*Gibson, 2014*) and the affordance competition hypothesis (*Cisek, 2007*), propose that the sensorimotor system selects movements based on opportunities for action that emerge from the fit between an individual's capabilities and surrounding environment. Our finding that humans sacrifice energetic cost to support a partner's goal extends this perspective by suggesting that the sensorimotor system may also consider 'social affordances', which depend not only on one's own goals but also those of others. An interesting future direction would be to explore how the overlap of the self and partner goals might influence the degree to which humans help one another during collaborative, cooperative, and competitive sensorimotor interactions.

The nervous system can form representations of both self and others. Research studying reaching movements for a single individual has shown that the nervous system forms a representation of one's own limb dynamics (*Kurtzer et al., 2008*; *Miall and King, 2008*; *Lillicrap and Scott, 2013*) and environment (*Franklin et al., 2017*; *Cothros et al., 2006*), which are expressed prior to and following volitional control. Additionally, it has been well-established that the human sensorimotor system can form representations of others (*Sebanz et al., 2003*; *Sebanz and Knoblich, 2021*; *Sebanz and Knoblich, 2009*). *Ramnani and Miall, 2004* showed evidence using fMRI that the human brain even has a dedicated system to predict the actions of others (*Ramnani and Miall, 2004*). Behavioral evidence for these partner representations has been shown across cognitive (*Xiang et al., 2023*; *Bicho et al., 2011*) and perceptual (*Atmaca et al., 2011*; *Bahrami et al., 2010*) decision-making, response time (*Sebanz et al., 2003*), and reaching (*Yoshida et al., 2011*; *Schmitz et al., 2017*; *Chackochan and Sanguineti, 2019*) tasks. *Schmitz et al., 2017* observed that an obstacle in the partner's movement path influenced one's own voluntary reach trajectory (*Schmitz et al., 2017*). Computational and empirical work from *Chackochan and Sanguineti, 2019* suggested that humans use a representation of their partner to select movement trajectories during a reaching task where they are haptically connected to a partner (*Chackochan and Sanguineti, 2019*). Further, others have shown that the sensorimotor system modifies movement selection according to game-theoretic predictions (*Braun et al., 2009*), and that the sensorimotor system modifies movements using an estimate of the joint goal during human-human interactions (*Takagi et al., 2017*; *Takagi et al., 2019*). While neural data, behavioral experiments, and computational modeling have suggested that partner representations influence voluntary reaching movements, to our knowledge none have examined whether a representation of others can be expressed at an involuntary timescale.

The neural basis of upper limb control has been well-studied. For the upper limb, neural recordings in monkeys have shown that activity in the primary motor cortex (M1) reflects visuospatial representations including target goals (*Pruszynski et al., 2008*). High-level visuospatial representations can be rapidly expressed via the muscular long-latency reflex and involuntary visuomotor feedback responses. These lower-level sensorimotor feedback responses are prior to volitional control. The long-latency reflex involves a transcortical pathway with contributions from likely both cortical and subcortical circuitry (*Pruszynski et al., 2014*; *Pruszynski et al., 2011*). It has also been shown that both cortical and subcortical (e.g., superior colliculus) regions are involved when responding to visual perturbations (*Day and Brown, 2001*; *Corneil and Munoz, 2014*; *Pruszynski et al., 2010*; *Kozak and Corneil, 2021*). Collectively, these studies suggest top-down projections from high-level cortical representations to lower-level sensorimotor circuits (*Contemori et al., 2021*), enabling fast and flexible feedback responses.

Just as the sensorimotor system forms representations of its own actions and goals, it has also been shown to represent the actions of others. Observing the actions of others increases neural activity in motor regions such as primary motor and the dorsal premotor cortex (*Kilner and Lemon, 2013*; *Cook*

*et al., 2014*; *Cattaneo and Rizzolatti, 2009*; *Bruni et al., 2018*; *Yoshida et al., 2011*). These so-called 'mirror neurons' may help the sensorimotor system understand the actions of others (*Rizzolatti et al., 2014*; *Umiltà et al., 2001*; but see *Heyes, 2010* for an alternative perspective). Importantly, activation of primary motor and premotor regions has also been shown during the prediction of others' actions, even without directly observing the movement (*Ramnani and Miall, 2004*). Other work has shown that the cerebellum, which uses a self-representation (i.e., internal model) to predict future motor actions (*Kawato, 1999*; *Wolpert et al., 1998*; *Ebner and Pasalar, 2008*), may also form an internal model of others to predict their future actions (*Sokolov et al., 2017*; *Sokolov, 2018*). Therefore, the neural circuitry for the representations of others' actions and for the control of movement seems to be tightly linked (*Miall, 2003*; *Wolpert et al., 2003*). In light of these findings, our work suggests that there is top-down modulation from high-level circuits involved with partner representations to lower-level sensorimotor circuitry. That is, the nervous system appears to leverage high-level partner representations in lower-level sensorimotor circuits to anticipate and respond to a partner's future actions. Future work could use neural recordings while non-human primates perform a cooperative sensorimotor interaction task to further understand how a representation of others might influence the control of movement. From an evolutionary perspective (*Cisek, 2022*; *Cisek, 2019*), it would be interesting to know where along the phylogenetic history a high-level representation of others regulates lower-level sensorimotor circuits involved with rapid and involuntary feedback responses.

Across two experiments and a computational model, we showed that involuntary visuomotor feedback responses reflect a partner representation and consideration of a partner's cost. Our novel results suggest that high-level partner representations influence lower-level involuntary sensorimotor circuitry. Our paradigm offers a powerful new window to probe how human sensorimotor interactions are influenced by cognitive processes, theory of mind, and social dynamics.

## Methods

### Participants

96 participants participated across two experiments. Each experiment was fully counterbalanced, where 24 pairs (48 individuals; 20 male and 28 female) participated in Experiment 1 and 24 pairs (48 individuals; 24 male and 24 female) participated in Experiment 2. All participants reported they were free from musculoskeletal injuries, neurological conditions, or sensory impairments, and were between 18 and 30 years of age. In addition to a base compensation of $5.00, we informed them they would receive a performance-based compensation of up to $5.00. Each participant received the full $10.00 once they completed the experiment irrespective of their performance. All participants provided written informed consent to participate in the experiment and the procedures were approved by the University of Delaware's Institutional Review Board.

### Apparatus

For both experiments, we used two end-point KINARM robots (*Figure 1A*; BKIN Technologies, Kingston, ON). Each participant was seated on an adjustable chair in front of one of the end-point robots. Each participant grasped the handle of a robotic manipulandum and made reaching movements in the horizontal plane. A semi-silvered mirror blocked the vision of the upper limb, and also reflected virtual images (e.g., targets, cursors) from an LCD to the horizontal plane of hand motion. In all experiments, the participant's own (self) cursor was aligned with the position of their hand. Kinematic data were recorded at 1000 Hz and stored offline for data analysis.

### Experimental design

We designed two experiments where participants used knowledge of both their own and partner's target to successfully complete a jointly coordinated reaching task. During both experiments, each participant viewed a self cursor that was aligned with their hand and another cursor that represented their partner's position. They also saw a center cursor at the midpoint between their cursor and their partner's cursor. Finally, they also observed both their own target and their partner's target. The center cursor and both targets were laterally aligned to the center of each participant's screen.

Both targets were 25 cm forward from the start position. Both participants began each trial with their hand placed 13 cm to the right of the center cursor. Each participant observed their partner's

cursor, which was reflected over the center line that intersected the center cursor and the targets. Thus, both participants viewed a mirrored position of their partner. By mirroring both partners, this allowed each participant to view themselves on the right side of the center cursor and their partner on the left side of the center cursor. Further, it allowed for control of the center cursor in a smooth and intuitive manner as if their partner was sitting beside them.

The center cursor was at the midpoint between the participant's hands, except during Experiment 1 perturbation and probe trials when the center cursor was laterally jumped (see further below). The movement of each participant contributed to half the movement of the center cursor. For example, if one participant moved forward 6 cm and their partner did not move, then the center cursor would move 3 cm forward. Likewise, if one participant moved 6 cm to the right and their partner did not move, then the center cursor would move 3 cm to the right.

At the start of a trial, the robot guided each participant's self cursor to their respective start circle. The white start circle (diameter 2 cm) was displayed 13 cm to the right of the initial center cursor location for both participants. After a constant delay of 700 ms the partner cursor, center cursor, self target, and partner target appeared on the screen (*Figure 1A*). Then, after a constant delay of 750 ms, participants heard a tone that indicated they should begin their reach. Instead of self-initiating their movements, we specifically had participants move at the sound of a tone so that the movement onset between participants in a pair was as synchronous as possible (see Appendix 6 for movement onset synchrony analysis).

Both participants in the pair were instructed to move the center cursor into their self target. To complete a trial, each participant had to stabilize the center cursor within their self target for 500 ms. Participants received timing feedback based on the time between the start tone and completing the trial. Participants received the message 'Good', 'Too Slow', or 'Too Fast' if they stabilized within their self target between 1400 ms and 1600 ms, >1600 ms, or <1400 ms, respectively. They therefore had 700–900 ms to first reach the target, since humans generally have response times ~200 ms, and they needed to stabilize within the target for 500 ms (i.e., 1400–200 - 500=700 ms and 1600–200 - 500=900 ms). Movement times of 700–900 ms are consistent with previous human reaching studies (*Franklin and Wolpert, 2008*; *Franklin et al., 2012*; *Nashed et al., 2012*).

Participants were explicitly informed that their timing feedback depended on the center cursor entering and stabilizing within *only* their own target. For example, if the center cursor entered and stabilized within the participant's self target at 1500 ms, but entered and stabilized within the partner target at 1700 ms, then the participant would receive 'Good' feedback and their partner would receive 'Too Slow' feedback. In other words, we ensured participants had a clear understanding that their performance in the task was only based on stabilizing the center cursor in their own self target within the time constraint. Therefore, the instructions and timing constraints did not enforce participants to work together.

The goal of Experiment 1 and Experiment 2 was to study how a representation of a partner's goal influences involuntary visuomotor feedback responses. Therefore, in experimental blocks, we manipulated the width of both the self and partner goal to be either narrow (1.05 cm) or wide (20 cm). The narrow target reflects a task-relevant goal because a participant must correct for lateral deviations of the center cursor to successfully complete their task. The wide target reflects a task-irrelevant goal because a participant does not have to correct for lateral deviations of the center cursor to successfully complete their task. Both targets had a height of 1.25 cm and were aligned horizontally and vertically throughout both experiments. Human pairs performed four blocked conditions in a two-way, repeated measures experimental design: (i) *partner-irrelevant/self-irrelevant*, (ii) *partner-relevant/self-irrelevant*, (iii) *partner-irrelevant/self-relevant*, (iv) *partner-relevant/self-relevant* (*Figure 1B*). The order of the experimental conditions was fully counterbalanced in both Experiment 1 and Experiment 2.

For both experiments, participants first performed a familiarization block of trials, and then 4 experimental blocks that were separated by a washout block. The self and partner targets were 10 cm in width and 1.25 cm in height in both familiarization and washout blocks. Participants performed 50 non-perturbation trials (see below) during the familiarization block. They performed 25 non-perturbation trials during each of the three washout blocks.

Each human pair completed four experimental blocks, where for a block they experienced the (i) *partner-irrelevant/self-irrelevant*, (ii) *partner-relevant/self-irrelevant*, (iii) *partner-irrelevant/*

*self-relevant*, or (iv) *partner-relevant/self-relevant* condition. In the experimental blocks, participants experienced 81 non-perturbation trials, 40 perturbation trials, and 30 probe trials.

### Non-perturbation trials

During non-perturbation trials, the cursor center was always at the midpoint between the human pair. There was neither center cursor (Experiment 1) nor target jumps (Experiment 2).

### Perturbation trials

During perturbation trials within an experimental block, the center cursor (Experiment 1) or both targets (Experiment 2) jumped to either the left (20 trials) or right (20 trials) once the center cursor crossed 25% of the distance to the goals (6.25 cm forward from the start position; *Figure 1C–D*). The cursor or target jump was a 3 cm linear shift in the lateral position over 25ms. The center cursor remained laterally displaced for the duration of the trial. Thus, participants were required to correct for the center cursor or target jump to successfully complete their task when they had a self-relevant target. However, they would not have to correct for the center cursor or target jump to successfully complete their task when they had a self-irrelevant target.

### Probe trials

During probe trials within an experimental block, the center cursor (Experiment 1) or both targets (Experiment 2) jumped to either the left (10 trials) or right (10 trials) once the center cursor crossed 25% of the distance to the goals (6.25 cm forward from the start position; *Figure 1E–F*). We also included 10 null probe trials where the center cursor or both targets did not jump. Here, both participants in the pair were constrained to a force channel that allowed forward hand movement but prevented lateral hand movement. The center cursor or target jump was a 3 cm linear shift in the lateral position over 25 ms. The center cursor or target remained displaced for 200ms and then linearly shifted back to the original lateral position over 25 ms. Critically, as a metric of visuomotor feedback responses, we measured the lateral force participants applied against the channel in response to center cursor or target jumps.

During an experimental block, the non-perturbation trials, perturbation trials, and probe trials were randomly interleaved such that each set of 15 trials contained 8 non-perturbation trials, 2 left perturbation trials, 2 right perturbation trials, 1 left probe trial, 1 right probe trial, and 1 neutral probe trial. Participants performed 10 sets of trials within a block. We also ensured the first trial of an experimental condition was not a probe trial by adding a non-perturbation trial to the start of each experimental condition. In total, participants performed 729 reaches consisting of 50 non-perturbation trials in the familiarization block, 75 non-perturbation trials across the three washout blocks, as well as 480 non-perturbation trials, 240 perturbation trials, and 120 probe trials across the four experimental blocks.

## Dynamic game theory model

We used a dynamic game theory model to predict movement behavior and visuomotor feedback responses of human pairs. Dynamic game theory is a multi-controller extension of the typical optimal feedback control framework that describes a single controller (*Todorov and Jordan, 2002*). This framework has previously been used to model human movement during collaborative tasks (*Li et al., 2019*; *Chackochan and Sanguineti, 2019*; *De Vicariis et al., 2024b*). Here, we modeled our experiments as a linear-quadratic game with two players (controllers; *Başar and Olsder, 1999*). Each controller had direct control of its own hand and attempted to move the center cursor toward its own self target.

### System dynamics

Each hand in the linear-quadratic game was modeled as a point mass. Throughout, the subscript $i$ refers to each controller, where $i = \{1, 2\}$. We describe the model with controller 1 as the self and controller 2 as the partner. The continuous-time dynamics of the point mass representing the hand of controller 1 were as follows:

$$m\ddot{p}_1 = -b\dot{p}_1 + f_1 \tag{8}$$

$$\tau\dot{f}_1 = u_1 - f_1 \tag{9}$$

where $m_1$ is the mass of the hand, $p_1$ is the two-dimensional position vector of the point mass, $b$ is the viscous constant, $f_1$ is the two-dimensional controlled forces, and $u_1$ is the two-dimensional control signal for controller 1. $m$ was set to 1.5 kg, $b$ was set to $0.1 \ \mathrm{N \cdot s \cdot m^{-1}}$ and the time constant of the linear filter ($\tau$) was set to 20 ms. These parameters were identical for controller 1 and controller 2. The parameters were selected so that the model visuomotor feedback response magnitudes closely matched the measured visuomotor feedback response magnitudes.

Controllers 1 and 2 each move their hand and interact to move the center cursor ($cc$). The dynamics of the center cursor are:

$$\dot{p}_{cc} = \frac{\dot{p}_1 + \dot{p}_2}{2}. \tag{10}$$

The state vector $x$ is

$$x = [p_1, \dot{p}_1, f_1, p_2, \dot{p}_2, f_2, p_{cc}, p_{target}]^T \tag{11}$$

where each element in the vector contains an $x$ and $y$ dimension. $T$ is the transpose operator. The system dynamics were transformed into a system of first-order differential equations and discretized. The linear-quadratic state space model is

$$x_{k+1} = Ax_k + B_1 u_{1,k} + B_2 u_{2,k}. \tag{12}$$

Here $x_k$ is the state vector at time $k$ and $A$ is the dynamics matrix. $B_i$ maps the control vector $u_{i,k}$ of player $i$ to muscle force $f_{i,k}$ at time $k$. $A$, $B_1$, and $B_2$ are fully defined in Appendix 3.

## State feedback design

Each controller receives delayed sensory feedback of its own hand position, velocity, and force, as well as the partner's hand position and velocity. Further, each controller receives delayed sensory feedback of the center cursor position and target position. To incorporate sensory delays, we augmented the state vector with previous states (*Lokesh et al., 2023*; *Crevecoeur et al., 2011*):

$$x_k^{aug} = [x_k, \ x_{k-1}, \ ... , x_{k-n_{\delta v}}]^T. \tag{13}$$

Here, $\delta v$ = 110 ms (corresponding to $n_{\delta v}$ = 11 time steps when discretized) to reflect the transmission delay associated with vision and aligned the model and experimental visuomotor response onset times. The sensory states available to controller 1 are

$$y_{1,k} = C_1^{aug} x_k^{aug} + \omega_{1,k} \tag{14}$$

where $y_1$ is the vector of delayed state observations and $\omega_{1,k}$ is a sensory noise vector. $C_1^{aug}$ is an observation matrix designed to selectively observe some of the delayed states. The observation matrices $C_1^{aug}$ and $C_2^{aug}$ and noise vector $\omega_{1,k}$ are fully defined in Appendix 3. We drop the superscript $aug$ to minimize extra notation going forward.

Like previous work, we used a linear Kalman filter to model participants' sensory estimates of the state variables. The posterior state estimate $\hat{x}_{1,k}$ of controller 1 is obtained using an online filter of the form:

$$\hat{x}_{1,k} = \bar{x}_{1,k} + K_{1,k}(y_{1,k} - H_1 \bar{x}_{1,k}) \tag{15}$$

$$\bar{x}_{1,k} = A\hat{x}_{1,k-1} + B_1 u_{1,k} + B_2 u_{2,k}. \tag{16}$$

Here, $\bar{x}_{1,k}$ is the prior prediction of the state. That is, we assume the sensorimotor system obtains a prior prediction of the states using an accurate internal model of the state dynamics, which includes a prediction of the partner's motor command. The prior prediction uses the previous posterior estimate ($\hat{x}_{1,k-1}$), the efference copy ($u_1$), and the prediction of the partner's motor command ($u_2$). The prior prediction of the state is updated using sensory measurements to obtain the posterior estimate $\hat{x}_1$ (*Equation 15*). The sequence of Kalman gains $K_1$ and $K_2$ were updated recursively (Appendix 3).

## Control design

The goal of each controller $i$ is to move the state of the system from an initial state $x_0$ to a target state $x^{target}$ at the final time step $N$ by each minimizing a quadratic cost functional $J_i$:

$$J_1 = \frac{1}{2} \sum_{k=0}^{N-1} \left( x_k^T Q_1 x_k + u_{1,k}^T R_{11} u_{1,k} \right) + \frac{1}{2} x_N^T Q_{1,N} x_N \tag{17}$$

$$J_2 = \frac{1}{2} \sum_{k=0}^{N-1} \left( x_k^T Q_2 x_k + u_{2,k}^T R_{22} u_{2,k} \right) + \frac{1}{2} x_N^T Q_{2,N} x_N. \tag{18}$$

Here, $J_1$ is the individual cost for controller 1 (e.g., self) and $J_2$ is the individual cost for controller 2 (e.g., partner). The quadratic costs penalize deviations from the target state at the final step ($Q_{i,N}$) and controller $i$'s control signals ($R_{ii}$). We then define the joint cost functions as

$$J_1^{\alpha_1} = J_1 + \alpha_1 J_2 \tag{19}$$

$$J_2^{\alpha_2} = J_2 + \alpha_2 J_1, \tag{20}$$

where $\alpha_i \in [0, 1]$ determines the degree to which controller $i$ considers their partner's costs.

The optimal control signal for controllers 1 and 2 is defined as

$$u_{1,k}^* = -F_{1,k} \hat{x}_{1,k} \tag{21}$$

$$u_{2,k}^* = -F_{2,k} \hat{x}_{2,k}, \tag{22}$$

where $\hat{x}_{i,k}$ is the posterior estimate and $F_{i,k}$ is the time-varying feedback gain for controller $i$. The feedback gains $F_{i,k}$, also known as the control policy, are the Nash equilibrium solution to the linear quadratic game described by **Equation 11** and **Equations 16–19** (**Başar and Olsder, 1999**). See Appendix 3 for details.

## Modelling different control policies

We tested four different control policies, each reflecting a hypothesis about how a partner representation and consideration of a partner's cost influences visuomotor feedback responses. In our modelling framework, a partner representation indicates knowledge of the partner's control policy. Further, we can also vary whether a controller considers only their own self cost, or both a self and partner cost (i.e., joint cost). We tested the following four models: (i) No Partner Representation and Self Cost, (ii) Partner Representation and Self Cost, (iii) Partner Representation and Equal Joint Cost, and (iv) Partner Representation and Weighted Joint Cost.

The No Partner Representation and Self Cost model implies that the sensorimotor system does not use a control policy that has a representation of a partner. Mathematically, we set $F_{2,k} = 0$ for all $t$ when calculating the feedback gains for controller 1. That is, if there is no partner representation, then controller 1 does not account for the partner's control policy when selecting its own control policy. Since there is no partner representation, the model can only consider a self cost (i.e., $\alpha_1 = 0$). The Partner Representation and Self Cost model suggests that the sensorimotor system uses a control policy that has a partner representation, but only considers a self cost. That is, controller 1 will produce movements using knowledge of how their partner will move. Further, controller 1 will only produce movements that lead to a minimal self cost, without consideration of the partner's cost. A self cost is obtained by setting $\alpha_1 = 0$ in **Equation 19**.

The Partner Representation and Equal Joint Cost model implies that the sensorimotor system uses a control policy that has a partner representation, and equally weights the self and partner costs. Here, controller 1 will produce movements that use knowledge of how their partner will move. Further, controller 1 will produce movements that lead to an equal minimization of both the self and partner cost. That is, one is willing to potentially spend additional energy so that a partner reaches their goal. An equal joint cost is obtained by setting $\alpha_1 = 1.0$ in **Equation 19**.

The Partner Representation and Weighted Joint Cost model implies that the sensorimotor system uses a control policy that has a partner representation, and partially weights the partner cost. Again, controller 1 will produce movements that use knowledge of how their partner will move. However, controller 1 will produce movements that primarily minimize the self cost and to a lesser extent the partner cost. That is, one will mostly spend energy to reach their own goal, but will still spend some energy to help their partner. A weighted joint cost that weighs the self cost higher than the partner cost is obtained by setting $\alpha_1 = 0.5$ in **Equation 19**.

## Model simulations

We simulated each of the self and partner target structures from the experiment: (i) *partner-irrelevant/self-irrelevant*, (ii) *partner-relevant/self-irrelevant*, (iii) *partner-irrelevant/self-relevant*, and (iv) *partner-relevant/self-relevant*. For relevant targets, we set the x-dimension of the center cursor position in the final state cost matrix $Q_{i,N}$ to 40,000 for $i = \{1, 2\}$. That is, controller $i$ incurs a cost and will correct for lateral deviations of the center cursor away from a relevant target. For irrelevant targets, we set the x-dimension of the center cursor position in the final state cost matrix $Q_{i,N}$ to 100 for $i = \{1, 2\}$. That is, controller $i$ does not incur a cost for lateral deviations of the center cursor if their target is irrelevant. For a full description of $Q_1$ and $Q_2$, see Appendix 3.

We simulated 100 perturbation trials per condition to predict the position trajectories. Perturbation trials were simulated by jumping the center cursor laterally to the left by 3 cm once the center cursor reached 25% of the forward distance to the target.

We also simulated 100 probe trials per condition to predict visuomotor feedback responses. Probe trials were simulated by jumping the center cursor 3 cm laterally for 250 ms, then jumping it back to the original lateral position. To simulate a force channel, we set the x-force element in the $B_1$ and $B_2$ matrices to 0. Thus, the controllers could only move the center cursor in the forward dimension. We were able to calculate the applied force for controller 1 in the lateral dimension using the original $B_1$ matrix. That is, the applied force traces shown in *Figure 4* is $B_1 u_1$ in the x-dimension over time. Aligned with the literature on involuntary visuomotor feedback responses, we calculated the average feedback response from the model during the 180–230 ms epoch.

## Data analysis

We analyzed the results from the non-perturbation, perturbation, and probe trials for the experimental conditions. We recorded both hand positions and the center cursor position during all trials, as well as the force applied by the robot to the hand during the probe trials. All kinematic and kinetic data were filtered with a 5th-order, low-pass Butterworth filter with a 14 hz cutoff frequency.

## Visuomotor feedback responses

The recorded forces applied by both participants in the pair during visual probe trials were time-aligned with the cursor or target jump onset. The delay of the LCD for presentation of visual feedback was determined to be 42 ms. Visuomotor feedback responses in this study are presented relative to the onset of the actual perturbation time. That is, the LCD delay has been taken into account such that visuomotor feedback responses were aligned relative to the time the visual signal was actually presented to participants on their display. In Experiment 1, we define the visuomotor feedback response (N) as the difference between the recorded force during a left cursor jump probe and a right cursor jump probe (*Franklin and Wolpert, 2008*). In Experiment 2, we define the visuomotor feedback response (N) as the difference between the recorded force during a right target jump probe and a left target jump probe.

To investigate the involuntary visuomotor feedback response, we calculated the average force response for each participant during the 180–230 ms time window (*Franklin and Wolpert, 2008*). We also calculated the average force response during the 230–300 ms window, and 300–400 ms window. The 230–300 ms window may contain a mixture of involuntary and voluntary responses, which we term as the semi-involuntary visuomotor feedback response. The 300–400 ms window is the voluntary visuomotor feedback response.

## Final lateral hand deviation

We calculated the final lateral hand deviation as a metric of a participant's voluntary corrective response to perturbation trials. Final lateral hand position was determined as the x-position of each participant at the end of each trial. To calculate final lateral hand deviation, we took the difference between the average final lateral hand position during non-perturbation trials and the average final lateral hand position during perturbation trials (see Appendix 2).

## Statistical analysis

For both experiments, we used a 2 (Self Irrelevant or Self Relevant) x 2 (Partner Irrelevant or Partner Relevant) repeated-measures ANOVA for each dependent variable. We followed up the omnibus tests

with mean comparisons using non-parametric bootstrap hypothesis tests (n=1,000,000; *Cashaback et al., 2019*; *Lokesh et al., 2022*; *Roth et al., 2024*; *Roth et al., 2023*; *Cashaback et al., 2017*; *Sullivan et al., 2025*). Mean comparisons were Holm-Bonferroni corrected to account for multiple comparisons. We computed the common language effect sizes ($\hat{\theta}$) for all mean comparisons. Significance threshold was set at $\alpha$ = 0.05.

## Acknowledgements

National Science Foundation (NSF #2146888) awarded to JGAC, National Sciences and Engineering Research Council (NSERC) of Canada (RGPIN-2018- 05589) awarded to MJC.

## Additional information

### Funding

| Funder | Grant reference number | Author |
|---|---|---|
| U.S. National Science Foundation | 2146888 | Joshua GA Cashaback |
| Natural Sciences and Engineering Research Council of Canada | RGPIN-2018- 05589 | Michael J Carter |

The funders had no role in study design, data collection and interpretation, or the decision to submit the work for publication.

### Author contributions

Seth R Sullivan, Conceptualization, Resources, Data curation, Software, Formal analysis, Validation, Investigation, Visualization, Methodology, Writing – original draft, Project administration, Writing – review and editing; John H Buggeln, Jan A Calalo, Truc T Ngo, Conceptualization, Writing – review and editing; Jennifer A Semrau, Writing – review and editing; Michael J Carter, Funding acquisition, Writing – review and editing; Joshua GA Cashaback, Conceptualization, Supervision, Funding acquisition, Methodology, Project administration, Writing – review and editing

### Author ORCIDs

Seth R Sullivan https://orcid.org/0000-0002-0840-8292
John H Buggeln https://orcid.org/0000-0002-4276-8806
Michael J Carter https://orcid.org/0000-0002-0675-4271
Joshua GA Cashaback https://orcid.org/0000-0002-8642-6648

### Ethics

Human subjects: This study was approved by the University of Delaware's Institutional Review Board. All participants provided written informed consent to participate.

Reviewer #1 (Public review): https://doi.org/10.7554/eLife.109734.3.sa1
Reviewer #2 (Public review): https://doi.org/10.7554/eLife.109734.3.sa2
Author response https://doi.org/10.7554/eLife.109734.3.sa3

## Additional files

### Supplementary files

MDAR checklist

### Data availability

Data for both experiments can be found on figshare at https://doi.org/10.6084/m9.figshare.30132088. The code for the computational model can be found on GitHub at https://github.com/CashabackLab/human-human-linear-quadratic-game, copy archived at *Sullivan, 2025*. Analysis code can be found

on GitHub at https://github.com/CashabackLab/Involuntary-Visuomotor-HHI-Analysis-Code, copy archived at *Sullivan, 2026*.

The following dataset was generated:

| Author(s) | Year | Dataset title | Dataset URL | Database and Identifier |
|---|---|---|---|---|
| Sullivan SR, Buggeln JH, Calalo JA, Ngo TT, Semrau JA, Carter MJ, Cashaback JGA | 2025 | Data - Involuntary visuomotor feedback responses reflect a representation of partner actions | https://doi.org/10.6084/m9.figshare.30132088 | figshare, 10.6084/m9.figshare.30132088 |

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

## Appendix 1

### Model and Experiment 2 trajectories

To make a priori predictions of voluntary motor behavior, we simulated center cursor jump perturbation trials. Note that center cursor jumps and target jumps result in identical behavior in model simulations. The hand trajectory predictions for each of the four models on leftward cursor jump perturbation trials are shown for each condition in *Appendix 1—figure 1A–P*.

In the *partner-relevant/self-irrelevant* condition, we predicted that the self-controller in the No Partner Representation and Self Cost (*Appendix 1—figure 1B*) and Partner Representation and Self Cost model (*Appendix 1—figure 1F*) would not make a lateral correction for a cursor jump. Conversely, in the same *partner-relevant/self-irrelevant* condition, we predicted that the self-controller in the Partner Representation and Equal Joint Cost (*Appendix 1—figure 1J*) and Partner Representation and Weighted Joint Cost model (*Appendix 1—figure 1N*) would make a lateral correction for a cursor jump.

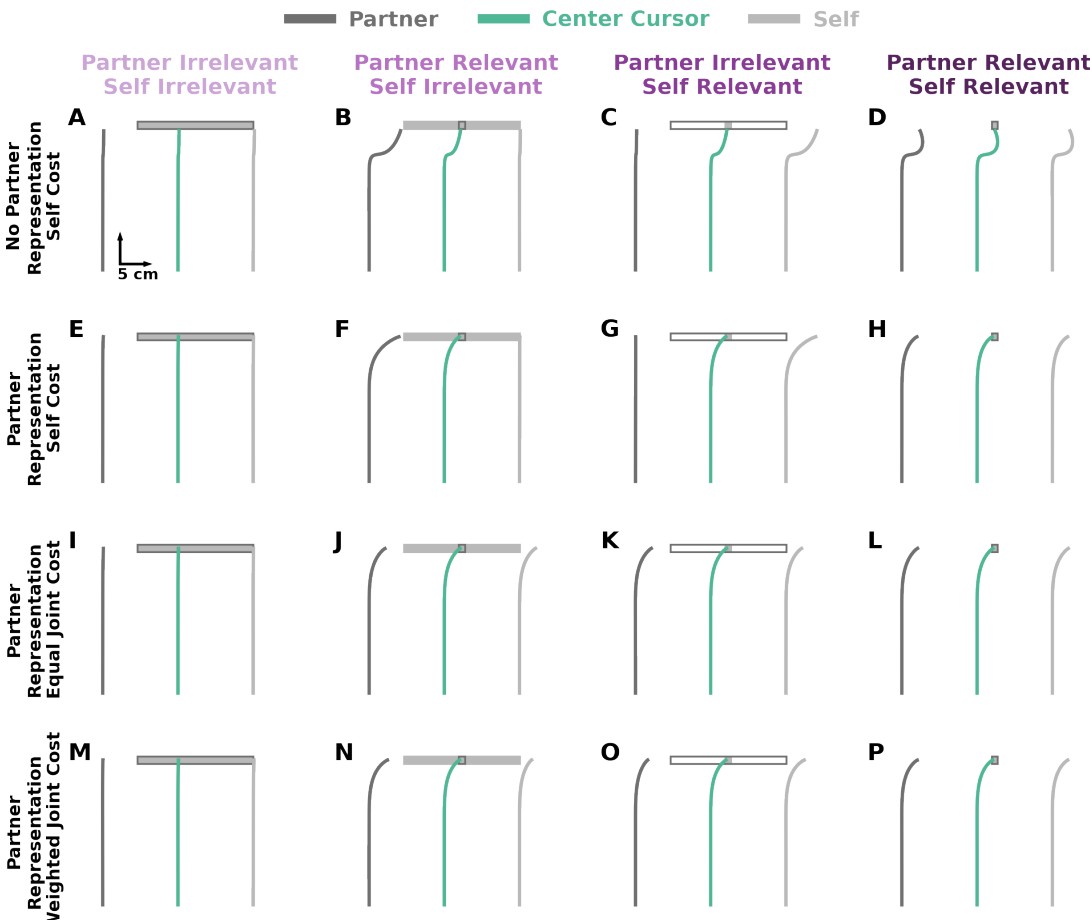

**Appendix 1—figure 1.** Model hand and center cursor trajectories. Predicted hand and center cursor positions during left cursor jumps for each condition and model: (**A-D**) No Partner Representation and Self Cost, (**E–H**) Partner Representation and Self Cost, (**I–L**) Partner Representation and Equal Joint Cost, and (**M-P**) Partner Representation and Weighted Joint Cost. Collectively, the self cursor in models with only a self cost do not laterally deviate to correct for the cursor jump in the *partner-relevant/self-irrelevant* condition. In contrast, the self cursor in models that consider a self and partner cost laterally deviate to correct for the cursor jump in the *partner-relevant/self-irrelevant* condition.

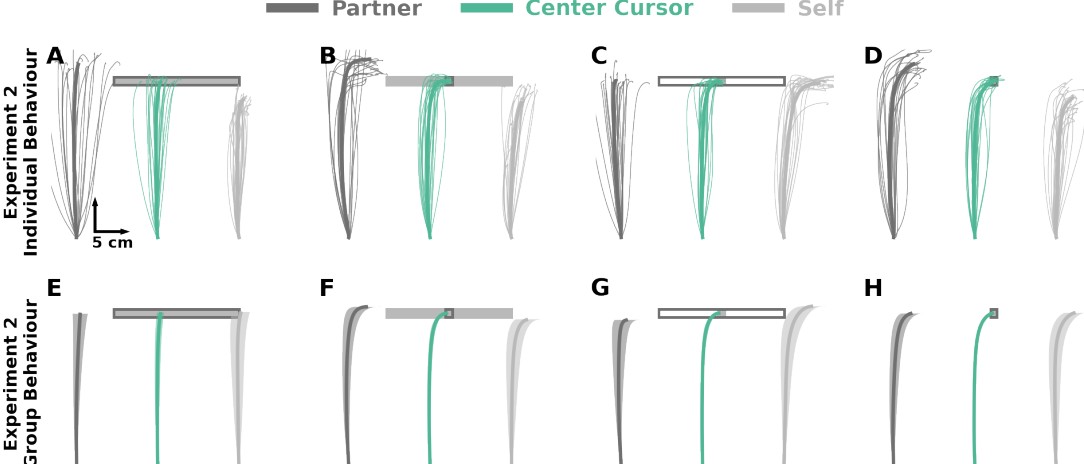

**Appendix 1—figure 2.** Experiment 2: trajectories. (**A–D**) Individual hand and center cursor positions of an exemplar pair for each condition in Experiment 2. Thin traces represent each trial. Thick traces represent the average across trials for the human pair. (**E–F**) Group average hand and center cursor positions in Experiment 2. Traces represent the mean, and shaded regions reflect ±1 standard error of the mean. The group average behavior in Experiment 2 closely aligns with the Partner Representation and Weighted Joint Cost model (*Appendix 1— figure 1*), suggesting that voluntary behavior reflects a partner representation and consideration of a partner's cost.

## Appendix 2

### Final lateral hand deviation

We calculated final lateral hand deviation for each participant as the average absolute difference between the endpoint hand position on regular trials and perturbation trials. *Appendix 2—figure 1* shows the predicted final lateral hand deviation from each of the four models. *Appendix 2—figure 2* shows the final lateral hand deviation in Experiment 1 (*Appendix 2—figure 2A*) and Experiment 2 (*Appendix 2—figure 2B*). We found a significant interaction between partner target and self target for both Experiment 1 ($F[1,47]=60.99$, $p < 0.001$) and Experiment 2 ($F[1,47]=42.66$, $p < 0.001$). Follow-up mean comparisons showed the same significant differences for Experiment 1 and Experiment 2. The empirical results of final lateral hand position most closely match the Partner Representation and Weighted Joint Cost model, suggesting that participants form a partner representation and consider a weighted joint cost.

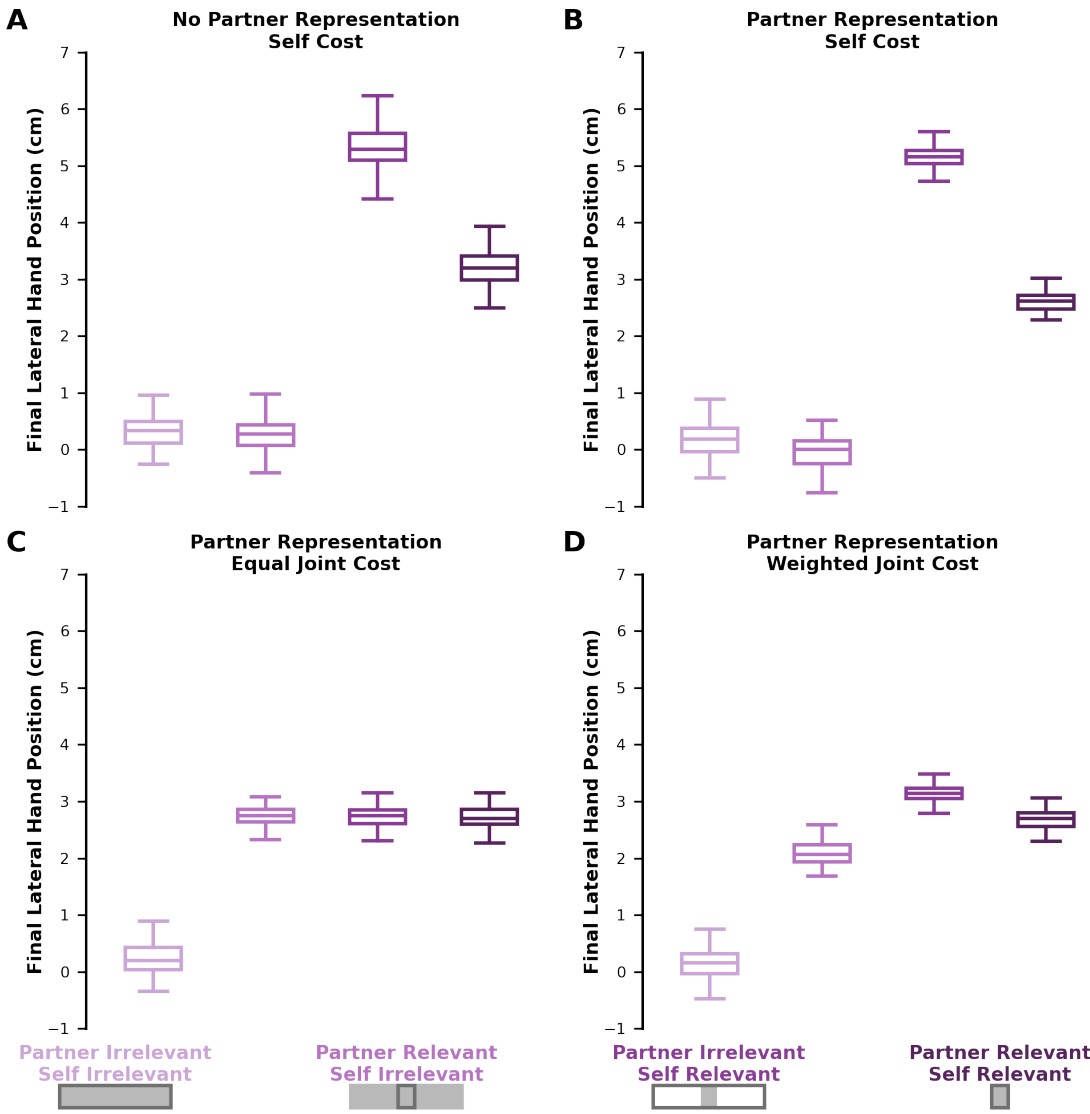

**Appendix 2—figure 1.** Model final hand lateral deviation. Final lateral hand deviation for (**A**) No Partner Representation and Self Cost, (**B**) Partner Representation and Self Cost, (**C**) Partner Representation and Equal Joint Cost, (**D**) Partner Representation and Weighted Joint Cost. The final lateral hand deviation was calculated as the mean of the absolute value of the difference between the final hand position on non-perturbation trials and the final hand position on perturbation trials.

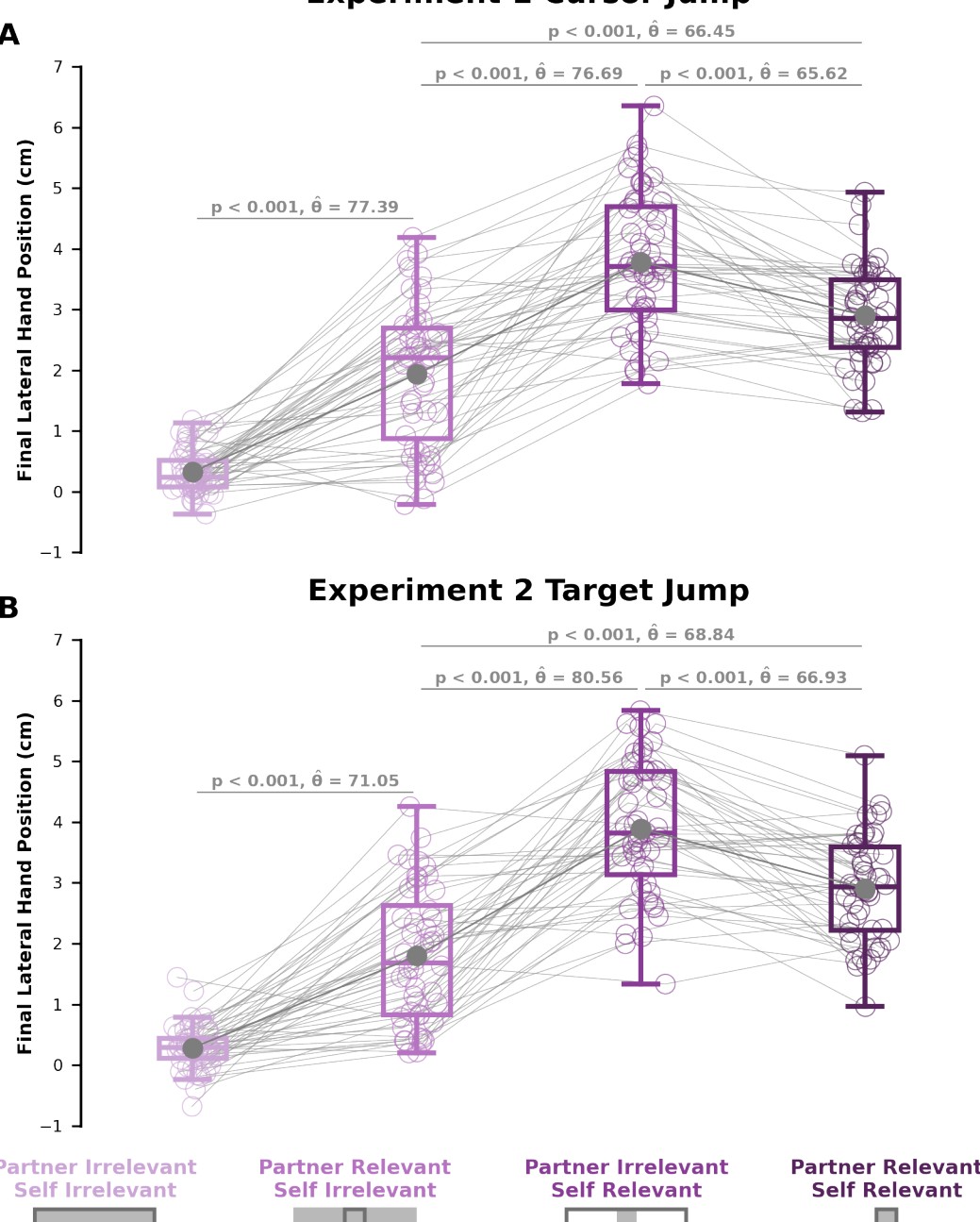

**Appendix 2—figure 2.** Experimental final hand lateral deviation. Final lateral hand deviation for (**A**) Experiment 1 and (**B**) Experiment 2. The final lateral hand deviation was calculated as the mean of the absolute value of the difference between the final hand position on non-perturbation trials and the final hand position on perturbation trials. We saw significant differences between each of our statistical comparisons for both Experiment 1 and Experiment 2. The results for both experiments closely match the Partner Representation and Weighted Joint Cost model (*Appendix 2—figure 1D*), showing that voluntary behavior considers a partner representation and a weighted joint cost.

# Appendix 3

## Modelling

### Linear quadratic game

As a reminder, we modeled our task as a linear quadratic game of the form

$$x_{k+1} = Ax_k + B_1 u_{1,k} + B_2 u_{2,k} + \Sigma_u. \tag{A1}$$

Here, the subscripts 1 and 2 respectively refer to controller 1 and 2, representing a pair of participants in our task. $x_k$ is the state (e.g., position) of the system at time step $k$, $A$ represents the task dynamics, $u_1$ and $u_2$ are the control signals, and $B_1$ and $B_2$ convert the control signals to a force that produces movement. $\Sigma_u$ is a covariance matrix that inputs additive noise to the system, representing noisy control signals (defined further below). Throughout, we describe the model with controller 1 as the self and controller 2 as the partner.

Controller 1 and 2 select their own control signal $u_1^*$ or $u_2^*$, which considers their respective costs. We can define an individual cost function $J_1$ and $J_2$ as:

$$J_1 = \frac{1}{2} \sum_{k=0}^{N-1} \left( x_k^\mathsf{T} Q_{1,k} x_k + u_{1,k}^\mathsf{T} R_{11} u_{1,k} \right) + \frac{1}{2} x_N^\mathsf{T} Q_{1,N} x_N \tag{A2}$$

$$J_2 = \frac{1}{2} \sum_{k=0}^{N-1} \left( x_k^\mathsf{T} Q_{2,k} x_k + u_{2,k}^\mathsf{T} R_{22} u_{2,k} \right) + \frac{1}{2} x_N^\mathsf{T} Q_{2,N} x_N. \tag{A3}$$

Here, $J_1$ is the individual cost for controller 1 (e.g., self) and $J_2$ is the individual cost for controller 2 (e.g., partner). $N$ is the final step, which represents the end of a trial. The term $Q$ penalizes deviations of the center cursor relative to each target. We modelled (i) a task-relevant target using a higher value of Q, and (ii) a task-irrelevant target using a lower value of Q. The term $R_{ii}$ penalizes controller $i$'s control signals, which can be thought of as an energetic cost. See further below and **Appendix 3— table 1** for the fully defined matrices $Q$ and $R$. We define a joint cost function as:

$$J_1^{\alpha_1} = J_1 + \alpha_1 J_2 \tag{A4}$$

$$J_2^{\alpha_2} = J_2 + \alpha_2 J_1, \tag{A5}$$

where $\alpha_i \in [0, 1]$ determines the degree to which controller $i$ considers their partner's cost function. Specifically, the term $\alpha_i$ was applied to both the partner's state cost and energetic cost:

$$Q_{1,k}^{\alpha_1} = max(Q_{1,k}, \alpha_1 Q_{2,k}) \tag{A6}$$

$$Q_{2,k}^{\alpha_2} = max(Q_{2,k}, \alpha_2 Q_{1,k}) \tag{A7}$$

$$R_{12} = \alpha_1 R_{11} \tag{A8}$$

$$R_{21} = \alpha_2 R_{22}. \tag{A9}$$

$Q_1^{\alpha_1}$ reflects how much controller 1 considers controller 2's state cost (i.e., controller 1's partner stabilizing the center cursor in the partner target). $Q_2^{\alpha_2}$ reflects how much controller 2 considers controller 1's state cost (i.e., controller 2's partner stabilizing the center cursor in the partner target). Specifically, we reasoned that each controller would only consider their partner's state cost if their partner's state cost was higher (i.e., higher value of an element in $Q$). We implemented this logic with the max function in **Equation A6 and A7**. That is, for controller 1, if the self target was relevant then $Q_{1,k}^{\alpha_1} = Q_{1,k}$ and if the self target was irrelevant and partner target was relevant then $Q_{1,k}^{\alpha_1} = \alpha Q_{2,k}$. Additionally, $R_{12}$ reflects how much controller 1 considers controller 2's energetic cost of movement by multiplying alpha by its own energetic cost. Likewise, $R_{21}$ reflects how much controller 2 considers controller 1's energetic cost of movement. We can rewrite the joint cost functions in **Equation A4 and A5** in the quadratic form as:

$$J_1^{\alpha_1} = \frac{1}{2} \sum_{k=0}^{N-1} \left( x_k^\mathsf{T} Q_{1,k}^{\alpha_1} x_k + u_{1,k}^\mathsf{T} R_{11} u_{1,k} + u_{2,k}^\mathsf{T} R_{12} u_{2,k} \right) + \frac{1}{2} x_N^\mathsf{T} Q_{1,N}^{\alpha_1} x_N \tag{A10}$$

$$J_2^{\alpha_2} = \frac{1}{2} \sum_{k=0}^{N-1} \left( x_k^\mathsf{T} Q_{2,k}^{\alpha_2} x_k + u_{2,k}^\mathsf{T} R_{22} u_{2,k} + u_{1,k}^\mathsf{T} R_{21} u_{1,k} \right) + \frac{1}{2} x_N^\mathsf{T} Q_{2,N}^{\alpha_2} x_N. \tag{A11}$$

## Linear quadratic game - matrices

$$\mathrm{x} = \begin{bmatrix} p_{1,x} & v_{1,x} & f_{1,x} & p_{1,y} & v_{1,y} & f_{1,y} & p_{2,x} & v_{2,x} & f_{2,x} & p_{2,y} & v_{2,y} & f_{2,y} & p_{cc,x} & p_{cc,y} & p_{ct,x} & p_{ct,y} \end{bmatrix}^\mathsf{T}$$

$$x_0 = \begin{bmatrix} 0.13 \, 0 \, 0 \, 0 \, 0 \, 0 \, -0.13 \, 0 \, 0 \, 0 \, 0 \, 0 \, 0 \, 0 \, 0 \, 0.25 \end{bmatrix}^\mathsf{T}$$

$x_0$ reflects the initial state at the start of each simulation, matching the experimental design.

$$A = \begin{bmatrix}
0 & 1 & 0 & 0 & 0 & 0 & 0 & 0 & 0 & 0 & 0 & 0 & 0 & 0 & 0 & 0 \\
0 & \frac{-b}{m} & \frac{1}{m} & 0 & 0 & 0 & 0 & 0 & 0 & 0 & 0 & 0 & 0 & 0 & 0 & 0 \\
0 & 0 & \frac{-1}{\tau} & 0 & 0 & 0 & 0 & 0 & 0 & 0 & 0 & 0 & 0 & 0 & 0 & 0 \\
0 & 0 & 0 & 0 & 1 & 0 & 0 & 0 & 0 & 0 & 0 & 0 & 0 & 0 & 0 & 0 \\
0 & 0 & 0 & 0 & \frac{-b}{m} & \frac{1}{m} & 0 & 0 & 0 & 0 & 0 & 0 & 0 & 0 & 0 & 0 \\
0 & 0 & 0 & 0 & 0 & \frac{-1}{\tau} & 0 & 0 & 0 & 0 & 0 & 0 & 0 & 0 & 0 & 0 \\
0 & 0 & 0 & 0 & 0 & 0 & 0 & 1 & 0 & 0 & 0 & 0 & 0 & 0 & 0 & 0 \\
0 & 0 & 0 & 0 & 0 & 0 & 0 & \frac{-b}{m} & \frac{1}{m} & 0 & 0 & 0 & 0 & 0 & 0 & 0 \\
0 & 0 & 0 & 0 & 0 & 0 & 0 & 0 & \frac{-1}{\tau} & 0 & 0 & 0 & 0 & 0 & 0 & 0 \\
0 & 0 & 0 & 0 & 0 & 0 & 0 & 0 & 0 & 0 & 1 & 0 & 0 & 0 & 0 & 0 \\
0 & 0 & 0 & 0 & 0 & 0 & 0 & 0 & 0 & 0 & \frac{-b}{m} & \frac{1}{m} & 0 & 0 & 0 & 0 \\
0 & 0 & 0 & 0 & 0 & 0 & 0 & 0 & 0 & 0 & 0 & \frac{-1}{\tau} & 0 & 0 & 0 & 0 \\
0 & 0.5 & 0 & 0 & 0 & 0 & 0 & 0.5 & 0 & 0 & 0 & 0 & 0 & 0 & 0 & 0 \\
0 & 0 & 0 & 0 & 0.5 & 0 & 0 & 0 & 0 & 0 & 0.5 & 0 & 0 & 0 & 0 & 0 \\
0 & 0 & 0 & 0 & 0 & 0 & 0 & 0 & 0 & 0 & 0 & 0 & 0 & 0 & 0 & 0 \\
0 & 0 & 0 & 0 & 0 & 0 & 0 & 0 & 0 & 0 & 0 & 0 & 0 & 0 & 0 & 0
\end{bmatrix}$$

$$B_1 = \begin{bmatrix} 0 & 0 & \frac{1}{\tau} & 0 & 0 & 0 & 0 & 0 & 0 & 0 & 0 & 0 & 0 & 0 & 0 & 0 \\ 0 & 0 & 0 & 0 & 0 & \frac{1}{\tau} & 0 & 0 & 0 & 0 & 0 & 0 & 0 & 0 & 0 & 0 \end{bmatrix}^\mathsf{T}$$

$$B_2 = \begin{bmatrix} 0 & 0 & 0 & 0 & 0 & 0 & 0 & 0 & \frac{1}{\tau} & 0 & 0 & 0 & 0 & 0 & 0 & 0 \\ 0 & 0 & 0 & 0 & 0 & 0 & 0 & 0 & 0 & 0 & 0 & \frac{1}{\tau} & 0 & 0 & 0 & 0 \end{bmatrix}^\mathsf{T}$$

$$\Sigma_u = \begin{bmatrix}
0 & 0 & 0 & 0 & 0 & 0 & 0 & 0 & 0 & 0 & 0 & 0 & 0 & 0 & 0 & 0 & 0 \\
0 & 0 & 0 & 0 & 0 & 0 & 0 & 0 & 0 & 0 & 0 & 0 & 0 & 0 & 0 & 0 & 0 \\
0 & 0 & \sigma_u^2 & 0 & 0 & 0 & 0 & 0 & 0 & 0 & 0 & 0 & 0 & 0 & 0 & 0 & 0 \\
0 & 0 & 0 & 0 & 0 & 0 & 0 & 0 & 0 & 0 & 0 & 0 & 0 & 0 & 0 & 0 & 0 \\
0 & 0 & 0 & 0 & 0 & 0 & 0 & 0 & 0 & 0 & 0 & 0 & 0 & 0 & 0 & 0 & 0 \\
0 & 0 & 0 & 0 & 0 & \sigma_u^2 & 0 & 0 & 0 & 0 & 0 & 0 & 0 & 0 & 0 & 0 & 0 \\
0 & 0 & 0 & 0 & 0 & 0 & 0 & 0 & 0 & 0 & 0 & 0 & 0 & 0 & 0 & 0 & 0 \\
0 & 0 & 0 & 0 & 0 & 0 & 0 & 0 & 0 & 0 & 0 & 0 & 0 & 0 & 0 & 0 & 0 \\
0 & 0 & 0 & 0 & 0 & 0 & 0 & 0 & \sigma_u^2 & 0 & 0 & 0 & 0 & 0 & 0 & 0 & 0 \\
0 & 0 & 0 & 0 & 0 & 0 & 0 & 0 & 0 & 0 & 0 & 0 & 0 & 0 & 0 & 0 & 0 \\
0 & 0 & 0 & 0 & 0 & 0 & 0 & 0 & 0 & 0 & 0 & 0 & 0 & 0 & 0 & 0 & 0 \\
0 & 0 & 0 & 0 & 0 & 0 & 0 & 0 & 0 & 0 & 0 & \sigma_u^2 & 0 & 0 & 0 & 0 & 0 \\
0 & 0 & 0 & 0 & 0 & 0 & 0 & 0 & 0 & 0 & 0 & 0 & 0 & 0 & 0 & 0 & 0 \\
0 & 0 & 0 & 0 & 0 & 0 & 0 & 0 & 0 & 0 & 0 & 0 & 0 & 0 & 0 & 0 & 0 \\
0 & 0 & 0 & 0 & 0 & 0 & 0 & 0 & 0 & 0 & 0 & 0 & 0 & 0 & 0 & 0 & 0 \\
0 & 0 & 0 & 0 & 0 & 0 & 0 & 0 & 0 & 0 & 0 & 0 & 0 & 0 & 0 & 0 & 0 \\
0 & 0 & 0 & 0 & 0 & 0 & 0 & 0 & 0 & 0 & 0 & 0 & 0 & 0 & 0 & 0 & 0
\end{bmatrix}$$

$$Q_{1,k} = 0_{17 \times 17}, \; k = [0, ..., N-1]$$

$$Q_{1,N} = \begin{bmatrix}
0 & 0 & 0 & 0 & 0 & 0 & 0 & 0 & 0 & 0 & 0 & 0 & 0 & 0 & 0 & 0 \\
0 & w_v & 0 & 0 & 0 & 0 & 0 & 0 & 0 & 0 & 0 & 0 & 0 & 0 & 0 & 0 \\
0 & 0 & w_f & 0 & 0 & 0 & 0 & 0 & 0 & 0 & 0 & 0 & 0 & 0 & 0 & 0 \\
0 & 0 & 0 & w_y & 0 & 0 & 0 & 0 & 0 & -w_y & 0 & 0 & 0 & 0 & 0 & 0 \\
0 & 0 & 0 & 0 & w_v & 0 & 0 & 0 & 0 & 0 & 0 & 0 & 0 & 0 & 0 & 0 \\
0 & 0 & 0 & 0 & 0 & w_f & 0 & 0 & 0 & 0 & 0 & 0 & 0 & 0 & 0 & 0 \\
0 & 0 & 0 & 0 & 0 & 0 & 0 & 0 & 0 & 0 & 0 & 0 & 0 & 0 & 0 & 0 \\
0 & 0 & 0 & 0 & 0 & 0 & 0 & 0 & 0 & 0 & 0 & 0 & 0 & 0 & 0 & 0 \\
0 & 0 & 0 & 0 & 0 & 0 & 0 & 0 & 0 & 0 & 0 & 0 & 0 & 0 & 0 & 0 \\
0 & 0 & 0 & -w_y & 0 & 0 & 0 & 0 & 0 & w_y & 0 & 0 & 0 & 0 & 0 & 0 \\
0 & 0 & 0 & 0 & 0 & 0 & 0 & 0 & 0 & 0 & 0 & 0 & 0 & 0 & 0 & 0 \\
0 & 0 & 0 & 0 & 0 & 0 & 0 & 0 & 0 & 0 & 0 & 0 & 0 & 0 & 0 & 0 \\
0 & 0 & 0 & 0 & 0 & 0 & 0 & 0 & 0 & 0 & 0 & 0 & w_{1,cc_x} & 0 & -w_{1,cc_x} & 0 \\
0 & 0 & 0 & 0 & 0 & 0 & 0 & 0 & 0 & 0 & 0 & 0 & 0 & w_{cc_y} & 0 & -w_{cc_y} \\
0 & 0 & 0 & 0 & 0 & 0 & 0 & 0 & 0 & 0 & 0 & 0 & -w_{1,cc_x} & 0 & w_{1,cc_x} & 0 \\
0 & 0 & 0 & 0 & 0 & 0 & 0 & 0 & 0 & 0 & 0 & 0 & 0 & -w_{cc_y} & 0 & w_{cc_y}
\end{bmatrix}$$

$$Q_{2,k} = 0_{17 \times 17}, \; k = [0, ..., N-1]$$

$$Q_{2,N} = \begin{bmatrix}
0 & 0 & 0 & 0 & 0 & 0 & 0 & 0 & 0 & 0 & 0 & 0 & 0 & 0 & 0 & 0 \\
0 & 0 & 0 & 0 & 0 & 0 & 0 & 0 & 0 & 0 & 0 & 0 & 0 & 0 & 0 & 0 \\
0 & 0 & 0 & 0 & 0 & 0 & 0 & 0 & 0 & 0 & 0 & 0 & 0 & 0 & 0 & 0 \\
0 & 0 & 0 & w_y & 0 & 0 & 0 & 0 & 0 & -w_y & 0 & 0 & 0 & 0 & 0 & 0 \\
0 & 0 & 0 & 0 & 0 & 0 & 0 & 0 & 0 & 0 & 0 & 0 & 0 & 0 & 0 & 0 \\
0 & 0 & 0 & 0 & 0 & 0 & 0 & 0 & 0 & 0 & 0 & 0 & 0 & 0 & 0 & 0 \\
0 & 0 & 0 & 0 & 0 & 0 & 0 & 0 & 0 & 0 & 0 & 0 & 0 & 0 & 0 & 0 \\
0 & 0 & 0 & 0 & 0 & 0 & 0 & w_v & 0 & 0 & 0 & 0 & 0 & 0 & 0 & 0 \\
0 & 0 & 0 & 0 & 0 & 0 & 0 & 0 & w_f & 0 & 0 & 0 & 0 & 0 & 0 & 0 \\
0 & 0 & 0 & -w_y & 0 & 0 & 0 & 0 & 0 & w_y & 0 & 0 & 0 & 0 & 0 & 0 \\
0 & 0 & 0 & 0 & 0 & 0 & 0 & 0 & 0 & 0 & w_v & 0 & 0 & 0 & 0 & 0 \\
0 & 0 & 0 & 0 & 0 & 0 & 0 & 0 & 0 & 0 & 0 & w_f & 0 & 0 & 0 & 0 \\
0 & 0 & 0 & 0 & 0 & 0 & 0 & 0 & 0 & 0 & 0 & 0 & w_{2,cc_x} & 0 & -w_{2,cc_x} & 0 \\
0 & 0 & 0 & 0 & 0 & 0 & 0 & 0 & 0 & 0 & 0 & 0 & 0 & w_{cc_y} & 0 & -w_{cc_y} \\
0 & 0 & 0 & 0 & 0 & 0 & 0 & 0 & 0 & 0 & 0 & 0 & -w_{2,cc_x} & 0 & w_{2,cc_x} & 0 \\
0 & 0 & 0 & 0 & 0 & 0 & 0 & 0 & 0 & 0 & 0 & 0 & 0 & -w_{cc_y} & 0 & w_{cc_y}
\end{bmatrix}$$

Writing out $xQ_ix$ from the cost function $J_i$ gives the state cost:

$$StateCost_1 = w_v * (v_{1,x}^2 + v_{1,y}^2) + w_f * (f_{1,x}^2 + f_{1,y}^2)$$

$$+ w_y * (p_{1,y} - p_{2,y})^2 + w_{1,cc_x} * (p_{cc,x} - p_{ct,x})^2 + w_{cc_y} * (p_{cc,y} - p_{ct,y})^2$$

$$StateCost_2 = w_v * (v_{2,x}^2 + v_{2,y}^2) + w_f * (f_{2,x}^2 + f_{2,y}^2)$$

$$+ w_y * (p_{1,y} - p_{2,y})^2 + w_{2,cc_x} * (p_{cc,x} - p_{ct,x})^2 + w_{cc_y} * (p_{cc,y} - p_{ct,y})^2$$

The control costs for controller 1 and controller 2 are:

$$R_{11} = \begin{bmatrix} r & 0 \\ 0 & r \end{bmatrix} \qquad R_{22} = \begin{bmatrix} r & 0 \\ 0 & r \end{bmatrix}.$$

Note that all the parameter weightings for the state and control costs are held constant across all experimental conditions except for $w_{1,cc_x}$ and $w_{2,cc_x}$ which reflected the width of the target. For values of all parameters, see **Appendix 3—table 1**.

## Control policy

The optimal control signal for controller 1 ($u_{1,k}^*$) and controller 2 ($u_{1,k}^*$) is determined by the time-varying feedback gains $F_1$ and $F_2$ that minimize the joint cost function $J_1^{\alpha_1}$ and $J_2^{\alpha_2}$, respectively:

$$u_{1,k}^* = -F_{1,k}\hat{x}_{1,k} \tag{A12}$$

$$u_{2,k}^* = -F_{2,k}\hat{x}_{2,k} \tag{A13}$$

Here, $\hat{x}_{i,k}$ for $i = \{1, 2\}$ is controller $i$'s posterior estimate of the state (see Methods in the main manuscript). The feedback gains $F_1$ and $F_2$ constitute a Nash equilibrium solution to the linear quadratic game defined above (from Basar and Olsder Chapter 6, Corollary 6.1; **Başar and Olsder, 1999**):

$$F_{1,k} = (R_{11} + B_1^{\mathsf{T}}P_{1,k}B_1)^{-1}(B_1^{\mathsf{T}}P_{1,k}A - B_1^{\mathsf{T}}P_{1,k}B_2F_{2,k}) \tag{A14}$$

$$F_{2,k} = (R_{22} + B_2^\mathsf{T} P_{2,k} B_2)^{-1}(B_2^\mathsf{T} P_{2,k} A - B_2^\mathsf{T} P_{2,k} B_1 F_{1,k}). \tag{A15}$$

Importantly, note that the solution $F_{1,k}$ contains the term $B_1^\mathsf{T} P_{1,k} B_2 F_{2,k}$ demonstrating knowledge of the partner's feedback gains $F_{2,k}$. In our modelling framework, this knowledge of the partner's control policy reflects a partner representation.

Here, $P_{i,k}$ is the solution to the set of coupled Riccati equations derived via dynamic programming for a linear quadratic game.

$$P_{1,k} = S_k^\mathsf{T} P_{1,k+1} S_k + F_{1,k}^\mathsf{T} R_{11} F_{1,k} + F_{2,k}^\mathsf{T} R_{12} F_{2,k} + Q_{1,k}^\alpha \tag{A16}$$

$$P_{2,k} = S_k^\mathsf{T} P_{2,k+1} S_k + F_{2,k}^\mathsf{T} R_{22} F_{2,k} + F_{1,k}^\mathsf{T} R_{21} F_{1,k} + Q_{2,k}^\alpha \tag{A17}$$

where $S_k = A - B_1 F_{1,k} - B_2 F_{2,k}$

Note that $P_{i,N} = Q_{i,N}$ where $N$ is the final timestep, and $P_{i,k}$ is recursively solved backwards in time.

## State estimation and sensory delays

Each controller receives delayed sensory feedback of its own hand position, velocity, and force, as well as the partner's hand position and velocity. Further, each controller receives delayed sensory feedback of the center cursor position and target position. To incorporate sensory delays, we augmented the state vector with previous states (*Lokesh et al., 2023*; *Crevecoeur et al., 2011*).

$$x_k^{aug} = [x_k,\ x_{k-1},\ \dots, x_{k-n_{\delta v}}]^\mathsf{T} \tag{A18}$$

Here, $\delta v$ = 110 ms to reflect the transmission delay associated with vision and aligned the model and experimental visuomotor response onset times. This value was converted into time steps in our program ($n_{\delta v}$ = 11). To accommodate the augmented state vector, we also augmented $A$, $B_i$, and $Q_i$:

$$A^{aug} = \begin{bmatrix} A & 0_{n_x \times n_{xd}} \\ I_{n_{xd}} & 0_{n_{xd} \times n_x} \end{bmatrix}$$

$$B^{aug} = \begin{bmatrix} B \\ 0_{n_{xd} \times n_u} \end{bmatrix}$$

$$Q^{aug} = \begin{bmatrix} Q & 0_{n_x \times n_{xd}} \\ 0_{n_{xd} \times n_x} & 0_{n_{xd} \times n_{xd}} \end{bmatrix}$$

Here, $n_x$ is the total number of states (16), $n_u$ is the number of control states (2), and $n_{xd}$ is the total number of time delayed states ($n_x \cdot n_{\delta v}$ = 176). $I_p$ is the square identity matrix with $n_{xd}$ rows and $n_{xd}$ columns. 0 is a matrix with all zeros of the specified dimensions. $B^{aug}$ is designed so the controllers only act on the current (non-delayed) state. $Q^{aug}$ is designed so the controllers only incur a cost on the current (non-delayed) state. The sensory states available to controller $i$ is

$$y_1 = C_1^{aug} x_k^{aug} + \omega_{1,k} \tag{A19}$$

$$y_2 = C_2^{aug} x_k^{aug} + \omega_{2,k} \tag{A20}$$

where $y_1, y_2$ are the vectors of delayed state observations for controller 1 and 2, respectively. $\omega_{1,k}$ and $\omega_{2,k}$ are each noise vectors whose elements are drawn from a Gaussian distribution with zero mean and covariance according to:

$$
\Sigma =
\begin{bmatrix}
\sigma_p^2 & 0 & 0 & 0 & 0 & 0 & 0 & 0 & 0 & 0 & 0 & 0 & 0 & 0 & 0 & 0 \\
0 & \sigma_v^2 & 0 & 0 & 0 & 0 & 0 & 0 & 0 & 0 & 0 & 0 & 0 & 0 & 0 & 0 \\
0 & 0 & \sigma_f^2 & 0 & 0 & 0 & 0 & 0 & 0 & 0 & 0 & 0 & 0 & 0 & 0 & 0 \\
0 & 0 & 0 & \sigma_p^2 & 0 & 0 & 0 & 0 & 0 & 0 & 0 & 0 & 0 & 0 & 0 & 0 \\
0 & 0 & 0 & 0 & \sigma_v^2 & 0 & 0 & 0 & 0 & 0 & 0 & 0 & 0 & 0 & 0 & 0 \\
0 & 0 & 0 & 0 & 0 & \sigma_f^2 & 0 & 0 & 0 & 0 & 0 & 0 & 0 & 0 & 0 & 0 \\
0 & 0 & 0 & 0 & 0 & 0 & \sigma_p^2 & 0 & 0 & 0 & 0 & 0 & 0 & 0 & 0 & 0 \\
0 & 0 & 0 & 0 & 0 & 0 & 0 & \sigma_v^2 & 0 & 0 & 0 & 0 & 0 & 0 & 0 & 0 \\
0 & 0 & 0 & 0 & 0 & 0 & 0 & 0 & \sigma_f^2 & 0 & 0 & 0 & 0 & 0 & 0 & 0 \\
0 & 0 & 0 & 0 & 0 & 0 & 0 & 0 & 0 & \sigma_p^2 & 0 & 0 & 0 & 0 & 0 & 0 \\
0 & 0 & 0 & 0 & 0 & 0 & 0 & 0 & 0 & 0 & \sigma_v^2 & 0 & 0 & 0 & 0 & 0 \\
0 & 0 & 0 & 0 & 0 & 0 & 0 & 0 & 0 & 0 & 0 & \sigma_f^2 & 0 & 0 & 0 & 0 \\
0 & 0 & 0 & 0 & 0 & 0 & 0 & 0 & 0 & 0 & 0 & 0 & \sigma_p^2 & 0 & 0 & 0 \\
0 & 0 & 0 & 0 & 0 & 0 & 0 & 0 & 0 & 0 & 0 & 0 & 0 & \sigma_p^2 & 0 & 0 \\
0 & 0 & 0 & 0 & 0 & 0 & 0 & 0 & 0 & 0 & 0 & 0 & 0 & 0 & \sigma_p^2 & 0 \\
0 & 0 & 0 & 0 & 0 & 0 & 0 & 0 & 0 & 0 & 0 & 0 & 0 & 0 & 0 & \sigma_p^2
\end{bmatrix}
\tag{A21}
$$

See **Appendix 3—table 1** for the measurement noise values used in $\sigma$.

$C_i^{aug}$ is an observation matrix designed to selectively observe some of the delayed states. First we define $C_1$ and $C_2$:

$$
C_1 =
\begin{bmatrix}
1 & 0 & 0 & 0 & 0 & 0 & 0 & 0 & 0 & 0 & 0 & 0 & 0 & 0 & 0 & 0 \\
0 & 1 & 0 & 0 & 0 & 0 & 0 & 0 & 0 & 0 & 0 & 0 & 0 & 0 & 0 & 0 \\
0 & 0 & 1 & 0 & 0 & 0 & 0 & 0 & 0 & 0 & 0 & 0 & 0 & 0 & 0 & 0 \\
0 & 0 & 0 & 1 & 0 & 0 & 0 & 0 & 0 & 0 & 0 & 0 & 0 & 0 & 0 & 0 \\
0 & 0 & 0 & 0 & 1 & 0 & 0 & 0 & 0 & 0 & 0 & 0 & 0 & 0 & 0 & 0 \\
0 & 0 & 0 & 0 & 0 & 1 & 0 & 0 & 0 & 0 & 0 & 0 & 0 & 0 & 0 & 0 \\
0 & 0 & 0 & 0 & 0 & 0 & 1 & 0 & 0 & 0 & 0 & 0 & 0 & 0 & 0 & 0 \\
0 & 0 & 0 & 0 & 0 & 0 & 0 & 1 & 0 & 0 & 0 & 0 & 0 & 0 & 0 & 0 \\
0 & 0 & 0 & 0 & 0 & 0 & 0 & 0 & 1 & 0 & 0 & 0 & 0 & 0 & 0 & 0 \\
0 & 0 & 0 & 0 & 0 & 0 & 0 & 0 & 0 & 1 & 0 & 0 & 0 & 0 & 0 & 0 \\
0 & 0 & 0 & 0 & 0 & 0 & 0 & 0 & 0 & 0 & 1 & 0 & 0 & 0 & 0 & 0 \\
0 & 0 & 0 & 0 & 0 & 0 & 0 & 0 & 0 & 0 & 0 & 1 & 0 & 0 & 0 & 0 \\
0 & 0 & 0 & 0 & 0 & 0 & 0 & 0 & 0 & 0 & 0 & 0 & 1 & 0 & 0 & 0 \\
0 & 0 & 0 & 0 & 0 & 0 & 0 & 0 & 0 & 0 & 0 & 0 & 0 & 1 & 0 & 0 \\
0 & 0 & 0 & 0 & 0 & 0 & 0 & 0 & 0 & 0 & 0 & 0 & 0 & 0 & 1 & 0 \\
0 & 0 & 0 & 0 & 0 & 0 & 0 & 0 & 0 & 0 & 0 & 0 & 0 & 0 & 0 & 1
\end{bmatrix}
$$

$$C_2 = \begin{bmatrix} 1 & 0 & 0 & 0 & 0 & 0 & 0 & 0 & 0 & 0 & 0 & 0 & 0 & 0 & 0 & 0 & 0 \\ 0 & 1 & 0 & 0 & 0 & 0 & 0 & 0 & 0 & 0 & 0 & 0 & 0 & 0 & 0 & 0 & 0 \\ 0 & 0 & 0 & 1 & 0 & 0 & 0 & 0 & 0 & 0 & 0 & 0 & 0 & 0 & 0 & 0 & 0 \\ 0 & 0 & 0 & 0 & 1 & 0 & 0 & 0 & 0 & 0 & 0 & 0 & 0 & 0 & 0 & 0 & 0 \\ 0 & 0 & 0 & 0 & 0 & 1 & 0 & 0 & 0 & 0 & 0 & 0 & 0 & 0 & 0 & 0 & 0 \\ 0 & 0 & 0 & 0 & 0 & 0 & 0 & 1 & 0 & 0 & 0 & 0 & 0 & 0 & 0 & 0 & 0 \\ 0 & 0 & 0 & 0 & 0 & 0 & 0 & 0 & 1 & 0 & 0 & 0 & 0 & 0 & 0 & 0 & 0 \\ 0 & 0 & 0 & 0 & 0 & 0 & 0 & 0 & 0 & 1 & 0 & 0 & 0 & 0 & 0 & 0 & 0 \\ 0 & 0 & 0 & 0 & 0 & 0 & 0 & 0 & 0 & 0 & 1 & 0 & 0 & 0 & 0 & 0 & 0 \\ 0 & 0 & 0 & 0 & 0 & 0 & 0 & 0 & 0 & 0 & 0 & 1 & 0 & 0 & 0 & 0 & 0 \\ 0 & 0 & 0 & 0 & 0 & 0 & 0 & 0 & 0 & 0 & 0 & 0 & 1 & 0 & 0 & 0 & 0 \\ 0 & 0 & 0 & 0 & 0 & 0 & 0 & 0 & 0 & 0 & 0 & 0 & 0 & 1 & 0 & 0 & 0 \\ 0 & 0 & 0 & 0 & 0 & 0 & 0 & 0 & 0 & 0 & 0 & 0 & 0 & 0 & 1 & 0 & 0 \\ 0 & 0 & 0 & 0 & 0 & 0 & 0 & 0 & 0 & 0 & 0 & 0 & 0 & 0 & 0 & 1 & 0 \\ 0 & 0 & 0 & 0 & 0 & 0 & 0 & 0 & 0 & 0 & 0 & 0 & 0 & 0 & 0 & 0 & 1 \end{bmatrix}$$

Here controller $i$ observes all states except their partner's force production in both the x and y dimensions. We augment both observation matrices to only observe the delayed state.

$$C_i^{aug} = \begin{bmatrix} 0_{14 \times p} & C_i \end{bmatrix}$$

To minimize extra notation, we drop the *aug* superscript moving forward. Like previous work, we used a linear Kalman filter to model participants' sensory estimates of the state variables (**Todorov and Jordan, 2002**). The posterior state estimate $\hat{x}_{1,k}$ of controller 1 is obtained using an online filter of the form:

$$\hat{x}_{1,k} = \bar{x}_{1,k} + K_{1,k}(y_{1,k} - H_1\bar{x}_{1,k}) \tag{A22}$$

$$\bar{x}_{1,k} = A\hat{x}_{1,k-1} + B_1 u_{1,k} + B_2 u_{2,k} + \bar{\omega}_{1,k}. \tag{A23}$$

Here, $\bar{x}_{1,k}$ is the prior prediction of the state corrupted by Gaussian noise $\bar{\omega}_{1,k}$ with zero mean and standard deviation of $1e$-5. We assume the sensorimotor system obtains a noisy prior prediction of the states using an internal model of the state dynamics, which includes a prediction of the partner's motor command.

The prior prediction uses the previous posterior estimate ($\hat{x}_{1,k-1}$), the efference copy ($u_1$), and the prediction of the partner's motor command ($u_2$). The prior prediction of the state is updated using sensory measurements to obtain the posterior estimate $\hat{x}_1$ (**Equation A22**). The sequence of Kalman gains $K_1$ and $K_2$ were updated recursively according to the classic algorithm:

$$P_{i,k+1}^{prior} = A_d P_{i,k}^{post} A_d^{\mathsf{T}} + V_{i,k} \tag{A24}$$

$$S_{i,k+1} = C_i P_{i,k+1}^{prior} C_i^{\mathsf{T}} + W_{k+1} \tag{A25}$$

$$K_{i,k+1} = P_{i,k+1}^{prior} C_i^{\mathsf{T}} S_{i,k+1}^{-1} \tag{A26}$$

$$P_{i,k+1}^{post} = (I - K_{i,k+1} C_i)P_{i,k+1}^{prior} \tag{A27}$$

W is the measurement covariance matrix defined as:

$$W = \begin{bmatrix}
\sigma_w^2 & 0 & 0 & 0 & 0 & 0 & 0 & 0 & 0 & 0 & 0 & 0 & 0 & 0 & 0 & 0 \\
0 & \sigma_w^2 & 0 & 0 & 0 & 0 & 0 & 0 & 0 & 0 & 0 & 0 & 0 & 0 & 0 & 0 \\
0 & 0 & \sigma_w^2 & 0 & 0 & 0 & 0 & 0 & 0 & 0 & 0 & 0 & 0 & 0 & 0 & 0 \\
0 & 0 & 0 & \sigma_w^2 & 0 & 0 & 0 & 0 & 0 & 0 & 0 & 0 & 0 & 0 & 0 & 0 \\
0 & 0 & 0 & 0 & \sigma_w^2 & 0 & 0 & 0 & 0 & 0 & 0 & 0 & 0 & 0 & 0 & 0 \\
0 & 0 & 0 & 0 & 0 & \sigma_w^2 & 0 & 0 & 0 & 0 & 0 & 0 & 0 & 0 & 0 & 0 \\
0 & 0 & 0 & 0 & 0 & 0 & \sigma_w^2 & 0 & 0 & 0 & 0 & 0 & 0 & 0 & 0 & 0 \\
0 & 0 & 0 & 0 & 0 & 0 & 0 & \sigma_w^2 & 0 & 0 & 0 & 0 & 0 & 0 & 0 & 0 \\
0 & 0 & 0 & 0 & 0 & 0 & 0 & 0 & \sigma_w^2 & 0 & 0 & 0 & 0 & 0 & 0 & 0 \\
0 & 0 & 0 & 0 & 0 & 0 & 0 & 0 & 0 & \sigma_w^2 & 0 & 0 & 0 & 0 & 0 & 0 \\
0 & 0 & 0 & 0 & 0 & 0 & 0 & 0 & 0 & 0 & \sigma_w^2 & 0 & 0 & 0 & 0 & 0 \\
0 & 0 & 0 & 0 & 0 & 0 & 0 & 0 & 0 & 0 & 0 & \sigma_w^2 & 0 & 0 & 0 & 0 \\
0 & 0 & 0 & 0 & 0 & 0 & 0 & 0 & 0 & 0 & 0 & 0 & \sigma_w^2 & 0 & 0 & 0 \\
0 & 0 & 0 & 0 & 0 & 0 & 0 & 0 & 0 & 0 & 0 & 0 & 0 & \sigma_w^2 & 0 & 0 \\
0 & 0 & 0 & 0 & 0 & 0 & 0 & 0 & 0 & 0 & 0 & 0 & 0 & 0 & \sigma_w^2 & 0 \\
0 & 0 & 0 & 0 & 0 & 0 & 0 & 0 & 0 & 0 & 0 & 0 & 0 & 0 & 0 & \sigma_w^2
\end{bmatrix}$$

Both controllers used the same measurement covariance matrix $W$. $V_i$ is the process covariance matrix for controller $i$ defined as:

$$V_1 = \begin{bmatrix}
\sigma_s^2 & 0 & 0 & 0 & 0 & 0 & 0 & 0 & 0 & 0 & 0 & 0 & 0 & 0 & 0 & 0 \\
0 & \sigma_s^2 & 0 & 0 & 0 & 0 & 0 & 0 & 0 & 0 & 0 & 0 & 0 & 0 & 0 & 0 \\
0 & 0 & \sigma_s^2 & 0 & 0 & 0 & 0 & 0 & 0 & 0 & 0 & 0 & 0 & 0 & 0 & 0 \\
0 & 0 & 0 & \sigma_s^2 & 0 & 0 & 0 & 0 & 0 & 0 & 0 & 0 & 0 & 0 & 0 & 0 \\
0 & 0 & 0 & 0 & \sigma_s^2 & 0 & 0 & 0 & 0 & 0 & 0 & 0 & 0 & 0 & 0 & 0 \\
0 & 0 & 0 & 0 & 0 & \sigma_s^2 & 0 & 0 & 0 & 0 & 0 & 0 & 0 & 0 & 0 & 0 \\
0 & 0 & 0 & 0 & 0 & 0 & \sigma_z^2 & 0 & 0 & 0 & 0 & 0 & 0 & 0 & 0 & 0 \\
0 & 0 & 0 & 0 & 0 & 0 & 0 & \sigma_z^2 & 0 & 0 & 0 & 0 & 0 & 0 & 0 & 0 \\
0 & 0 & 0 & 0 & 0 & 0 & 0 & 0 & \sigma_z^2 & 0 & 0 & 0 & 0 & 0 & 0 & 0 \\
0 & 0 & 0 & 0 & 0 & 0 & 0 & 0 & 0 & \sigma_z^2 & 0 & 0 & 0 & 0 & 0 & 0 \\
0 & 0 & 0 & 0 & 0 & 0 & 0 & 0 & 0 & 0 & \sigma_z^2 & 0 & 0 & 0 & 0 & 0 \\
0 & 0 & 0 & 0 & 0 & 0 & 0 & 0 & 0 & 0 & 0 & \sigma_z^2 & 0 & 0 & 0 & 0 \\
0 & 0 & 0 & 0 & 0 & 0 & 0 & 0 & 0 & 0 & 0 & 0 & \sigma_s^2 & 0 & 0 & 0 \\
0 & 0 & 0 & 0 & 0 & 0 & 0 & 0 & 0 & 0 & 0 & 0 & 0 & \sigma_s^2 & 0 & 0 \\
0 & 0 & 0 & 0 & 0 & 0 & 0 & 0 & 0 & 0 & 0 & 0 & 0 & 0 & \sigma_s^2 & 0 \\
0 & 0 & 0 & 0 & 0 & 0 & 0 & 0 & 0 & 0 & 0 & 0 & 0 & 0 & 0 & \sigma_s^2
\end{bmatrix}$$

$$
V_2 = \begin{bmatrix}
\sigma_z^2 & 0 & 0 & 0 & 0 & 0 & 0 & 0 & 0 & 0 & 0 & 0 & 0 & 0 & 0 & 0 \\
0 & \sigma_z^2 & 0 & 0 & 0 & 0 & 0 & 0 & 0 & 0 & 0 & 0 & 0 & 0 & 0 & 0 \\
0 & 0 & \sigma_z^2 & 0 & 0 & 0 & 0 & 0 & 0 & 0 & 0 & 0 & 0 & 0 & 0 & 0 \\
0 & 0 & 0 & \sigma_z^2 & 0 & 0 & 0 & 0 & 0 & 0 & 0 & 0 & 0 & 0 & 0 & 0 \\
0 & 0 & 0 & 0 & \sigma_z^2 & 0 & 0 & 0 & 0 & 0 & 0 & 0 & 0 & 0 & 0 & 0 \\
0 & 0 & 0 & 0 & 0 & \sigma_z^2 & 0 & 0 & 0 & 0 & 0 & 0 & 0 & 0 & 0 & 0 \\
0 & 0 & 0 & 0 & 0 & 0 & \sigma_s^2 & 0 & 0 & 0 & 0 & 0 & 0 & 0 & 0 & 0 \\
0 & 0 & 0 & 0 & 0 & 0 & 0 & \sigma_s^2 & 0 & 0 & 0 & 0 & 0 & 0 & 0 & 0 \\
0 & 0 & 0 & 0 & 0 & 0 & 0 & 0 & \sigma_s^2 & 0 & 0 & 0 & 0 & 0 & 0 & 0 \\
0 & 0 & 0 & 0 & 0 & 0 & 0 & 0 & 0 & \sigma_s^2 & 0 & 0 & 0 & 0 & 0 & 0 \\
0 & 0 & 0 & 0 & 0 & 0 & 0 & 0 & 0 & 0 & \sigma_s^2 & 0 & 0 & 0 & 0 & 0 \\
0 & 0 & 0 & 0 & 0 & 0 & 0 & 0 & 0 & 0 & 0 & \sigma_s^2 & 0 & 0 & 0 & 0 \\
0 & 0 & 0 & 0 & 0 & 0 & 0 & 0 & 0 & 0 & 0 & 0 & \sigma_s^2 & 0 & 0 & 0 \\
0 & 0 & 0 & 0 & 0 & 0 & 0 & 0 & 0 & 0 & 0 & 0 & 0 & \sigma_s^2 & 0 & 0 \\
0 & 0 & 0 & 0 & 0 & 0 & 0 & 0 & 0 & 0 & 0 & 0 & 0 & 0 & \sigma_s^2 & 0 \\
0 & 0 & 0 & 0 & 0 & 0 & 0 & 0 & 0 & 0 & 0 & 0 & 0 & 0 & 0 & \sigma_s^2
\end{bmatrix}
$$

The process covariance is higher on the partner's states to more heavily weigh sensory information about the partner's states than the prediction of the partner's states.

## Simulating perturbation and probe trials

We Euler integrated the state equation with a step size of $h = 0.01$. Based on the state cost, the controllers were required to move the center cursor into either a relevant or irrelevant target. They were required to stop the center cursor in the target at final time $T = 0.8s$ (final timestep of $N = 79$). Just like the experiment, perturbations were implemented by shifting the center cursor –3 cm or target +3 cm when the center cursor crossed 25% of the forward distance to the target. On probe trials, we simulated a force channel by setting the x-force element in the $B_1$ and $B_2$ matrices to 0. Thus, the controllers could only move the center cursor in the forward dimension. We were able to calculate the applied force for controller 1 in the lateral dimension using the original $B_1$ matrix. During probe trials, the center cursor or target laterally shifted 3 cm when the center cursor crossed 25% of the distance to the target (6.25 cm), then shifted back to the original lateral position after 250 ms.

**Appendix 3—table 1.** Model parameters.

| Symbol | Value | Description |
| --- | --- | --- |
| $h$ | 0.01 s | Simulation time step |
| $T$ | 0.8 s | Total movement duration |
| $N$ | 79 | Final discretized time step |
| $m$ | 1.5 kg | Hand mass |
| $b$ | 0.1.Ns/m | Hand damping coefficient |
| $\tau$ | 20ms | Muscle time constant |
| $\delta v$ | 110ms | Visual feedback delay |
| $w_{cc_x}^{rel}$ | 40000 | Penalty on center cursor hitting a relevant target in x-dimension |
| $w_{cc_x}^{irrel}$ | 100 | Penalty on center cursor hitting an irrelevant target in x-dimension |

*Appendix 3—table 1 Continued on next page*

*Appendix 3—table 1 Continued*

| Symbol | Value | Description |
| --- | --- | --- |
| $w_{cc_y}$ | 40000 | Penalty on center cursor y-position |
| $w_v$ | 8000 | Penalty on hand velocity |
| $w_f$ | 40 | Penalty on force production |
| $w_y$ | 4000 | Penalty on difference in y-position between controllers 1 and 2 |
| $r$ | 1e-5 | Energy cost parameter |
| $\sigma_p^2$ | 1e-4 | Position measurement noise |
| $\sigma_v^2$ | 1e-3 | Velocity measurement noise |
| $\sigma_f^2$ | 1e-1 | Force measurement noise |
| $\sigma_u^2$ | 1e-2 | Control signal noise |
| $\sigma_w^2$ | 1e-2 | Measurement covariance value |
| $\sigma_s^2$ | 3e-4 | Process covariance value on self |
| $\sigma_z^2$ | 3e-2 | Process covariance value on partner |
| - | 250ms | Duration of probe perturbation |
| - | 0.03 m | Size of cursor/target jump |
| - | 0.0625 m | y-position of center cursor at cursor/target jump onset |

# Appendix 4

## Center cursor time-to-target

Previous work by *Česonis and Franklin, 2020* showed that time-to-target is a key variable the sensorimotor system uses to modify feedback responses. In their experiment, they manipulated the time-to-target of the participant's cursor while controlling for other movement parameters (e.g., distance from goal; *Česonis and Franklin, 2020*). When compared to classical optimal feedback control models, they showed that a model that modifies feedback responses based on time-to-target best predicted their results. In our task, it's possible that the time-to-target could have influenced visuomotor feedback responses since the distance to the center of the target is greater for a narrow target than a wide target on perturbation trials.

We calculated the time from perturbation onset to the center cursor reaching the forward position of the targets (*Appendix 4—figure 1*). In Experiment 1, an ANOVA with center cursor time-to-target as the dependent variable showed no main effect of self target ($F_{[1,47]}=2.45$, $p=0.124$) or partner target ($F_{[1,47]}=2.50$, $p=0.120$), nor any interaction ($F_{[1,47]}=1.97$, $p=0.166$). In Experiment 2, an ANOVA with center cursor time-to-target as the dependent variable showed a significant interaction ($F_{[1,47]}=5.87$, $p=0.019$). Post-hoc mean comparisons showed that only the difference between the *partner-irrelevant/self-irrelevant* and *partner-relevant/self-irrelevant* condition was significant ($p=0.006$). Although time-to-target and hand position are important variables for the control of movement (*Česonis and Franklin, 2022*; *Česonis and Franklin, 2020*; *Crevecoeur et al., 2013*), they are likely not driving factors of the different involuntary visuomotor feedback responses between our experimental conditions.

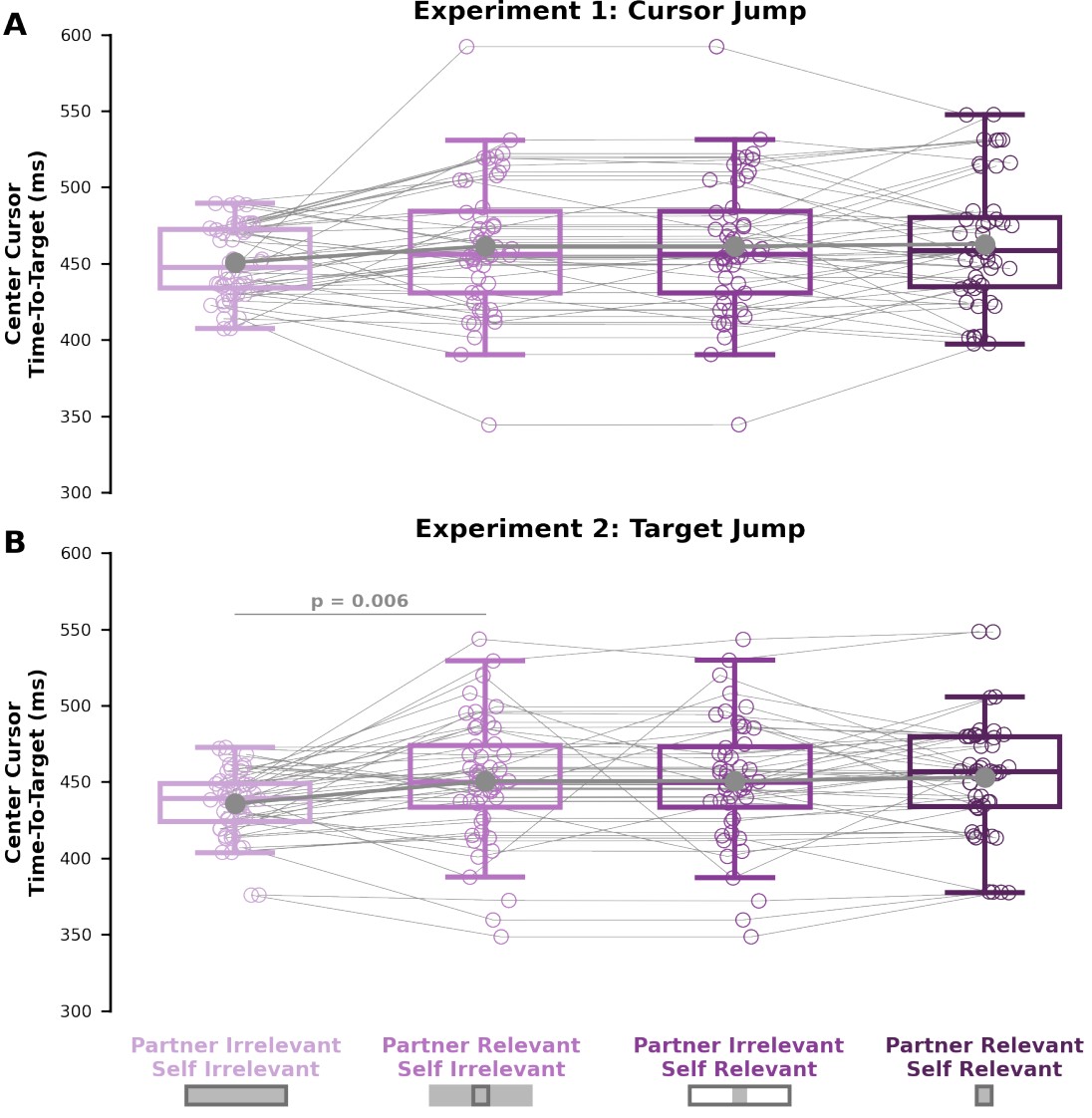

**Appendix 4—figure 1.** Center cursor time to target. Time from perturbation onset to the center cursor reaching the forward position of the targets (y-axis) for each experimental condition (x-axis) in Experiment 1 (**A**) and Experiment 2 (**B**).

# Appendix 5

## Participant forward hand position at perturbation onset

*Appendix 5—figure 1* shows the participant hand forward position at perturbation onset time for Experiment 1 (A) and Experiment 2 (B). It is possible that the participant forward hand position at perturbation onset time could influence their visuomotor feedback responses. Therefore, we ran an ANCOVA with self target and partner target as factors, and participant forward hand position at perturbation onset time as a covariate. In Experiment 1, we found no main effect of participant forward hand position on involuntary visuomotor feedback responses ($F[1,47]=1.466$, $p=0.228$). Further, when including the covariate, we still found a significant interaction between self target and partner target on involuntary visuomotor feedback responses ($F[1,47]=43.2$, $p < 0.001$).

In Experiment 2, we found a significant main effect of participant forward hand position on involuntary visuomotor feedback responses ($F[1,47]=6.73$, $p=0.010$). We still found a significant interaction between self target and partner target ($F[1,47]=9.78$, $p=0.002$). Since we found a main effect of participant forward hand position, we calculated the adjusted means of the involuntary visuomotor feedback responses. We then performed follow-up mean comparisons on the adjusted means of the involuntary visuomotor feedback responses (using emmeans in R). We found the same significant trends as the unadjusted means in the main manuscript. Specifically, we found involuntary visuomotor feedback responses to be: significantly greater in the *partner-relevant/self-irrelevant* condition compared to the *partner-irrelevant/self-irrelevant* condition ($p=0.003$), significantly greater in the *partner-relevant/self-irrelevant* condition compared to the *partner-irrelevant/self-relevant* condition ($p < 0.001$), significantly greater in the *partner-relevant/self-relevant* condition compared to the *partner-relevant/self-irrelevant* condition ($p < 0.001$), and not different between the *partner-irrelevant/self-relevant* and *partner-relevant/self-relevant* conditions ($p=0.381$).

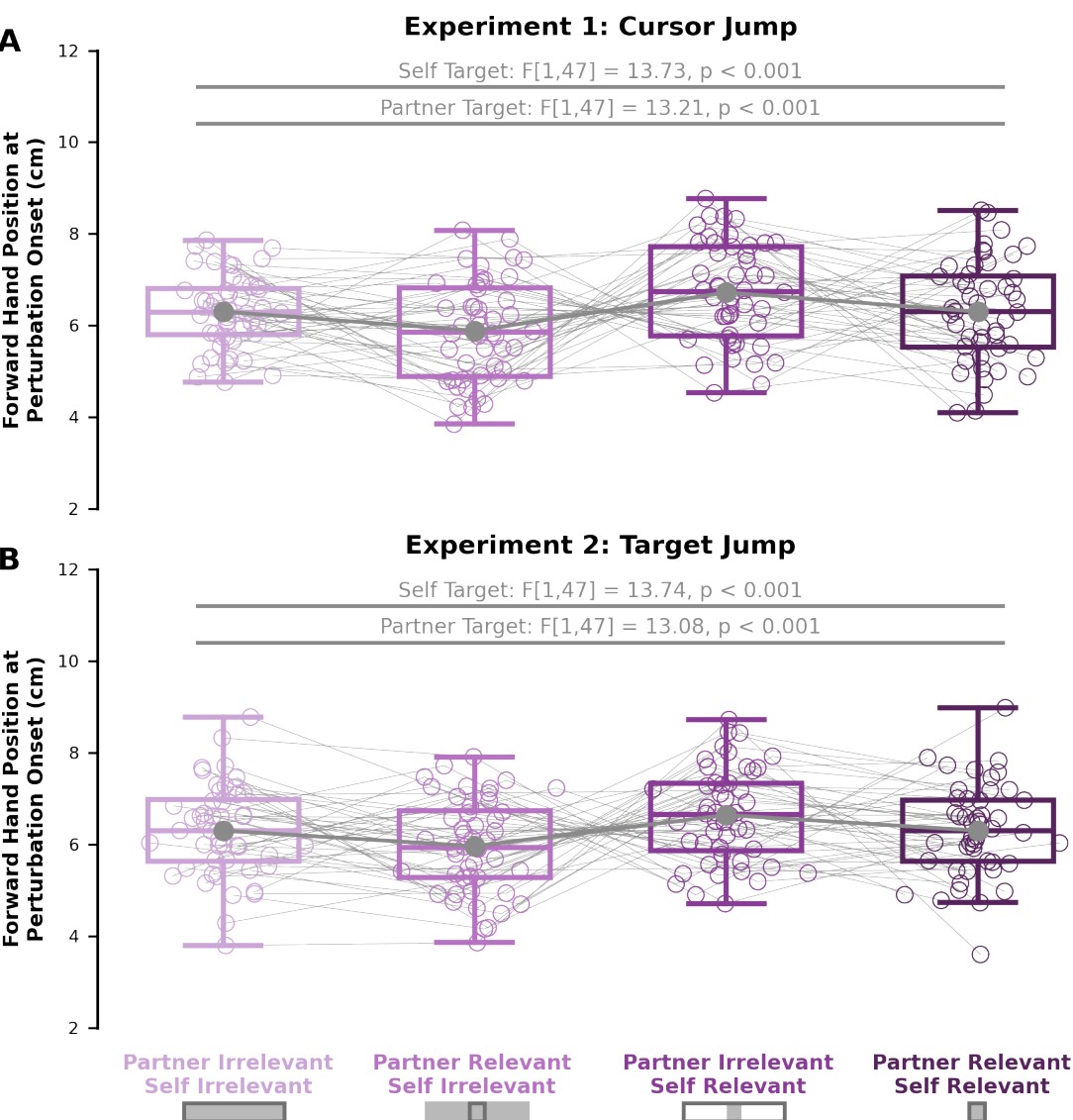

**Appendix 5—figure 1.** Participant forward hand position at perturbation onset. We calculated the participant forward hand position at perturbation onset for Experiment 1 (**A**) and Experiment 2 (**B**). We found significant main effects of self target and partner target for both experiments.

## Appendix 6

### Movement onset synchrony

We calculated movement onset times at the time that the participants left the start target (*Liu et al., 2025*). We then took the absolute value of the difference between the participants within a pair as a measure of movement onset synchrony. For Experiment 1, an ANOVA with movement onset synchrony as the dependent variable showed no main effect of self target (F[1,47]=1.38, p=0.252), no main effect of partner target (F[1,47]=0.057, p=0.813), and no interaction (F[1,47]=0.45, p=0.508). For Experiment 2, an ANOVA with movement onset synchrony as the dependent variable showed no main effect of self target (F[1,47]=0.07, p=0.788), no main effect of partner target (F[1,47]=2.75, p=0.111), and no interaction (F[1,47]=2.31, p=0.142). Movement onset synchrony results are shown in *Appendix 6—figure 1*.

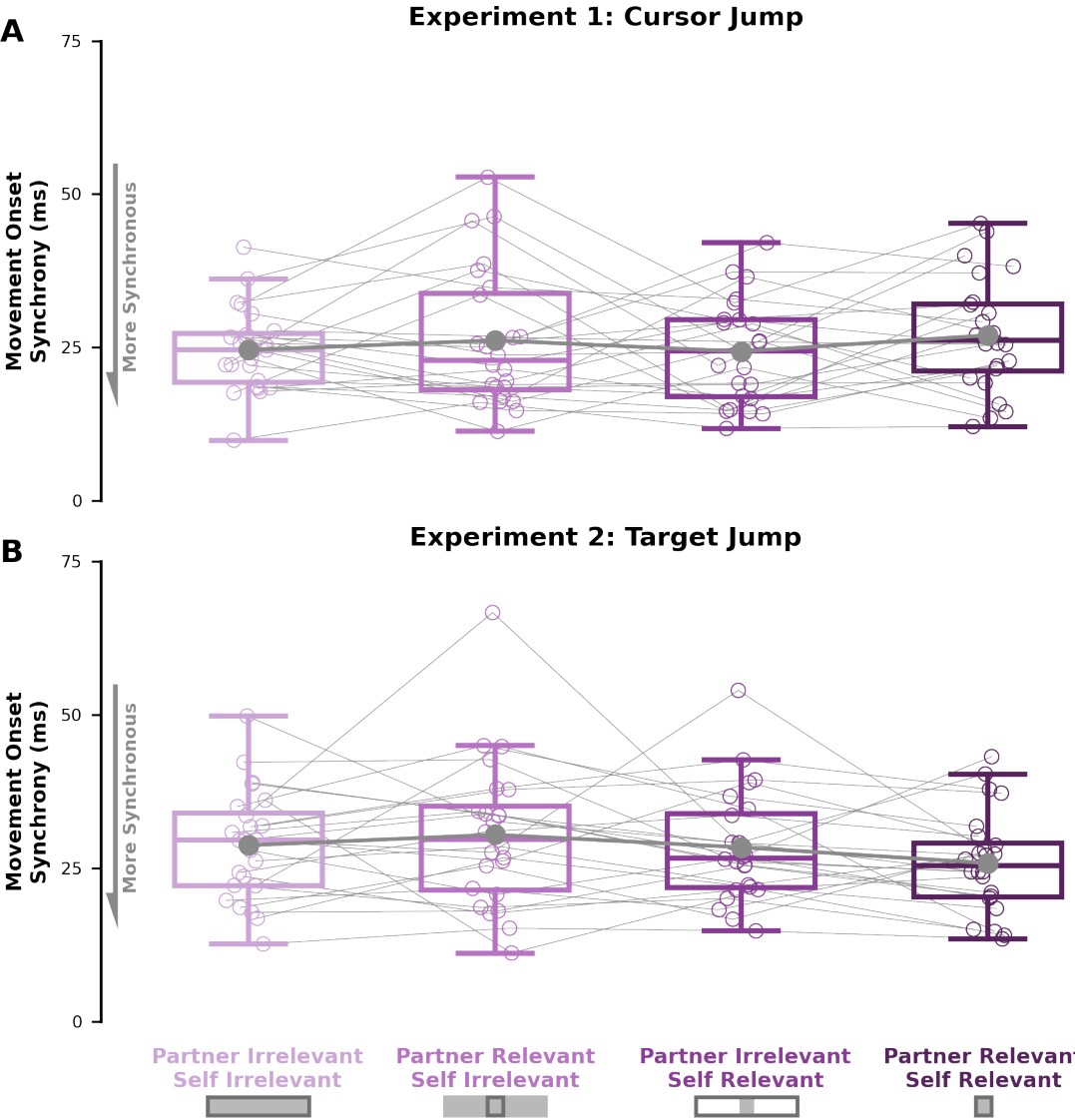

**Appendix 6—figure 1.** Movement onset synchrony. Here we show movement onset synchrony (y-axis) for each experimental condition (x-axis) in Experiment 1 (**A**) and Experiment 2 (**B**).

## Appendix 7

### Trial-by-trial involuntary visuomotor feedback responses

Given there were 151 trials and 10 left/right probe trials for each experimental condition, it is possible that completing more trials may have led to different involuntary visuomotor feedback responses. Therefore, we analyzed the involuntary visuomotor feedback responses over the course of each experimental condition. Visually, involuntary visuomotor feedback responses in neither Experiment 1 (*Appendix 7—figure 1*) nor Experiment 2 (*Appendix 7—figure 2*) show any consistent learning (see *Appendix 8—figure 1* for statistical analysis). Therefore, it appears participants rapidly formed a partner model that influenced their involuntary visuomotor feedback responses.

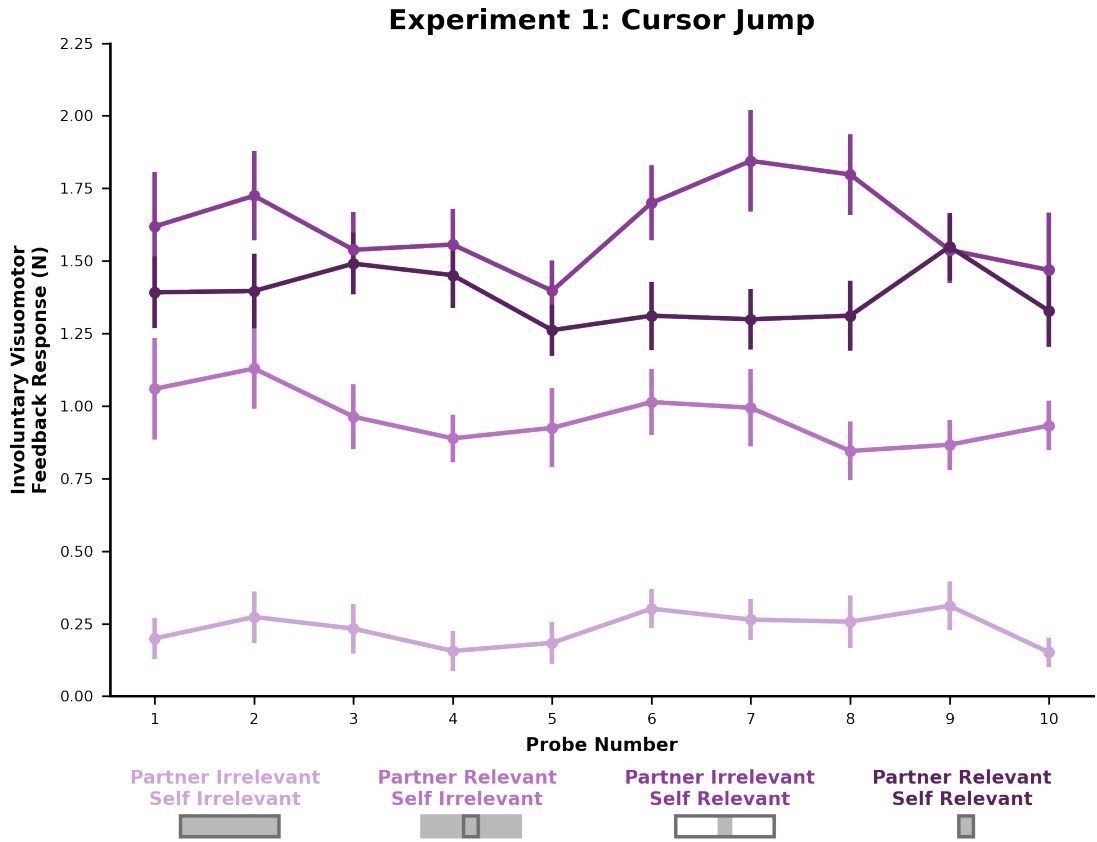

**Appendix 7—figure 1.** Experiment 1: trial-by-trial involuntary visuomotor feedback responses. Here, we show the Experiment 1 involuntary visuomotor feedback responses (y-axis) over each block (x-axis). We did not see any significant effect of learning within each experimental block (see Appendix 8 as well). Circles reflect the group mean, and vertical bars reflect the standard error of the mean.

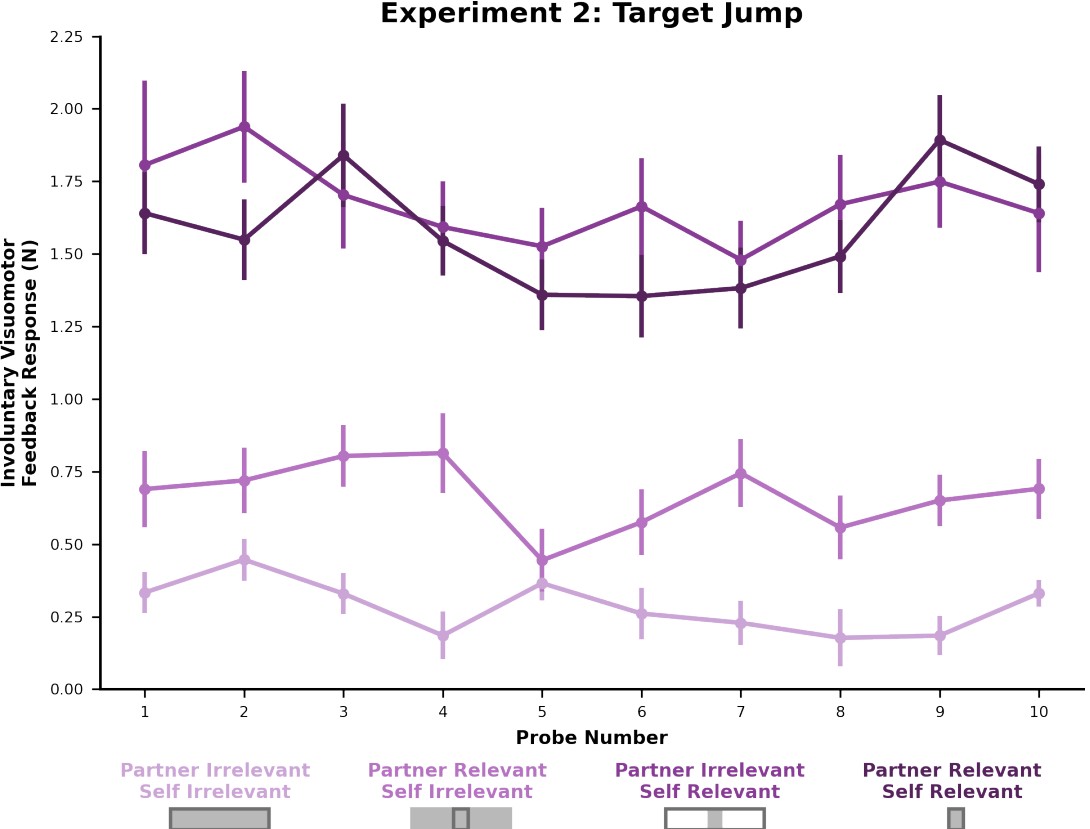

**Appendix 7—figure 2.** Experiment 1: trial-by-trial involuntary visuomotor feedback responses. Here, we show the Experiment 2 involuntary visuomotor feedback responses (y-axis) over each block (x-axis). We did not see any significant effect of learning within each experimental block (see Appendix 8 as well). Colors correspond to each blocked experimental condition; circles reflect the group mean, and vertical bars reflect the standard error of the mean.

# Appendix 8

## First half vs. second half involuntary visuomotor feedback responses

*Appendix 8—figure 1* shows the involuntary visuomotor feedback responses in the first half (A, C) and second half (B, D) for each experimental condition. In Experiment 1, we observed the same statistical results in the first half and second half of trials as the analysis of all trials. That is, we observed a significant interaction between self target and partner target in the first half ($F_{[1,47]}=37.09$, $p < 0.001$) and second half ($F_{[1,47]}=48.68$, $p < 0.001$) of trials. Follow-up mean comparisons showed the same significant trends as our analysis of all trials in the main manuscript (see *Appendix 8—figure 1A–B*).

In Experiment 2, we observed the same statistical results in the first half and second half of trials as the analysis of all trials. That is, we observed a significant interaction between self target and partner target in the first half ($F_{[1,47]}=9.42$, $p=0.004$) and second half ($F_{[1,47]}=17.40$, $p < 0.001$) of trials. Follow-up mean comparisons showed the same significant trends as our analysis of all trials in the main manuscript (*Appendix 8—figure 1C–D*).

Showing the same involuntary visuomotor feedback response trends across the experimental conditions for the first half, second half, and all trials suggests that the sensorimotor system used a model of a partner based on their goals and considered their costs to modify rapid motor responses.

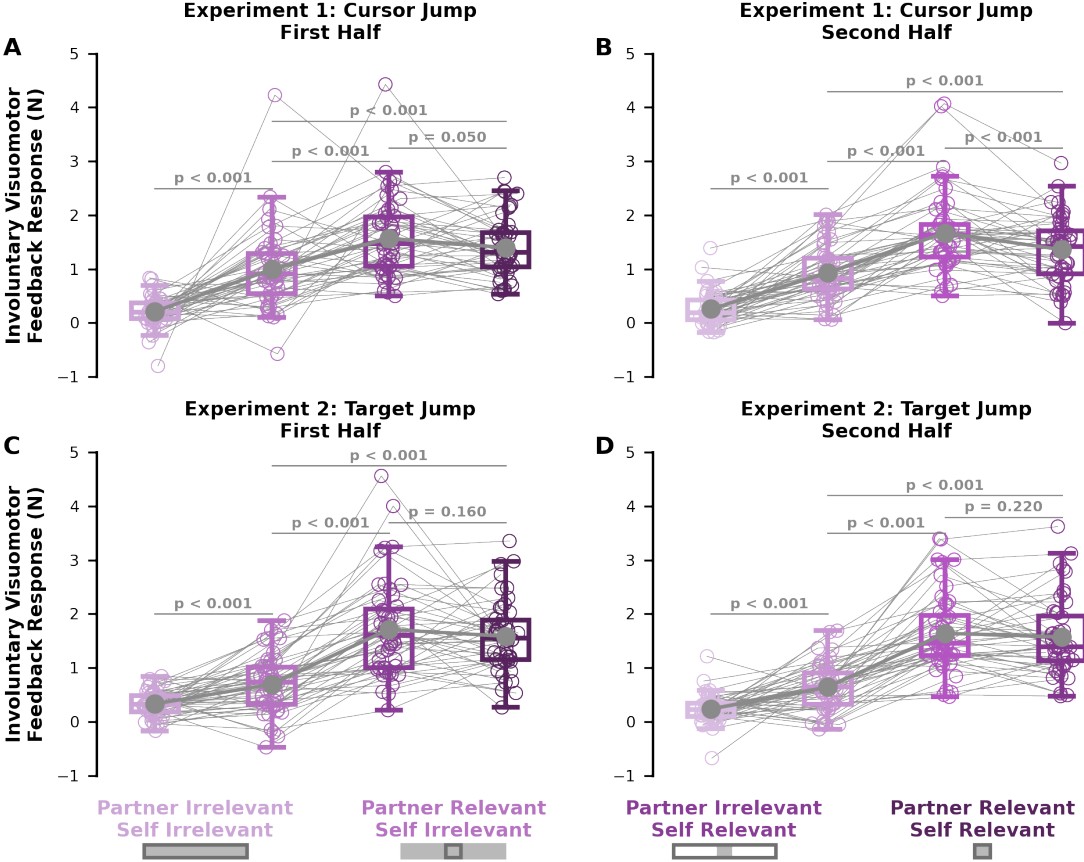

**Appendix 8—figure 1.** First half vs second half involuntary visuomotor feedback responses. Here, we show the first half (first column) and second half (second column) of trials for Experiment 1 (first row) and Experiment 2 (second row). In both Experiment 1 and Experiment 2, we found the same significant differences in involuntary visuomotor feedback responses between conditions for the first half and second half of trials analyzed separately. Additionally, these trends matched the analysis of all trials together in the main manuscript.

