## [Editor Report · eLife Assessment]

This **important** study combines a two-person joint hand-reaching paradigm with game-theoretical modeling to examine whether, and how, reflexive visuomotor responses are modulated by a partner's control policy and cost structure. The study provides a **convincing** set of behavioral findings suggesting that involuntary visuomotor feedback is indeed modulated in the context of interpersonal coordination. The work will be of interest to cognitive scientists studying the motor and social aspects of action control.

---

## [Referee Report · Reviewer #1 (Public review)]

Summary:

Sullivan and colleagues examined the modulation of reflexive visuomotor responses during collaboration between pairs of participants performing a joint reaching movement to a target. In their experiments, the players jointly controlled a cursor that they had to move towards narrow or wide targets. In each experimental block, each participant had a different type of target they had to move the joint cursor to. During the experiment, the authors used lateral perturbation of the cursor to test participants' fast feedback responses to the different target types. The authors suggest participants integrate the target type and related cost of their partner into their own movements, which suggests that visuomotor gains are affected by the partner's task.

Strengths:

The topic of the manuscript is very interesting, and the authors are using well-established methodology to test their hypothesis. They combine experimental studies with optimal control models to further support their work. Overall, the manuscript is very timely and shows important findings - that the feedback responses reflect both our and our partners tasks.

---

## [Referee Report · Reviewer #2 (Public review)]

Summary:

Sullivan and colleagues studied the fast, involuntary, sensorimotor feedback control in interpersonal coordination. Using a cleverly designed joint-reaching experiment that separately manipulated the accuracy demands for a pair of participants, they demonstrated that the rapid visuomotor feedback response of a human participant to a sudden visual perturbation is modulated by his/her partner's control policy and cost. The behavioral results are well matched with the predictions of the optimal feedback control framework implemented with the dynamic game theory model. Overall, the study provides an important and novel set of results on the fast, involuntary feedback response in human motor control in the context of interpersonal coordination.

Review:

Sullivan and colleagues investigated whether fast, involuntary sensorimotor feedback control is modulated by the partner's state (e.g., cost and control policy) during interpersonal coordination. They asked a pair of participants to make a reaching movement to control a cursor and hit a target, where the cursor's position was a combination of each participant's hand position. To examine fast visuomotor feedback response, the authors applied a sudden shift in either the cursor (experiment 1) or the target (experiment 2) position in the middle of movement. To test the involvement of partner's information in the feedback response, they independently manipulated the accuracy demand for each participant by varying the lateral length of the target (i.e., a wider/narrower target has a lower/higher demand for correction when movement is perturbed). Because participants could also see their partner's target, they could theoretically take this information (e.g., whether their partner would correct, whether their correction would help their partner, etc.) into account when responding to the sudden visual shift. Computationally, the task structure can be handled using dynamic game theory, and the partner's feedback control policy and cost function are integrated into the optimal feedback control framework. As predicted by the model, the authors demonstrated that the rapid visuomotor feedback response to a sudden visual perturbation is modulated by the partner's control policy and cost. When their partner's target was narrow, they made rapid feedback corrections even when their own target was wide (no need for correction), suggesting integration of their partner's cost function. Similarly, they made corrections to a lesser degree when both targets were narrower than when the partner's target was wider, suggesting that the feedback correction takes the partner's correction (i.e., feedback control policy) into account.

The strength of the current paper lies in the combination of clever behavioral experiments that independently manipulate each participant's accuracy demand and a sophisticated computational approach that integrates optimal feedback control and dynamic game theory. Both the experimental design and data analysis sound good and the main claim is well supported by the results.

A future direction would be to investigate how this mechanism is implemented in the CNS and to examine whether the same cooperative mechanism also applies to human-AI interactions.

---

## [Author Response]

The following is the authors’ response to the original reviews.

**Reviewer #1 (Public review):**
SummarySullivan and colleagues examined the modulation of reflexive visuomotor responses during collaboration between pairs of participants performing a joint reaching movement to a target. In their experiments, the players jointly controlled a cursor that they had to move towards narrow or wide targets. In each experimental block, each participant had a different type of target they had to move the joint cursor to. During the experiment, the authors used lateral perturbation of the cursor to test participants’ fast feedback responses to the different target types. The authors suggest participants integrate the target type and related cost of their partner into their own movements, which suggests that visuomotor gains are affected by the partner’s task.StrengthsThe topic of the manuscript is very interesting, and the authors are using well established methodology to test their hypothesis. They combine experimental studies with optimal control models to further support their work. Overall, the manuscript is very timely and shows important findings - that the feedback responses reflect both our and our partner’s tasks.

We thank the reviewer for the positive comments regarding our work.

WeaknessesHowever, in the current version of the manuscript, I believe the results could also be interpreted differently, which suggest that the authors should provide further support for their hypothesis and conclusions.Major Comments(1) Results of the relevant conditions:In addition to the authors’ explanation regarding the results, it is also possible that the results represent a simple modulation of the reflexive response to a scaled version of cursor movement. That is, when the cursor is partially controlled by a partner, which also contributes to reducing movement error, it can also be interpreted by the sensorimotor system as a scaling of hand-to-cursor movement. In this case, the reflexes are modulated according to a scaling factor (how much do I need to move to bring the cursor to the target). I believe that a single-agent simulation of an OFC model with a scaling factor in the lateral direction can generate the same predictions as those presented by the authors in this study. In other words, maybe the controller has learned about the nature of the perturbation in each specific context, that in some conditions I need to control strongly, whereas in others I do not (without having any model of the partner). I suggest that the authors demonstrate how they can distinguish their interpretation of the results from other explanations.

We thank the reviewer for the thoughtful comment. While it is possible that the change in the visuomotor feedback responses could be just from a scaling factor. This hypothesis could explain the difference between two conditions, but would fail to explain differences between two other conditions. Specifically, this hypothesis could explain a decrease in involuntary visuomotor feedback responses between partner-irrelevant/self-relevant and partner-relevant/self-relevant. Critically, this hypothesis could not explain the difference between partner-irrelevant/self-irrelevant and partner-relevant/self-irrelevant. That is, there is no reason to scale a response to correct for a partner’s relevant target when your own target is irrelevant. However, our finding that there is a greater involuntary visuomotor feedback response in partner-relevant/self-irrelevant compared to partner-irrelevant/self-irrelevant is predicted by the notion that humans form a representation of others and consider their movement costs.

We have added a paragraph in the discussion to justify our hypothesis over the scaling factor hypothesis.

“Our hypothesis that the sensorimotor system uses a representation of a partner and considers the partner’s costs to modify involuntary visuomotor feedback responses can parsimoniously explain all of our experimental findings. There are a few alternative hypotheses that could explain a subset of results. One alternative hypothesis is that participants simply learned the hand to center cursor mapping in each experimental condition. That is, instead of using a model of their partner, participants simply adapted to the dynamics of the center cursor. However, this hypothesis would not predict an increased involuntary visuomotor feedback response in the partner-relevant/self-irrelevant condition compared to the partner-irrelevant/self-irrelevant condition. If participants did not form a model of their partner nor consider their partner’s costs, then they would not display an increased feedback response when they had an irrelevant target and their partner’s target was relevant. An increased feedback response to help a partner achieve their goal is captured by our hypothesis that the sensorimotor system uses a representation of a partner and considers the partner’s costs to modify involuntary visuomotor feedback responses.”

(2) The effect of the partner target:The authors presented both self and partner targets together. While the effect of each target type, presented separately, is known, it is unclear how presenting both simultaneously affects individual response. That is, does a small target with a background of the wide target affect the reflexive response in the case of a single participant moving? The results of Experiment 2, comparing the case of partner- and self-relevant targets versus partner-irrelevant and self-relevant targets, may suggest that the system acted based on the relevant target, regardless of the presence and instructions regarding the self-target.

We thank the reviewer for bringing up another valid point, which we discussed at length as a group when designing the experiment. The reviewer is correct in pointing out the lack of difference in the involuntary epoch between the partner-relevant/self-relevant and partner-irrelevant/self-relevant could potentially suggest that the sensorimotor system acted based on only relevant targets, irrespective if it was a self or partner relevant target. While the effect of the simultaneous presentation of a narrow and wide target on an individual’s response by themselves is unknown, comparing the differences between our other experimental conditions control for this potential confound. Participants viewed a wide target and a narrow target on the screen, in both the partner-irrelevant/self-relevant condition and the partner-relevant/self-irrelevant condition. Crucially, we found that the visuomotor feedback responses were greater in the partner-irrelevant/self-relevant condition compared to the partner-relevant/self-irrelevant condition in both Experiment 1 and 2. That is, participants were able to distinguish between the self-target and partner target and appropriately modify their feedback responses in both Experiment 1 and 2, despite there being both a wide and narrow target on the screen in both conditions. Given that we found different visuomotor feedback responses between the two conditions that had both a narrow and wide target, this rules out the alternative hypothesis that the sensorimotor system acted based just on a relevant target being present. We have added to our discussion to clarify this point.

“Another alternative hypothesis would be that the sensorimotor system was responding only to the relevant target displayed on the screen. Again, this hypothesis would only explain a subset of our results. In particular, this relevant target hypothesis cannot explain the observed feedback response differences between the partner-relevant/self-irrelevant and partner-irrelevant/self-relevant conditions in both Experiments 1 and 2.”

(3) Experiment instructions:It is unclear what the general instructions were for the participants and whether the instructions provided set the proposed weighted cost, which could be altered with different instructions.

Our instructions explicitly informed participants that their performance bonus was only based on them stabilizing within their own self-target within the time constraint. We have added the following in the methods to emphasize this instruction.

“In other words, we ensured participants had a clear understanding that their performance in the task was only based on stabilizing the center cursor in their own self-target within the time constraint. Therefore, the instructions and timing constraints did not enforce participants to work together.”

(4) Some work has shown that the gain of visuomotor feedback responses reflects the time to target and that this is updated online after a perturbation (Cesonis & Franklin, 2020, eNeuro; Cesonis and Franklin, 2021, NBDT; also related to Crevecoeur et al., 2013, J Neurophysiol). These models would predict different feedback gains depending on the distance remaining to the target for the participant and the time to correct for the jump, which is directly affected by the small or large targets. Could this time be used to target instead of explaining the results? I don’t believe that this is the case, but the authors should try to rule out other interpretations. This is maybe a minor point, but perhaps more important is the location (&time remaining) for each participant at the time of the jump. It appears from the figures that this might be affected by the condition (given the change in movement lengths - see Figure 3 B & C). If this is the case, then could some of the feedback gain be related to these parameters and not the model of the partner, as suggested? Some evidence to rule this out would be a good addition to the paper - perhaps the distance of each partner at the time of the perturbation, for example. In addition, please analyze the synchrony of the two partners’ movements.

(1) Time to target and forward position

The reviewer raises an interesting point. In our task, the cursor/target jump occurs once the center cursor crosses 6.25 cm from the start. We analyzed the time it took for the center cursor to intercept the targets from perturbation onset (Supplementary D). In Experiment 1, an ANOVA with center cursor time-to-target as the dependent variable showed no main effect of self-target (F[1,47] = 2.45, p = 0.124) or partner target (F[1,47] = 2.50, p=0.120), nor any interaction (F[1,47] = 1.97, p = 0.166). In Experiment 2, an ANOVA with center cursor time-to-target as the dependent variable showed a significant interaction (F[1,47] = 5.87, p = 0.019). Post-hoc mean comparisons showed that only the difference between the partner-irrelevant/self-irrelevant and partner-relevant/self-irrelevant condition was significant (p = 0.006). Given that only one comparison in Experiment 2 showed a difference in time-to-target, we do not believe that time-to-target was a significant driver of the change in involuntary visuomotor feedback responses observed between conditions. While time-to-target is likely a metric the nervous system modifies feedback gains around, our results suggest that the nervous system can also use a partner model to modify feedback gains. We have added a supplemental analysis on time to target

“Previous work by Česonis and Franklin (2020) showed that time to-target is a key variable the sensorimotor system uses to modify feedback responses. In their experiment, they manipulated the time-to-target of the participant’s cursor, while controlling for other movement parameters (e.g., distance from goal) [1]. When compared to classical optimal feedback control models, they showed that a model that modifies feedback responses based on time-to-target best predicted their results. In our task, it’s possible that the time-to-target could have influenced visuomotor feedback responses, since the distance to the center of the target is greater for a narrow target than a wide target on perturbation trials.”

“We calculated the time from perturbation onset to the center cursor reaching the forward position of the targets (Supplementary Fig. S5). In Experiment1, an ANOVA with center cursor time-to-target as the dependent variable showed no main effect of self-target (F[1,47]=2.45,p=0.124) or partner target (F[1,47] = 2.50, p=0.120), nor any interaction (F[1,47] = 1.97, p = 0.166). In Experiment2, an ANOVA with center cursor time-to-target as the dependent variable showed a significant interaction (F [1,47] = 5.87, p = 0.019). Post-hoc mean comparisons showed that only the difference between the partner-irrelevant/self-irrelevant and partner-relevant/self-irrelevant condition was significant (p=0.006). Although time-to-target and hand position are important variables for the control ofmovement,[1,2,3] they are likely not driving factors of the different in voluntary visuomotor feedback responses between our experimental conditions.”

However, it is possible that the participant forward position at perturbation onset could also influence the involuntary feedback response. We show the forward positions at perturbation onset in Supplementary D. Statistical analysis of the forward positions in Experiment 1 showed a main effect of self-target (F[1,47] = 12.72, p < 0.001), main effect of partner target (F[1,47] = 12.82, p < 0.001), and no interaction (F[1,47] = 0.00, P = 0.991). We see the same trend in experiment 2, showing a main effect of self-target (F[1,47] = 12.11, p < 0.001), main effect of partner target (F[1,47] = 12.04, p < 0.001), and no interaction (F[1,47] = 0.00, p = 0.986). The fact that there was no interaction implies that the results could not solely be due to forward position. Nevertheless, given there were main effects, we proceeded to run an ANCOVA on the involuntary visuomotor feedback responses with forward position as a covariate. For experiment 1, we still observed a significant interaction between self and partner target (F[1,47] = 43.14, p < 0.001). Further, we also observed no significant main effect of forward position on the involuntary visuomotor feedback responses. The ANCOVA for Experiment 2 also showed that there was still a significant interaction of self and partner target on the involuntary visuomotor feedback responses (F[1,47] = 9.80, p = 0.002). However, here we did find a significant main effect of the forward position (F[1,47] = 5.06, p = 0.026). Therefore, we ran follow-up mean comparisons with the covariate adjusted means. We found the same statistical trend as reported in the main results. We found significant differences between the partner-irrelevant/self-irrelevant and partner-relevant/self-irrelevant conditions (p = 0.003), partner-relevant/self-irrelevant and partner-irrelevant/self-relevant conditions (p < 0.001), partner-relevant/self-irrelevant and partner-relevant/self-relevant conditions (p < 0.001). We found no significant difference between the partner-irrelevant/self-relevant and partner-relevant/self-relevant conditions (p = 0.381). Given that there was no main effect of forward position in Experiment 1, and that our adjusted mean comparisons in Experiment 2 showed the same trends as the unadjusted mean comparisons in the main manuscript, our results show that the forward position of the participants is not a significant factor in explaining the differences in involuntary visuomotor feedback responses between conditions.

“Supplementary Fig. 6 shows the participant hand forward position at perturbation onset time for Experiment 1 (A) and Experiment 2 (B). It is possible that the participant forward hand position at perturbation onset time could influence their visuomotor feedback responses. Therefore, we ran an ANCOVA with self-target and partner target as factors, and participant forward hand position at perturbation onset time as a covariate. In Experiment 1, we found no main affect of participant forward hand position on involuntary visuomotor feedback responses (F[1,47] = 1.466, p = 0.228). Further, when including the covariate, we still found a significant interaction between self-target and partner target on in voluntary visuomotor feedback responses (F[1,47]=43.2, p<0.001).”

“In Experiment 2, we found a significant main effect of participant forward hand position on involuntary visuomotor feedback responses (F[1,47] = 6.73, p = 0.010). We still found a significant interaction between self-target and partner target (F[1,47] = 9.78, p = 0.002). Since we found a main effect of participant forward hand position, we calculated the adjusted means of the involuntary visuomotor feedback responses. We then performed follow-up mean comparisons on the adjusted means of the involuntary visuomotor feedback responses (using emmeans in R). We found the same significant trends as the unadjusted means in the main manuscript. Specifically we found involuntary visuomotor feedback responses to be: significantly greater in the partner-relevant/self-irrelevant condition compared to the partner-irrelevant/self-irrelevant condition (p = 0.003),significantly greater in the partner-relevant/self-irrelevant condition compared to the partner-irrelevant/self-relevant condition (p<0.001), significantly greater in the partner-relevant/self-relevant condition compared to the partner-relevant/self-irrelevant condition (p<0.001),and not different between the partner-irrelevant/self-relevant and partner-relevant/self-relevant conditions (p = 0.824).”

We have also included in the discussion how time-to-target and participant forward hand position are important control variables to consider, and their potential relationship to our findings.

“Finally, we also considered whether time to target [1,2]. (Supplementary D), participant forward hand position (Supplementary E), or learning [4] (Supplementary G-H) influenced feedback responses, but found that none impacted the observed differences between experimental conditions nor changed our interpretation. Our hypothesis that the sensorimotor system uses a representation of a partner and considers the partner’s costs to modify involuntary visuomotor feedback responses parsimoniously accounts for the differences observed between all conditions.”

(2) Synchrony

In our task, participants movements were not self-initiated. We had them begin the movement as soon as they hear an audible tone so that they would begin their movements at as similar a time as possible. We have analyzed the movement onset synchrony between participants within a pair, shown in Supplementary F.

Supplementary: “We calculated movement onset times at the time that the participants left the start target [8]. We then took the absolute value of the difference between the participants within a pair as a measure of movement onset synchrony. For Experiment 1, an ANOVA with movement onset synchrony as the dependent variable showed no main effect of self-target (F[1,47] = 1.38, p = 0.252), no main effect of partner target (F[1,47] = 0.057, p = 0.813), and no interaction (F[1,47] = 0.45, p = 0.508). For Experiment 2, an ANOVA with movement onset synchrony as the dependent variable showed no main effect of self-target (F[1,47] = 0.07, p = 0.788), no main effect of partner target (F[1,47] = 2.75, p = 0.111), and no interaction (F[1,47] = 2.31, p = 0.142).”

Further, we have modified our methods to emphasize that participants within a pair generally began their movement at the same time.

“Instead of self-initiating their movements, we specifically had participants move at the sound of a tone so that the movement onset between participants in a pair was as synchronous as possible (see Supplementary F for movement onset synchrony analysis).”

**Reviewer #1 (Recommendations for the authors):**
(1) Lines 291-292: One study extensively examined cursor and target jump visuomotor on set times and found no difference (Franklin et al., 2016; J Neuroscience), which strongly argues against this interpretation.

We thank the reviewer for pointing out this work. We have modified the following lines:

“However, other work by Franklin and colleagues (2016) found no difference in visuomotor feedback response latencies between cursor and target jumps [6].”

(2) Line 411: What were the instructions regarding partner performance in terms of the reward? Did you explain that individual performance alone will determine the reward?

As addressed above, we have made the following changes to emphasize the instructions given to participants.

“In other words, we ensured participants had a clear understanding that their performance in the task was only based on stabilizing the center cursor in their own self-target within the time constraint. Therefore, the instructions and timing constraints did not enforce participants to work together.”

(3) Line 506: Ten probe trials in each direction is very low. Can this still be in the transition state of the feedback response, rather than at steady state? There are many studies done looking at the learning of visuomotor responses in which changes are still occurring after several hundred trials (e.g., Franklin et al., 2017 J Neurophysiol; Franklin et al., 2008; J Neuroscience). In this experiment, each block only lasts 151 trials total if my calculations are correct. How certain are you that the results are at a steady state and not continuously changing? Perhaps with further experimental experience, the feedback responses would approach the predictions of a different model.

The reviewer raises an important point. We had run these analyses prior to submitting the manuscript and did not see anything. However, we believe this information is important to include since both we and yourself asked the same question. Specifically, we have analyzed the visuomotor feedback responses over the trials (Supplementary G), which shows little to no learning over time. Additionally, we also found no difference in the visuomotor feedback response trends between the first and second half of trials in each condition (Supplementary H). Therefore, it appears that the sensorimotor system was at steady state behaviour very quickly and we do believe that the feedback responses would approach the predictions of a different model if participants performed more trials. We have added the following

Supplementary: “Given there were 151 trials and 10 left/right probe trials for each experimental condition, it is possible that completing more trials may have lead to different involuntary visuomotor feedback responses. Therefore, we analysed the in voluntary visuomotor feedback responses over the course of each experimental condition. Visually, involuntary visuomotor feedback responses in neither Experiment 1 (Fig. S8) nor Experiment 2 (Fig. S9) show any consistent learning (see Fig. S10 for statistical analysis). Therefore, it appears participants rapidly formed a partner model based on knowledge of their movement goal to modify their involuntary visuomotor feedback responses.”

Supplementary: “Supplementary Fig. S10 shows the involuntary visuomotor feedback responses in the first half (A,C) and second half (B,D) for each experimental condition. In Experiment 1, we observed the same statistical results in the first half and second half of trials as the analysis of all trials. That is, we observed a significant interaction between self-target and partner target in the first half (F[1,47] = 37.09, p < 0.001) and second half (F[1,47] = 48.68, p < 0.001) of trials. Follow-up mean comparisons showed the same significant trends as our analysis of all trials in the main manuscript (see Fig. S10A-B).”

Supplementary: “In Experiment 2, we observed the same statistical results in the first half and second half of trials as the analysis of all trials. That is, we observed a significant interaction between self-target and partner target in the first half (F[1,47] = 9.42, p = 0.004) and second half (F[1,47] = 17.40, p < 0.001) of trials. Follow-up mean comparisons showed the same significant trends as our analysis of all trials in the main manuscript (Fig. S10C-D).”

Supplementary: “Showing the same involuntary visuomotor feedback response trends across the experimental conditions for the first half, second half, and all trials suggests that the sensorimotor system quickly formed a model of a partner and considered their costs to modify rapid motor responses.”

We have also added to the discussion:

“Finally, we also considered whether time to target [1,2] (Supplementary D), participant forward hand position (Supplementary E), or learning [4] (Supplementary G) influenced feedback responses, but found that none impacted the observed differences between experimental conditions nor changed our interpretation.”

(4) The authors should also discuss some of the prior work which is very relevant to the tasks studied: (Knill, Bondata & Chhabra, 2011, J Neuroscience). There may also be other papers that use this task for visuomotor feedback responses and therefore, should be included.

We have included the Knill 2011 paper and also Cross 2019 in our discussion:

“This modification of feedback responses based on a relevant/irrelevant task goal has also been shown in response to visual perturbations [7,8].”

(5) Lines 301-303: The terms ’relevant’ and ’irrelevant’ here describe different concepts than the ones used in this study. I suggest making a distinction to avoid confusion for the reader.

We thank the reviewer for pointing out that this is confusing. We’ve made the following changes to improve the clarity:

“Further, Franklin and colleagues (2008) designed a visual perturbation to be relevant or irrelevant when reaching to the same target, showing greater involuntary visuomotor feedback responses to a relevant visual perturbation compared to an irrelevant visual perturbation [9].”

(6) Line 459: The reaching movement was quite slow (25cm in about 1.2 seconds). Is this needed to ensure that both participants can complete the movements, given potentially very different start times? Please comment as this is different than many previous studies.

Participants needed to stabilize the cursor for 500ms in their target within a time constraint of 1400 - 1600 ms. Therefore, they had to reach the target between 900 - 1100 ms (before stabilizing). Additionally, participants did not perform self-initiated movements, but were required to begin their movement as soon as they heard an audible tone. Given that reaction times are ~200ms, participants had ~700 - 900 ms to reach the target, which aligns with previous research (Franklin et al. (2008), Franklin et al. (2012), Nashed et al. (2012)). We have clarified the time constraints of the task in our Methods:

“They therefore had 700 - 900 ms to first reach the target, since humans generally have response times ~200 ms, and they needed to stabilize within the target for 500 ms (i.e., 1400 - 200 - 500 = 700 ms and 1600 - 200 - 500 = 900 ms). Movement times of 700 - 900 ms are thus consistent with previous human reaching studies [4,9,10].”

(7) Reference [25] is incomplete

Thank you for catching this.

And thank you for the thoughtful and clear review. We feel it has greatly improved the quality and clarity of our manuscript!

**Reviewer #2 (Public review):**
SummarySullivan and colleagues studied the fast, involuntary, sensorimotor feedback control in interpersonal coordination. Using a cleverly designed joint-reaching experiment that separately manipulated the accuracy demands for a pair of participants, they demonstrated that the rapid visuomotor feedback response of a human participant to a sudden visual perturbation is modulated by his/her partner’s control policy and cost. The behavioral results are well-matched with the predictions of the optimal feedback control framework implemented with the dynamic game theory model. Overall, the study provides an important and novel set of results on the fast, involuntary feedback response in human motor control, in the context of interpersonal coordination.

We thank the reviewer for the kind words!

Review:Sullivan and colleagues investigated whether fast, involuntary sensorimotor feedback control is modulated by the partner’s state (e.g., cost and control policy) during interpersonal coordination. They asked a pair of participants to make a reaching movement to control a cursor and hit a target, where the cursor’s position was a combination of each participant’s hand position. To examine fast visuomotor feedback response, the authors applied a sudden shift in either the cursor (experiment 1) or the target (experiment 2) position in the middle of movement. To test the involvement of partner’s information in the feedback response, they independently manipulated the accuracy demand for each participant by varying the lateral length of the target (i.e., a wider/narrower target has a lower/higher demand for correction when movement is perturbed). Because participants could also see their partner’s target, they could theoretically take this information (e.g., whether their partner would correct, whether their correction would help their partner, etc.) into account when responding to the sudden visual shift. Computationally, the task structure can be handled using dynamic game theory, and the partner’s feedback control policy and cost function are integrated into the optimal feedback control framework. As predicted by the model, the authors demonstrated that the rapid visuomotor feedback response to a sudden visual perturbation is modulated by the partner’s control policy and cost. When their partner’s target was narrow, they made rapid feedback corrections even when their own target was wide (no need for correction), suggesting integration of their partner’s cost function. Similarly, they made corrections to a lesser degree when both targets were narrower than when the partner’s target was wider, suggesting that the feedback correction takes the partner’s correction (i.e., feedback control policy) into account.The strength of the current paper lies in the combination of clever behavioral experiments that independently manipulate each participant’s accuracy demand and a sophisticated computational approach that integrates optimal feedback control and dynamic game theory. Both the experimental design and data analysis sound good. While the main claim is well-supported by the results, the only current weakness is the lack of discussion of limitations and an alternative explanation. Adding these points will further strengthen the paper.
**Reviewer #2 (Recommendations for the authors):**
(1) While the current version is already well-written, it would be helpful for readers to further discuss the relationship between the current study and some potentially relevant studies, such as Braun et al. (2009), Ganesh et al. (2014), and Takagi et al. (2017) (2019).

Thank you for pointing out these papers that we missed, which we now cite appropriately in light of our own work. In particular, we have added the following to our discussion, including Braun et al. (2009) and Takagi et al. (2017) (2019). However, Beckers et al. (2020) showed conflicting results from Ganesh et al. (2014), and since these works are about learning, we feel it is outside the scope of our work.

“Further, others have shown that the sensorimotor system modifies movement selection according to game-theoretic predictions, [11] and that the sensorimotor system modifies movements using an estimate of the joint goal during human-human interactions [12,13].”

(2) For an alternative interpretation of the results, one could consider, for instance, that the target’s visual appearance could have served as a contextual cue for learning different movement gains in the lateral direction (e.g., whether the partner corrects the shift might be approximated as a gain change). Although less likely, this alternative account could be tested by simulation and would strengthen the argument.

This a thoughtful comment, also brought up by Reviewer 1. Here we provide our previous response that addresses this concern. While it is possible that the change in the visuomotor feedback responses could be just from a scaling factor. This hypothesis could explain the difference between two conditions, but would fail to explain differences between two other conditions. Specifically, this hypothesis could explain a decrease in involuntary visuomotor feedback responses between partner-irrelevant/self-relevant and partner-relevant/self-relevant. Critically, this hypothesis could not explain the difference between partner-irrelevant/self-irrelevant and partner-relevant/self-irrelevant. That is, there is no reason to scale a response to correct for a partner’s relevant target when your own target is irrelevant. However, our finding that there is a greater involuntary visuomotor feedback response in partner-relevant/self-irrelevant compared to partner irrelevant/self-irrelevant is predicted by the notion that humans form a representation of others and consider their movement costs.

We have added a paragraph in the discussion to justify our hypothesis over the scaling factor hypothesis.

“Our hypothesis that the sensorimotor system uses a representation of a partner and considers the partner’s costs to modify involuntary visuomotor feedback responses can parsimoniously explain all of our experimental findings. There are a few alternative hypotheses that could explain a subset of results. One alternative hypothesis is that participants simply learned the hand to center cursor mapping in each experimental condition. That is, instead of using a model of their partner, participants simply adapted to the dynamics of the center cursor. However, this hypothesis would not predict an increased involuntary visuomotor feedback response in the partner-relevant/self-irrelevant condition compared to the partner-irrelevant/self-irrelevant condition. If participants did not form a model of their partner nor consider their partner’s costs, then they would not display an increased feedback response when they had an irrelevant target and their partner’s target was relevant. An increased feedback response to help a partner achieve their goal is captured by our hypothesis that the sensorimotor system uses a representation of a partner and considers the partner’s costs to modify involuntary visuomotor feedback responses.”

(3) Another (maybe unlikely) alternative interpretation is that the targets’ visual appearances might have been confusing. One might find that the closed square is common to both targets for the “Partner Relevant Self Irrelevant” and the “Partner Relevant Self Relevant”, and that this might have elicited the response to perturbation in “Partner Relevant Self Irrelevant”. Related to this point, it would be informative to describe how the “cooperative” fast feedback response developed over the course of the experiment, for instance, by comparing behaviors across experimental blocks.

We have partitioned this question into two responses, relating to visual appearance of the targets and the development (i.e., learning) of visuomotor feedback responses over the course of the experiments.

(1) Participants confused by visual appearance of the targets.

We were also concerned that participants might be confused by the targets, and therefore confirmed with participants after the experiment that they correctly understood that the light grey filled rectangle was their own target and the dark grey hollow rectangle was their partners. Furthermore, in the partner-relevant/self-irrelevant, partner-irrelevant/self-relevant, and partner-relevant/self-relevant conditions, there is a small square target in each of the conditions. However, we found that the partner-irrelevant/self-relevant and partner-relevant/self-relevant conditions both elicited significantly greater involuntary visuomotor feedback responses than the partner-relevant/self-irrelevant condition. Thus, participants involuntary visuomotor feedback responses suggest that they correctly formed different representations based on an accurate understanding of the self vs partner target. The other reviewer had related comments about the visual stimuli, which we also address within the discussion.

“Another alternative hypothesis would be that the sensorimotor system was responding only to the relevant target displayed on the screen. Again, this hypothesis would only explain a subset of our results. In particular, this relevant target hypothesis cannot explain the observed differences between the partner-relevant/self-irrelevant and partner-irrelevant/self-relevant conditions in both Experiments 1 and 2.”

(2) Comparing feedback responses over time

We have included the visuomotor feedback responses over each experimental condition in Supplementary G. Notably, we did not find any learning effect, suggesting that the sensorimotor system quickly developed a model of a partner’s behaviour and used that model to modify feedback responses. We have also added a paragraph on learning to our discussion.

We’ve addressed how learning did not play a role in this study:

“Finally, we also considered whether time to target [1,2] (Supplementary D), participant forward hand position (Supplementary E), or learning [4] (Supplementary G-H) influenced feedback responses, but found that none impacted the observed differences between experimental conditions nor changed our interpretation.”

Supplementary: “Given there were 151 trials and 10 left/right probe trials for each experimental condition, it is possible that completing more trials may have lead to different in voluntary visuomotor feedback responses. Therefore, we analysed the in voluntary visuomotor feedback responses over the course of each experimental condition. Visually, involuntary visuomotor feedback responses in neither Experiment 1 (Fig. S8) nor Experiment 2 (Fig. S9) show any consistent learning (see Fig. S10 for statistical analysis). Therefore, it appears participants rapidly formed a partner model based on knowledge of their movement goal to modify their involuntary visuomotor feedback responses.”

Supplementary: “Supplementary Fig. S10 shows the involuntary visuomotor feedback responses in the first half (A,C) and second half (B,D) for each experimental condition. In Experiment 1, we observed the same statistical results in the first half and second half of trials as the analysis of all trials. That is, we observed a significant interaction between self-target and partner target in the first half (F[1,47] = 37.09, p < 0.001) and second half (F[1,47] = 48.68, p < 0.001) of trials. Follow-up mean comparisons showed the same significant trends as our analysis of all trials in the main manuscript (see Fig. S10A-B).”

Supplementary: “Supplementary Fig. S10 shows the involuntary visuomotor feedback responses in the first half (A,C) and second half (B,D) for each experimental condition. In Experiment 1, we observed the same statistical results in the first half and second half of trials as the analysis of all trials. That is, we observed a significant interaction between self-target and partner target in the first half (F[1,47] = 37.09, p < 0.001) and second half (F[1,47] = 48.68, p <0.001) of trials. Follow-up mean comparisons showed the same significant trends as our analysis of all trials in the main manuscript (see Fig. S10A-B).”

Supplementary: “Showing the same involuntary visuomotor feedback response trends across the experimental conditions for the first half, second half, and all trials suggests that the sensorimotor system used a model of a partner based on their goals and considered their costs to modify rapid motor responses.”

(4) It looks slightly counter intuitive (and therefore interesting) that the participant shows some amount of fast feedback responses in the “Partner Relevant Self Irrelevant” condition, since they were instructed to only consider the self-target. Based on the results, the authors suggest an altruistic feature of the motor system (lines 333-340). It would be helpful to clarify the basis for this interpretation, whether it is formally derived from the game-theoretic framework or represents a more conceptual interpretation. Providing additional explanation that translates the game-theoretic reasoning into more accessible, intuitive terms would help readers better understand and evaluate this claim.

We are glad the reviewer also finds this result interesting. The reviewer raises an important point that there needs to be a more clear explanation for why we believe this result was found. We have made the following changes to the discussion:

“Furthermore, this result is predicted by our dynamic game theory models that include the partner’s costs in the self cost function. In other words, a dynamic game theory model that selects feedback gains to minimize both the self and partner cost reflects an altruistic control policy.”

(5) Please check whether all references are displayed correctly. Some of them (e.g., 25, 65) seemed not correctly shown in the References section.

We have fixed the citation.

We thank the reviewer for providing a clear and insightful review. Their comments have significantly improved the manuscript.

References

(1) Česonis, J., & Franklin, D. W. (2020). Time-to-Target Simplifies Optimal Control of Visuomotor Feedback Responses. eneuro, 7 (2), ENEURO.0514–19.2020.

(2) Česonis, J., & Franklin, D. W. (2022). Contextual Cues Are Not Unique for Motor Learning: Task-dependant Switching of Feedback Controllers. PLOS Computational Biology, 18 (6), ed. by Haith, A. M.: e1010192.

(3) Crevecoeur, F., Kurtzer, I., Bourke, T., & Scott, S. H. (2013). Feedback Responses Rapidly Scale with the Urgency to Correct for External Perturbations. Journal of Neurophysiology, 110 (6), 1323–1332.

(4) Franklin, S., Wolpert, D. M., & Franklin, D. W. (2012). Visuomotor Feedback Gains Upregulate during the Learning of Novel Dynamics. Journal of Neurophysiology, 108 (2), 467–478.

(5) Liu, Y., Leib, R., Dudley, W., Shafti, A., Faisal, A. A., & Franklin, D. W. (2025). Partner-Sourced Haptic Feedback Rather than Environmental Inputs Drives Coordination Improvement in Human Dyadic Collaboration. Scientific Reports, 15 (1), 40347.

(6) Franklin, D. W., Reichenbach, A., Franklin, S., & Diedrichsen, J. (2016). Temporal Evolution of Spatial Computations for Visuomotor Control. The Journal of Neuroscience, 36 (8), 2329–2341.

(7) Knill, D. C., Bondada, A., & Chhabra, M. (2011). Flexible, Task-Dependent Use of Sensory Feedback to Control Hand Movements. The Journal of Neuroscience, 31 (4), 1219–1237.

(8) Cross, K. P., Cluff, T., Takei, T., & Scott, S. H. (2019). Visual Feedback Processing of the Limb Involves Two Distinct Phases. The Journal of Neuroscience, 39 (34), 6751–6765.

(9) Franklin, D. W., & Wolpert, D. M. (2008). Specificity of Reflex Adaptation for Task-Relevant Variability. The Journal of Neuroscience, 28 (52), 14165–14175.

(10) Nashed, J. Y., Crevecoeur, F., & Scott, S. H. (2012). Influence of the Behavioral Goal and Environmental Obstacles on Rapid Feedback Responses. Journal of Neurophysiology, 108 (4), 999–1009.

(11) Braun, D. A., Ortega, P. A., & Wolpert, D. M. (2009). Nash Equilibria in Multi-Agent Motor Interactions. PLoS Computational Biology, 5 (8), ed. by Friston, K. J.: e1000468.

(10) Takagi, A., Ganesh, G., Yoshioka, T., Kawato, M., & Burdet, E. (2017). Physically Interacting Individuals Estimate the Partner’s Goal to Enhance Their Movements. Nature Human Behaviour, 1 (3), 0054.

(11) Takagi, A., Hirashima, M., Nozaki, D., & Burdet, E. (2019). Individuals Physically Interacting in a Group Rapidly Coordinate Their Movement by Estimating the Collective Goal. eLife, 8 , e41328.